

# Role of scaling dimensions in generalized noises in fractional quantum Hall tunneling due to a temperature bias

**Matteo Acciai[1,2]★, Gu Zhang[3]† and Christian Spånslätt[1,4]‡**

**1** Department of Microtechnology and Nanoscience (MC2),
Chalmers University of Technology, S-412 96 Göteborg, Sweden
**2** Scuola Internazionale Superiore di Studi Avanzati, Via Bonomea 265, 34136, Trieste, Italy
**3** Beijing Academy of Quantum Information Sciences, Beijing 100193, China
**4** Department of Engineering and Physics, Karlstad University, Karlstad, Sweden

★ macciai@sissa.it , † zhanggu@baqis.ac.cn , ‡ christian.spanslatt@kau.se

## Abstract

Continued improvement of heat control in mesoscopic conductors brings novel tools for probing strongly correlated electron phenomena. Motivated by these advances, we comprehensively study transport due to a temperature bias in a quantum point contact device in the fractional quantum Hall regime. We compute the charge-current noise (so-called delta-$T$ noise), heat-current noise, and mixed noise and elucidate how these observables can be used to infer strongly correlated properties of the device. Our main focus is the extraction of so-called scaling dimensions of the tunneling anyonic quasiparticles, of critical importance to correctly infer their anyonic exchange statistics.

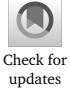

# 1 Introduction

Advancements in nanotechnology in the recent decade have paved the way towards detailed control of heat flows in small-scale electronic devices. This development permits experimental explorations of the quantum nature of heat [1], and in particular it introduces novel tools for probing quantum systems where strong electron correlations play an important role. A fundamental example is the quantum Hall effect [2,3], where in recent years it has been experimentally established that the heat conductance of the quantum Hall edge is quantized. This quantization holds both for the simpler integer [4] and for the strongly correlated fractional quantum Hall (FQH) edges [5–10], including those expected to host the elusive non-Abelian Majorana modes [11,12]. Measurements of the heat conductance provides crucial information about the edge structure, such as the number of edge channels and their chiralities: properties that are often obscured in charge conductance measurements due to strong charge equilibration. This is particularly relevant in the case of composite edges, such as the 2/3 and 5/2 FQH states. Here, the interplay of charge and thermal equilibration lengths can lead to different values of the charge conductance [13–18]. Via the bulk-boundary correspondence, access to the edge structure gives further insights into the corresponding bulk topological order [19], thereby demonstrating quantum heat transport as a powerful tool to pinpoint the topological order of FQH states.

In this paper, we analyze another possibility to probe nanoscale electronic devices with heat, namely with novel noise spectroscopy tools. We study three types of such noise tools with focus on the situation with temperature-biased contacts. The first is non-equilibrium charge-current noise in the absence of a voltage bias but instead due to a pure temperature bias. Such noise has been termed "thermally activated shot noise" or "delta-$T$ noise". While it bears some similarity to conventional voltage-bias-induced shot noise [20–22], delta-$T$ noise has the additional and quite peculiar feature of being a non-equilibrium noise arising when no net charge current flows. Delta-$T$ noise was first theoretically analyzed in diffusive conductors [23], while the first experimental observation was achieved in an atomic break junction [24], showing good agreement with the scattering theory of non-interacting electrons [20]. Since then, delta-$T$ noise has been analyzed for a broad range of systems and setups [25–45]. A second type of novel noise drawing increasing attention is heat-current noise, i.e., fluctuations in the heat current [46–57]. Such fluctuations emerge due to, e.g., thermal agitation, coupling to an electromagnetic environment, or from partitioning of heat currents due to scattering [1]. Finally, the third type of novel noise is the cross-correlation between charge and heat current fluctuations, known as "mixed noise". Mixed noise has been studied so far only theoretically for weakly interacting systems or in the presence of a voltage bias only [52,58,59].

In the context of the strongly correlated FQH effect, delta-$T$ noise was theoretically shown to disclose important properties of quasiparticles with anyonic statistics [32,37,38]. In particular, delta-$T$ noise was proposed as an experimental tool to extract the anyons' so-called scaling dimensions [60], which are important, observable parameters characterizing, e.g., the temporal decay of quasiparticle correlations. Under certain circumstances, the scaling dimensions can be further related to the FQH quasiparticle anyonic exchange statistics (a detailed discussion can be found, e.g., in Ref. [38]). As such, delta-$T$ noise holds promise as an important tool in the ongoing quest to detect and classify anyons [61–64], where an accurate identification of scaling dimensions is paramount to correctly infer anyonic exchange statistics. Also the heat-current noise due to a pure temperature bias was recently proposed to disclose scaling dimensions of FQH quasiparticles [55], an approach that does not require knowledge about the quasiparticle charges.

Making available a broad range of experimental tools to extract scaling dimensions of FQH quasiparticles is highly desirable for probing strong correlations, particularly as the originally proposed method to extract scaling dimensions from exponents of the temperature and voltage dependence of QPC tunneling conductances is highly challenging [65] (however, see Refs. [63, 66] for recent developments). As such, pushing the utility delta-$T$, heat-current, and mixed noises in the FQH effect is both important and pressing in order to disclose various phenomena related to strong electronic correlations. However, there are several missing ingredients in order to use these noises to confidently extract scaling dimensions in experiments: First, multi-terminal calculations explicitly connecting auto-correlated noise, cross-correlated noise and tunneling noise enforced by charge and energy conservation remain to be presented. Second, the similarities and differences between temperature- and voltage-biased noise have not been satisfactorily clarified. Third, the utility of mixed noise to probe FQH scaling dimensions remains unexplored. Finally, an in-depth analysis of the differences and similarities between noise in strongly correlated systems and non-interacting systems, where the latter are typically treated with a scattering approach, is absent.

In this work, we fill these gaps in the theory of temperature-biased noise and provide a comprehensive demonstration of how scaling dimensions in the FQH effect enter in novel, temperature-biased noise observables and thus how the scaling dimensions can be experimentally extracted. To this end, we provide a systematic study of temperature-biased noises generated in a quantum point contact (QPC) device in the FQH regime at Laughlin fillings $\nu = 1/(2n+1)$ (with $n$ a positive integer). Our main achievements are:

i) We perform a comprehensive calculation of the charge and heat-current noise (given in Eqs. (25) and (41), respectively) in the QPC device. Our findings not only recover previous results on auto-correlation and tunneling noise but describe also cross-correlation delta-$T$ and heat-current noise. We further provide fully analytical expressions for the small [Eqs. (27), (29), (42), (45)] and large [Eqs. (32), (36), (48)] temperature-bias limits. To the best of our knowledge, expressions for the cross-correlated noise have not been reported so far. However, an important advantage of considering cross-, rather than, auto-correlation noise is that the former vanishes in equilibrium, and therefore requires no subtraction of the thermal background noise. Moreover, our derived expressions manifest charge and energy conservation and can be used to accurately fit experimental data from both auto- and cross-correlation noise.

ii) We introduce a set of heat Fano factors (of which a single instance was previously introduced in Ref. [55]) and analyze how these quantities, given in Eq. (52), may be used to infer the scaling dimension of the tunneling particles. The heat Fano factors act as charge neutral analogues of the conventional Fano factor, used, e.g., to detect fractional charges [67–69]. The key utility of the heat Fano factors is to experimentally extract scaling dimensions without any reference to the tunneling charge, which is especially relevant for edges involving neutral modes [70–73].

iii) We introduce an effective density of states (EDOS), given in Sec. 5, for the QPC region, and thereby put strongly correlated tunneling on a similar footing with non-interacting tunneling analyzed within the scattering formalism. With this single-particle approach, we explicitly elucidate how delta-$T$ and heat-current noise in fact probe properties of the EDOS and, due to the device's temperature bias, scaling dimensions of the tunneling particles naturally enter in both delta-$T$ and heat current noise. A major benefit of this approach is that it can be straightforwardly adapted to analyze noise in related, strongly correlated systems.

iv) We provide general expressions, given in Eq. (74), for mixed charge and heat current noise and show that, close to equilibrium, the mixed noise is linked to thermoelectric conversion via the Seebeck coefficient. Our results thereby extend the utility of mixed noise in the presence of a temperature bias, previously considered only for weakly interacting electrons [58], to interacting, strongly correlated electrons. This connection provides not only a clear, physical interpretation of the mixed noise, but suggests a strategy for its experimental detection.

Together, these achievements significantly expand the scope and utility of temperature-biased noise as a novel tool to experimentally probe FQH edge physics and collective electron behavior. Moreover, our detailed calculations establish a natural starting point for modeling temperature-biased noise in related strongly correlated one-dimensional systems, such as disordered FQH line junctions [41, 74–77], disordered quantum wires [78], quantum spin Hall edges [79], quantum Hall systems in the presence of channel mixing [80] as well as similar systems in the presence of time-dependent drives [53, 81, 82].

We have organized this paper as follows: In Sec. 2, we introduce the FQH setup of interest and our theoretical formalism. In Sec. 3, we present expressions for delta-$T$ noise in the small and large bias regimes. The analogous analysis for the heat-current noise is given in Sec. 4, which includes the evaluation of the heat Fano factors. In Sec. 5, we exploit the effective density of states to elucidate the properties of noise generated by a temperature bias. After that, we derive and analyze expressions of mixed noise in Sec. 6.

For improved readability, in-depth details of our charge, heat, and mixed noise calculations are delegated to Appendix A, B, and C respectively. In Appendix D we provide a simple toy-model to highlight how scaling dimensions are modified by local interactions. We further include some useful integral identities in Appendix E and Fourier transforms of Green's functions in Appendix F. Finally, we provide a comprehensive analysis of charge- and heat-current noise for non-interacting electrons in Appendix G by using the scattering approach, calculations that we repeatedly refer to throughout the main text. As our unit convention, we generally set $\hbar = k_\mathrm{B} = 1$ throughout our calculations, but restore these quantities for major results.

## 2 Setup, conservation laws, and formalism

### 2.1 Setup and conservation laws

We study the setup in Fig. 1, consisting of two chiral quantum Hall edges bridged by a quantum point contact (QPC, indicated by the dashed line). The QPC brings the two edges in proximity and causes inter-edge charge and energy exchange. Given a temperature difference $\Delta T$ between the two source contacts, labeled by $\alpha = 1, 2$, our goal in this paper is to compute the resulting noise correlations in the two drain contacts, $\alpha = 3, 4$. We define the correlations between currents in contacts $\alpha$ and $\beta$ in terms of the symmetrized noise powers

$$S^{XX}_{\alpha\beta}(\omega) \equiv \int_{-\infty}^{+\infty} dt \big\langle \{\delta \hat{X}_\alpha(t), \delta \hat{X}_\beta(0)\} \big\rangle e^{i\omega t}, \tag{1}$$

where $\{..., ...\}$ denotes the anticommutator, $\omega$ is the frequency, and $\delta \hat{X}_\alpha(t) = \hat{X}_\alpha(t) - \langle \hat{X}_\alpha(t) \rangle$ is the operator describing the charge ($X = I$) or heat ($X = J$) fluctuations at drain $\alpha$. The operators evolve in the Heisenberg picture (see next section), and the bracket $\langle \ldots \rangle$ denotes a statistical average with respect to the local equilibrium states in the two source contacts at

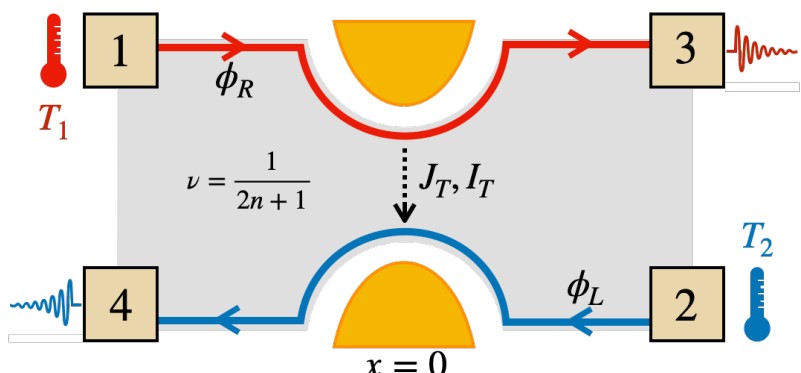

Figure 1: A quantum point contact device in the fractional quantum Hall regime at Laughlin filling $\nu = (2n+1)^{-1}$, with $n$ a positive integer. The source contacts 1 and 2 have temperatures $T_1$ and $T_2$, respectively, and inject one right ($\hat{\phi}_R$) and left ($\hat{\phi}_L$) moving edge mode at these temperatures, respectively. Tunneling of charge and heat ($I_T$ and $J_T$ respectively) between the edge modes occur at $x = 0$. In this work, we analyze the resulting charge and heat currents and their fluctuations in drain contacts 3 and 4.

$t \to -\infty$. From Eq. (1), it follows that the noise powers satisfy the symmetry relation

$$S_{\alpha\beta}^{XX}(\omega) = S_{\beta\alpha}^{XX}(-\omega).\tag{2}$$

By using conservation of charge, we relate the incoming ($\alpha = 1, 2$) and outgoing ($\alpha = 3, 4$) charge currents, $\hat{X} = \hat{I}$ in the device. Likewise, in the absence of a voltage bias in the device, $V = 0$, we can relate the incoming and outgoing heat currents by energy conservation. We thus have

$$\hat{X}_3(t) = \hat{X}_1(t) - \hat{X}_T(t),\tag{3a}$$

$$\hat{X}_4(t) = \hat{X}_2(t) + \hat{X}_T(t).\tag{3b}$$

These relations define $\hat{X}_T(t)$ as the charge ($\hat{X} = \hat{I}$) and heat ($\hat{X} = \hat{J}$) tunneling current, namely the currents leaving the upper edge and entering the lower one. By inserting Eqs. (3) into Eq. (1), we further express the noise measured in the drains in terms of the noises from the sources, or at the tunneling bridge, as

$$S_{33}^{XX}(\omega) = S_{11}^{XX}(\omega) - S_{1T}^{XX}(\omega) - S_{T1}^{XX}(\omega) + S_{TT}^{XX}(\omega),\tag{4a}$$

$$S_{44}^{XX}(\omega) = S_{22}^{XX}(\omega) + S_{2T}^{XX}(\omega) + S_{T2}^{XX}(\omega) + S_{TT}^{XX}(\omega),\tag{4b}$$

$$S_{34}^{XX}(\omega) = S_{12}^{XX}(\omega) + S_{1T}^{XX}(\omega) - S_{T2}^{XX}(\omega) - S_{TT}^{XX}(\omega),\tag{4c}$$

$$S_{43}^{XX}(\omega) = S_{21}^{XX}(\omega) + S_{T1}^{XX}(\omega) - S_{2T}^{XX}(\omega) - S_{TT}^{XX}(\omega),\tag{4d}$$

in which

$$S_{TT}^{XX}(\omega) \equiv \int_{-\infty}^{+\infty} dt \langle \{\delta\hat{X}_T(t), \delta\hat{X}_T(0)\}\rangle e^{i\omega t},\tag{5a}$$

$$S_{\alpha T}^{XX}(\omega) \equiv \int_{-\infty}^{+\infty} dt \langle \{\delta\hat{X}_T(t), \delta\hat{X}_\alpha(0)\}\rangle e^{i\omega t},\tag{5b}$$

$$S_{T\alpha}^{XX}(\omega) \equiv \int_{-\infty}^{+\infty} dt \langle \{\delta\hat{X}_\alpha(t), \delta\hat{X}_T(0)\}\rangle e^{i\omega t}.\tag{5c}$$

At zero frequency, $\omega = 0$, the charge and heat (i.e., energy) conservation (4) becomes manifest via the sum rule

$$\sum_{\alpha,\beta=3,4} S_{\alpha\beta}^{XX}(0) = S_{11}^{XX}(0) + S_{22}^{XX}(0),\tag{6}$$

where we used Eq. (2) together with $S_{12}^{XX}(\omega) = S_{21}^{XX}(\omega) = 0$, which follows since the two source current fluctuations are uncorrelated. Note that in our description, we have omitted currents and fluctuations propagating from contact 4 to contact 1 as well as from contact 3 to contact 2. In the following sections, we compute the average currents $\langle X_\alpha(t)\rangle$ and noise contributions $S_{\alpha\beta}^{XX}(\omega)$ in the FQH regime.

## 2.2 Chiral Luttinger liquid formalism

At low energies, the FQH edge dynamics is described by the chiral Luttinger model [65,83,84]. Within this model, the combined Hamiltonian of the top and bottom edge segments is given as

$$\hat{H}_0 = \frac{v_F}{4\pi} \int_{-\infty}^{+\infty} dx \left[ : (\partial_x \hat{\phi}_R)^2 : + : (\partial_x \hat{\phi}_L)^2 : \right],\tag{7}$$

in which $\hat{\phi}_{R/L}$ are bosonic field operators describing low-energy excitations propagating to the right ($R$, on the top edge) or left ($L$, on the bottom edge) with speed $v_F$. The notation ": ... :" indicates the usual normal ordering in the bosonization formalism. For notational convenience, we will omit the normal ordering symbols from now on. The bosons obey the equal-time commutation relations

$$\left[ \hat{\phi}_{R/L}(x), \hat{\phi}_{R/L}(x') \right] = \mp i\pi \text{sgn}(x - x').\tag{8}$$

By using Eq. (8) and the Heisenberg equation of motion with $\hat{H}_0$, we obtain the time evolution of the free bosonic modes $\hat{\phi}_{L,R}$ as

$$\hat{\phi}_{R/L}(x,t) = \hat{\phi}_{R/L}(x \mp v_F t),\tag{9}$$

and we see that the $R$ ($L$) boson indeed propagates to the right (left). From this chiral evolution, it follows that the time derivative reads $\partial_t = \mp v_F \partial_x$ when acting on $\hat{\phi}_{R/L}(x,t)$.

We model the QPC region, taken at $x = 0$, by the tunneling Hamiltonian

$$\hat{H}_\Lambda = \Lambda e^{iev t}\hat{\psi}_R^\dagger(0)\hat{\psi}_L(0) + \text{H.c.}\tag{10}$$

This Hamiltonian describes weak tunneling of quasiparticles with fractional charge $q^* = -\nu e$ (where $-e$ is the electron charge) and includes, for the moment, also a voltage bias $V \equiv V_1 - V_2$ between the two source contacts.[1] The operators $\hat{\psi}_{R/L}$ are quasiparticle annihilation operators related to the bosonic fields via the well-known bosonization identity

$$\hat{\psi}_{R/L}(x) = \frac{F_{R/L}}{\sqrt{2\pi a}} e^{\pm ik_F x} e^{-i\sqrt{\nu}\hat{\phi}_{R/L}(x)}.\tag{11}$$

Moreover, $\Lambda$ in (10) is the tunneling amplitude, assumed as energy-independent within all relevant energy scales. In Eq. (11), $a$ is a short-distance cutoff, $F_{R/L}$ are Klein factors, $k_F$ is the electronic Fermi momentum, and $\nu$ is the FQH filling factor. In this work, we limit

---

[1]Although our focus in this work is mainly on a pure temperature bias, we consider here the more general case with finite voltage bias $V \neq 0$, which is necessary in order to introduce the charge tunneling conductance (see Sec. 3.1) and to have a non-vanishing mixed noise (see Sec. 6).

our calculations to the Laughlin states (see, e.g., Refs. [13, 14, 37, 85, 86] for details on noise generation in QPCs for other FQH states) for which

$$\nu = \frac{1}{2n+1}, \qquad n \in \mathbb{N}^+. \tag{12}$$

In the bosonized language, the charge and heat current operators along the edges read

$$\hat{I}_{R/L} \equiv \frac{e v_F \sqrt{\nu}}{2\pi} \partial_x \hat{\phi}_{R/L}, \tag{13a}$$

$$\hat{J}_{R/L} \equiv \pm \frac{v_F^2}{4\pi} (\partial_x \hat{\phi}_{R/L})^2 - V_{1,2} \hat{I}_{R/L}, \tag{13b}$$

where $V_{1,2}$ are the voltages applied at the source contacts 1 and 2, respectively. Having defined $\hat{H}_0$ and $\hat{H}_\Lambda$, we next compute the charge and heat tunneling currents at the generic position $x_0$ along the device. To do so, we compute the time evolution of the charge and heat current operators perturbatively in $\Lambda$ up to order $|\Lambda|^2$ (amounting to the weak tunneling limit). We thus write

$$\hat{X}_{R/L}(x_0, t) = \hat{X}_{R/L}^{(0)}(x_0, t) + \hat{X}_{R/L}^{(1)}(x_0, t) + \hat{X}_{R/L}^{(2)}(x_0, t), \tag{14}$$

where the superscript (0) denotes time evolution with respect to the free Hamiltonian $\hat{H}_0$ and

$$\hat{X}_{R/L}^{(1)}(x_0, t) = -i \int_{-\infty}^{t} dt' \left[ \hat{X}_{R/L}^{(0)}(x_0, t), \hat{H}_\Lambda^{(0)}(t') \right], \tag{15a}$$

$$\hat{X}_{R/L}^{(2)}(x_0, t) = -\int_{-\infty}^{t} dt' \int_{-\infty}^{t'} dt'' \left[ \hat{H}_\Lambda^{(0)}(t''), \left[ \hat{H}_\Lambda^{(0)}(t'), \hat{X}_{R/L}^{(0)}(x_0, t) \right] \right], \tag{15b}$$

for $\hat{X} = \hat{I}, \hat{J}$. The currents $\hat{X}_{R/L}$ are related to the currents flowing *into* the drain contacts as

$$\hat{X}_3(t) = \hat{X}_R(x_3, t), \tag{16}$$

$$\hat{X}_4(t) = -\hat{X}_L(x_4, t), \tag{17}$$

where $x_3$ and $x_4$ are the locations of the drains and we adopted a convention where currents are positive when they enter the associated contact. In Secs. 3 and 4 below, we give the results for the charge and the heat transport, respectively.

# 3 Charge currents and delta-T noise

In this section, we present our results for the charge-current noise to leading order in the tunneling (10), based on Eqs. (14) and (15). Full details of our calculations are presented in Appendix A. The general expressions (25) below agree with several well-known results, see e.g., Refs. [37, 87, 88], and we have included them to make the paper self-contained. Our new results in this work are mainly the analyses of the cross-correlations, both in the small temperature bias regime —especially the explicit expressions (30)—, and in the large-bias regime (Sec. 3.3).

## 3.1 General expressions and scaling dimension

We start with the average charge tunneling current through the QPC, located at $x = 0$, which we obtain as (see Appendix A for details)

$$I_T \equiv \langle \hat{I}_T \rangle = 2ie\nu|\Lambda|^2 \int_{-\infty}^{+\infty} d\tau \sin(e\nu V\tau) G_R(\tau) G_L(\tau), \tag{18}$$

where $V = V_1 - V_2$ is the voltage difference between the source contacts and

$$G_{R/L}(\tau) \equiv G_{R/L}(x = 0, \tau) = \frac{1}{2\pi a} e^{\lambda \mathcal{G}_{R/L}(\tau)}, \tag{19}$$

are the quasiparticle Green's functions evaluated at the location of the QPC. In Eq. (19), the exponents are given in terms of equilibrium bosonic Green's functions

$$\mathcal{G}_{R/L}(\tau) = \left\langle \hat{\phi}_{R/L}(0, \tau) \hat{\phi}_{R/L}(0, 0) \right\rangle - \left\langle \hat{\phi}_{R/L}^2(0, 0) \right\rangle = \ln \left[ \frac{\sinh(i\pi T_{1/2} \tau_0)}{\sinh(\pi T_{1/2}(i\tau_0 - \tau))} \right], \tag{20}$$

with $\tau_0 \equiv a/v_F$ being the short-time cutoff. The Green's functions for the chiral right and left movers depend on $T_1$ and $T_2$, respectively (the temperatures of the two source contacts), and manifest that the edge states injected from the sources are in equilibrium with their respective contact until they reach $x = 0$.

The exponent in Eq. (19) contains also $\lambda$, which is the so-called scaling dimension of the tunneling operator [60]. This parameter can be thought of as a dynamical exponent governing the decay of the time correlation of the tunneling particles. Generally, $\lambda$ is affected by non-universal effects, e.g., inter-channel interactions [89–92], coupling to phonon modes [90], disorder [70], neutral modes [70–73], and $1/f$ noise [93,94]. For completeness, we present in Appendix D a simple toy model that showcases how scaling dimensions are modified by local density-density interactions near the QPC. We thus stress that for the Laughlin states (12), it is only in the very ideal case where such effects are absent that $\lambda$ equals the filling factor $\nu$ (in the weak backscattering regime). We further emphasize that universal, topological properties like the charge of the tunneling quasiparticles are not affected by any scaling dimension modification. In the present work, the fractional charge $q^*$ of the tunneling quasiparticles is always set by the filling factor $\nu$ via the relation $q^* = -\nu e$. Due to a well-known duality (see e.g., Ref. [32]), our calculations in the ideal weak backsattering regime can be mapped onto the ideal strong backscattering regime by taking $\lambda = 1/\nu$ and $q^* = -\nu e \to q^* = -e$.

By inspecting Eq. (18), we see that $I_T$ vanishes for $V = 0$, as expected, independently of the temperatures $T_1$ and $T_2$. This feature is a consequence of the particle-hole symmetry of the linear spectrum of the edge modes, in combination with the assumption of an energy-independent tunneling amplitude $\Lambda$. Based on the tunneling current (18), we next define the associated *differential* charge tunneling conductance as

$$\frac{\partial I_T}{\partial V} = 2i(e\nu)^2 |\Lambda|^2 \int_{-\infty}^{+\infty} d\tau \, \tau \cos(e\nu V \tau) G_R(\tau) G_L(\tau). \tag{21}$$

Close to equilibrium, i.e., for $T_1 = T_2 = \bar{T}$ and $V = 0$, we have the conductance

$$g_T(\bar{T}) \equiv \frac{\partial I_T}{\partial V} \bigg|_{\substack{V=0 \\ T_1 = T_2 = \bar{T}}} = \frac{e^2 \nu^2}{2\pi} \left( \frac{|\Lambda|}{v_F} \right)^2 (2\pi \bar{T} \tau_0)^{2\lambda - 2} \frac{\Gamma^2(\lambda)}{\Gamma(2\lambda)}, \tag{22}$$

which displays the well-known characteristic power-law scaling $\bar{T}^{2\lambda - 2}$ of the edge channels (see, e.g., Ref. [65]). In Eq. (22), $\Gamma(z)$ denotes Euler's Gamma function. In the non-interacting, integer case $\lambda = \nu = 1$, the conductance becomes

$$g_T(\bar{T})\big|_{\lambda \to 1} = \frac{e^2}{2\pi} \left( \frac{|\Lambda|}{v_F} \right)^2 = \frac{e^2}{2\pi} D, \tag{23}$$

where we defined

$$D \equiv \frac{|\Lambda|^2}{v_F^2}. \tag{24}$$

A comparison to the scattering approach for tunneling of non-interacting electrons (see Appendix G) shows that $D$ is the QPC reflection probability for this setup.

Considering next the charge-current noise, we obtain the following results for the zero-frequency charge-current noise components (finite-frequency expressions are given in Appendix A)

$$S_{11}^{II} = 2\frac{\nu e^2}{h}k_B T_1\,, \tag{25a}$$

$$S_{22}^{II} = 2\frac{\nu e^2}{h}k_B T_2\,, \tag{25b}$$

$$S_{TT}^{II} = 4(e\nu)^2|\Lambda|^2 \int_{-\infty}^{+\infty} d\tau \cos\left(\frac{e\nu V\tau}{\hbar}\right) G_R(\tau)G_L(\tau)\,, \tag{25c}$$

$$S_{33}^{II} = 2\frac{\nu e^2}{h}k_B T_1 + S_{TT}^{II} - 4\frac{\partial I_T}{\partial V}k_B T_1\,, \tag{25d}$$

$$S_{44}^{II} = 2\frac{\nu e^2}{h}k_B T_2 + S_{TT}^{II} - 4\frac{\partial I_T}{\partial V}k_B T_2\,, \tag{25e}$$

$$S_{34}^{II} = 2\frac{\partial I_T}{\partial V}k_B(T_1 + T_2) - S_{TT}^{II}\,, \tag{25f}$$

$$S_{43}^{II} = S_{34}^{II}\,. \tag{25g}$$

As a first check of the validity of these expressions, we see that indeed they fulfill the conservation equation (6). We also check the equilibrium case situation $T_1 = T_2 = \bar{T}$ and $V = 0$ which produces $S_{11}^{II} = S_{22}^{II} = S_{33}^{II} = S_{44}^{II} = 2\nu e^2 k_B\bar{T}/h$ and $S_{34}^{II} = S_{43}^{II} = 0$. These are indeed the expected equilibrium (Johnson-Nyquist) noises. The equilibrium form of $S_{TT}^{II}$ is given below in Eq. (27) and (28a).

We now move on to the main focus in this work, i.e., the non-equilibrium noise under the condition where there is no voltage bias, $V = 0$, but instead a finite temperature bias $T_1 - T_2 \neq 0$. In this case, the integrals in Eq. (25) are analytically intractable, and we therefore resort to asymptotic expansions to obtain analytical expressions for two special cases of the temperature bias. To this end, we choose a symmetric parametrization

$$T_{1,2} = \bar{T} \pm \frac{\Delta T}{2}\,, \tag{26}$$

and focus on two important regimes. In the small bias limit, we have $|\Delta T| \ll \bar{T}$ and we can expand all integrals in powers of the small parameter $\Delta T/(2\bar{T})$. In the opposite regime of a large temperature bias, one temperature is negligible compared to the other. This limit is reached for $|\Delta T| \to 2\bar{T}$. For positive $\Delta T$ we then have $T_1 \to 2\bar{T} \equiv T_{\text{hot}}$ and $T_2 \to 0$. When $\Delta T$ is negative, $T_1 \to 0$ and $T_2 \to 2\bar{T} \equiv T_{\text{hot}}$. We present results for the small and large bias limits in Secs. 3.2 and 3.3, respectively.

## 3.2 Delta-T noise for a small temperature bias

We start our charge-noise analysis with the tunneling noise $S_{TT}^{II}$ in (25c). As shown in Appendix G and further discussed in Sec. 5, for $\lambda = \nu = 1$, $S_{TT}^{II}$ coincides with the full noise of a weakly-coupled two-terminal system connected to reservoirs described by Fermi functions at temperatures $T_1$ and $T_2$, thus providing a link to standard scattering theory for non-interacting fermions.

By expanding the integrand in (25c) in powers of $\Delta T/(2\bar{T})$ and integrating term by term (see Appendix E for additional details of this approach), we obtain

$$S_{TT}^{II} = S_0^{II}\left[1 + \mathcal{C}^{(2)}\left(\frac{\Delta T}{2\bar{T}}\right)^2 + \mathcal{C}^{(4)}\left(\frac{\Delta T}{2\bar{T}}\right)^4 + \dots\right]\,, \tag{27}$$

with the prefactor and two expansion coefficients given as

$$S_0^{II} = 4g_T(\bar{T})\bar{T}\,, \tag{28a}$$

$$\mathcal{C}^{(2)} = \lambda\left\{\frac{\lambda}{1+2\lambda}\left[\frac{\pi^2}{2}-\psi^{(1)}(1+\lambda)\right]-1\right\}\,, \tag{28b}$$

$$\mathcal{C}^{(4)} = \lambda\frac{\pi^4\lambda^2(4+3\lambda)-12\pi^2\lambda(2\lambda^2+3\lambda-3)+12(4\lambda^3+4\lambda^2-5\lambda-3)}{24(4\lambda^2+8\lambda+3)}$$
$$+\lambda^2\frac{4\lambda^2+6\lambda-6-\pi^2\lambda(4+3\lambda)}{8\lambda^2+16\lambda+6}\psi^{(1)}(\lambda+1)+\lambda^3\frac{4+3\lambda}{2(4\lambda^2+8\lambda+3)}\left[\psi^{(1)}(\lambda+1)\right]^2$$
$$+\lambda^3\frac{4+3\lambda}{12(4\lambda^2+8\lambda+3)}\psi^{(3)}(\lambda+1)\,, \tag{28c}$$

where $\psi^{(n)}(z)$ are polygamma functions. These expressions confirm those previously reported in Ref. [32] for $\lambda = \nu$ and in Ref. [38] for more generic tunneling setups and scaling dimensions $\lambda$. As noted in these works, $\mathcal{C}^{(2)}$ takes negative values for $\lambda < 1/2$. Moreover, $|\mathcal{C}^{(4)}| \ll |\mathcal{C}^{(2)}|$ (see Fig. 2a), so that in the small-temperature bias limit, $|\Delta T| \ll \bar{T}$, the sign of the correction to the equilibrium term can be directly read off from the sign of the coefficient $\mathcal{C}^{(2)}$. Moreover, all odd coefficients vanish, $\mathcal{C}^{(2n+1)} = 0$, as a consequence of equal edge structures on the top and bottom edge segments, together with the choice of a symmetric temperature bias, see Eq. (26). Linear terms in $\Delta T$ can only arise for asymmetric temperature biases and/or unequal edge structures [40].

From an experimental perspective, the tunneling noise $S_{TT}^{II}$ is not directly accessible, because what is measured is either cross- or auto-correlations of current fluctuations detected in the drain contacts 3 and 4. Here, we choose to focus on the cross-correlations, as these have the advantage of being zero at equilibrium, in contrast to the auto-correlations which are finite. Before presenting the results in the FQH regime, we remark that for the integer case $\lambda = \nu = 1$, the cross-correlation $S_{34}^{II}$ coincides with the *shot noise* component in a non-interacting two-terminal system (see Appendix G). Moving on to the FQH regime, we expand the cross-correlation delta-$T$ noises (25f)-(25g) in powers of the temperature bias, integrate term by term, and obtain

$$S_{34}^{II} = S_{43}^{II} = S_0^{II}\left[(-\mathcal{C}^{(2)}+\mathcal{D}^{(2)})\left(\frac{\Delta T}{2\bar{T}}\right)^2+(-\mathcal{C}^{(4)}+\mathcal{D}^{(4)})\left(\frac{\Delta T}{2\bar{T}}\right)^4+\dots\right]. \tag{29}$$

Here, we have parametrized this noise expansion by introducing an additional set of coefficients $\mathcal{D}^{(n)}$, in which the leading ones are

$$\mathcal{D}^{(2)} = \lambda\left\{\frac{3\lambda}{1+2\lambda}\left[\frac{\pi^2}{6}-\psi^{(1)}(1+\lambda)\right]-1\right\}\,, \tag{30a}$$

$$\mathcal{D}^{(4)} = -\frac{\lambda\{12+\lambda[12+\pi^4+12(\pi^2-2)\lambda]\}}{24(1+2\lambda)}+\frac{\lambda^2(5\pi^2+18\lambda)}{6(1+2\lambda)}\psi^{(1)}(1+\lambda) \tag{30b}$$
$$-\frac{5\lambda^2}{2(1+2\lambda)}[\psi^{(1)}(1+\lambda)]^2-\frac{5\lambda^2}{12(1+2\lambda)}\psi^{(3)}(1+\lambda)$$
$$+\frac{\lambda^2(1+\lambda^2)}{8[3+4\lambda(2+\lambda)]}\left\{\pi^4-20\pi^2\psi^{(1)}(2+\lambda)+60[\psi^{(1)}(2+\lambda)]^2+10\psi^{(3)}(2+\lambda)\right\}.$$

The origin of the $\mathcal{D}^{(n)}$ coefficients can be traced to the temperature dependence of the differential charge tunneling conductance (21) which enters in Eq. (25f) and (25g), in addition to the tunneling noise $S_{TT}^{II}$. To the best of our knowledge, expressions for the the cross-correlated delta-$T$ noise and the coefficients $\mathcal{D}^{(2)}$ and $\mathcal{D}^{(4)}$ have not been reported before. Notice again

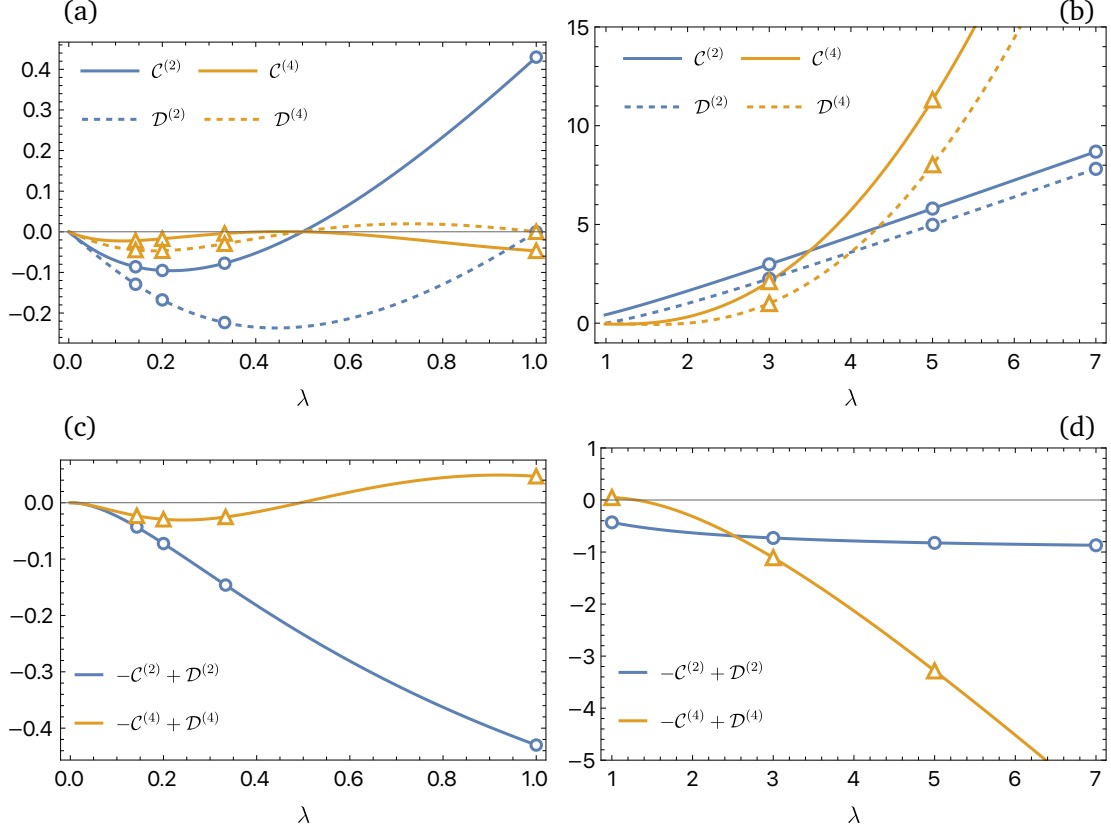

Figure 2: (a-b) Second- and fourth-order delta-$T$ noise expansion coefficients $\mathcal{C}^{(2)}$, $\mathcal{C}^{(4)}$, $\mathcal{D}^{(2)}$, and $\mathcal{D}^{(4)}$ (Eq. (28b), (28c), (30a), and (30b), respectively) as functions of the scaling dimension $\lambda$. Panels (c-d) show the difference $\mathcal{D}^{(n)} - \mathcal{C}^{(n)}$ that appears in the expansion for the full cross correlation noise (29). Triangles and circles mark the values for $\lambda = \nu$ (panels a and c) and $\lambda = 1/\nu$ (panels b and d) for fillings $\nu = 1, 1/3, 1/5, 1/7$.

the absence of terms with odd powers of $\Delta T/(2\bar{T})$ in Eq. (29) due to the symmetric setup and bias.

We plot the expansion coefficients (28b), (28c), (30a), and (30b) as functions of the scaling dimension $\lambda$ in Fig. 2(a-b). We also mark the values $\lambda = \nu$ and $\lambda = 1/\nu$ (for $\nu = 1, 1/3, 1/5, 1/7$), corresponding to ideal weak and strong backscattering limits. We thus confirm that the weak back-scattering regime for Laughlin states, i.e., $\lambda < 1/2$, produces negative delta-$T$ noise [32], $S_{TT}^{II}/S_0^{II} < 1$, since for such scaling dimensions $\mathcal{C}^{(2)} < 0$ and $|\mathcal{C}^{(4)}| < |\mathcal{C}^{(2)}|$. For $1/2 < \lambda \le 1$, we still have $|\mathcal{C}^{(4)}| < |\mathcal{C}^{(2)}|$ but $\mathcal{C}^{(2)} > 0$ so that $S_{TT}^{II}/S_0^{II} \ge 1$. In the strong back-scattering regime for Laughlin states, $\lambda > 1$, we see that $|\mathcal{C}^{(4)}| > |\mathcal{C}^{(2)}|$ for $\lambda \gtrsim 3$. For completeness, we show in Fig. 2(c-d) the behavior of the combination $-\mathcal{C}^{(n)} + \mathcal{D}^{(n)}$ (for $n = 2, 4$) that appears in the expansion of the cross-correlation noise $S_{34}^{II}$ in Eq. (29). We see that the leading-order correction is *always negative*, independently of the scaling dimension. Therefore, recalling that $S_{34}^{II} = 0$ at equilibrium, the temperature induced cross correlation noise is always negative, in contrast to the tunneling noise.

We find it further instructive to separately analyze the noise expansion terms for the special and important case of non-interacting electrons, obtained here for $\lambda = \nu = 1$. Then,

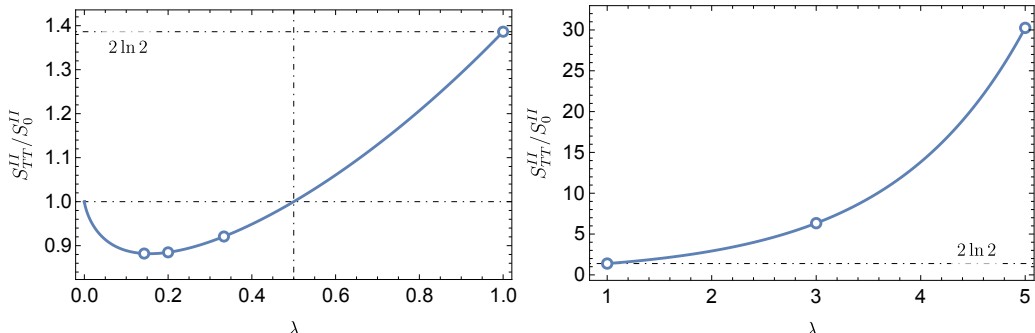

Figure 3: Tunneling delta-$T$ noise (32) in the large bias regime, normalized to the equilibrium noise $S_0^{II}$, as a function of the scaling dimension $\lambda$. Circles mark the values for $\lambda = \nu$ for $\nu = 1, 1/3, 1/5, 1/7$ (left panel) and $\lambda = 1/\nu$ for $\nu = 1, 1/3, 1/5$ (right panel). The free-electron value $2\ln 2$, given by Eq. (34), is highlighted.

the coefficients (28b), (28c), (30a), and (30b) reduce to

$$\mathcal{C}^{(2)} = \frac{\pi^2}{9} - \frac{2}{3} \approx 0.43\,, \tag{31a}$$

$$\mathcal{C}^{(4)} = -\frac{7\pi^4}{675} + \frac{\pi^2}{9} - \frac{2}{15} \approx -0.05\,, \tag{31b}$$

$$\mathcal{D}^{(2)} = 0\,, \tag{31c}$$

$$\mathcal{D}^{(4)} = 0\,, \tag{31d}$$

where $\mathcal{C}^{(2)}, \mathcal{C}^{(4)}$ are precisely those reported in Ref. [32]. The coefficients (31) may be obtained also with a scattering approach (see Appendix G). We thus deduce that the finite coefficients $\mathcal{D}^{(2)}$ and $\mathcal{D}^{(4)}$ (which both vanish for in the non-interacting case $\lambda = 1$) are a result of the strongly correlated nature of the FQH edge, due to the non-trivial temperature dependence of the differential tunneling conductance (21). In turn, this temperature dependence is a consequence of the slow power-law decay of the dynamical correlations of the tunneling particles in the FQH regime.

## 3.3 Delta-T noise for a large temperature bias

In the large bias limit, we choose $T_1 = T_{\text{hot}} \gg T_2$, effectively setting $T_2 \to 0$. Then, we find that the tunneling charge-current noise (25c) reduces to

$$S_{TT}^{II} = 4g_T(T_{\text{hot}})k_B T_{\text{hot}} \mathcal{I}_{-1}(\lambda)\,, \tag{32}$$

with the integral function

$$\mathcal{I}_n(\lambda) \equiv \frac{\Gamma(2\lambda)}{\pi^\lambda \Gamma(\lambda)^4} \int_0^{+\infty} dx\, e^{-x} x^{\lambda+n} \left| \Gamma\left(\frac{\lambda}{2} + \frac{ix}{\pi}\right) \right|^2\,. \tag{33}$$

For generic values of $\lambda$, we resort to a numerical integration of the function $\mathcal{I}_{-1}(\lambda)$ and plot the tunneling noise in Fig. 3. We observe that for scaling dimensions $\lambda < 1/2$, the non-equilibrium delta-$T$ noise is always smaller than the equilibrium contribution $S_0^{II}$. This behavior is directly linked to that of the tunneling conductance $g_T(\bar{T})$ in Eq. (22), which is a *decreasing* function of the temperature when $\lambda < 1/2$. Then, given that $T_{\text{hot}} = 2\bar{T}$ in the large bias limit [see discussion below Eq. (26)], the decrease in $g_T(T_{\text{hot}})$ is the reason why $S_{TT}^{II} < S_0^{II}$, despite that the function $\mathcal{I}_{-1}(\lambda)$ grows with $\lambda$ even for $\lambda < 1/2$.

An exact evaluation of Eq. (32) is possible for $\lambda = \nu = 1$ for which $\mathcal{I}_{-1}(1) = \ln 2$, thus reproducing the known result [29, 34, 95]

$$S_{TT}^{II} = 4D\frac{e^2}{h}k_B T_{\text{hot}}\ln 2 = 4D\frac{e^2}{h}k_B \bar{T} \times 2\ln 2, \tag{34}$$

where we reinstated $h$ and $k_B$, and identified the reflection probability $D$ from Eq. (24).

We confirm the result (34) with a scattering approach in Eq. (G.14) in Appendix G. Equation (34) can be re-written in a form which is reminiscent of a fluctuation-dissipation relation, by defining an effective noise temperature [29]

$$S_{TT}^{II} = 4D\frac{e^2}{h}k_B T_{\text{noise}}, \qquad T_{\text{noise}} \equiv T_{\text{hot}}\ln 2. \tag{35}$$

The effective noise temperature $T_{\text{noise}} = T_{\text{hot}}\ln 2$ in the large temperature bias limit has been experimentally established [29] for non-interacting electrons in a two-terminal setup. We note that a corresponding effective noise temperature in the FQH regime is not straightforward to define, as in this case the charge tunneling conductance depends on the temperature, preventing a clear separation between conductance and temperature. We point out here that Ref. [96] explored the possibility of defining an effective noise temperature associated with an effective distribution induced by the tunneling process. This requires the introduction of a second QPC (used as a detector), after which the noise is measured. We do not consider this situation here, as it goes beyond the scope of our work.

For completeness, we also present the large-bias limit of the cross-correlation noise (25f). It reads

$$\frac{S_{34}^{II}}{S_0^{II}} = -\frac{1}{2}\mathcal{I}_{-1}(\lambda) + \frac{2^{2\lambda-1}}{\pi^{\lambda+1}}\frac{\Gamma(2\lambda)}{\Gamma^4(\lambda)}\int_0^{+\infty} dx\, e^{-x}x^{\lambda-1}\left|\Gamma\left(\frac{\lambda}{2} + i\frac{x}{\pi}\right)\right|^2 \text{Im}\left[\psi^{(0)}\left(\frac{\lambda}{2} + i\frac{x}{\pi}\right)\right], \tag{36}$$

where $\mathcal{I}_{-1}(\lambda)$ is given in Eq. (33) and $\psi^{(0)}(z)$ is the digamma function. For $\lambda = 1$, the expression reduces to $S_{34}^{II} = -S_0^{II}(2\ln 2 - 1)$, corresponding (up to a sign) to the shot noise of a temperature-biased, two-terminal, non-interacting system [24, 34].

## 3.4 Full delta-T noise and comparison to asymptotic limits

We gain further insights into the delta-$T$ noise by numerically computing the full noise ratio $S_{TT}^{II}/S_0^{II}$ in Eq. (25c) and plotting it together with the asymptotic expansions (27) and (32). The result is presented in Fig. 4. The most striking feature is the very contrasting curve shape for non-interacting electrons, $\nu = \lambda = 1$, in comparison to the $\nu = \lambda = 1/3$ and $\nu = \lambda = 1/5$ FQH edge states. Whereas $1 \leq S_{TT}^{II}/S_0^{II} \leq 2\ln 2$ for $\nu = \lambda = 1$ [see Eq. (34)], this ratio is instead bounded as $S_{TT}^{II}/S_0^{II} \leq 1$ for the Laughlin edges. This feature reflects the non-trivial scaling dimension $\lambda \neq 1$ of the tunneling quasiparticles in the FQH regime [32, 37, 38]. The bounded noise in the FQH regime further highlights that the noise on top of the equilibrium one is indeed negative in this case [32], i.e., the non-equilibrium conditions *reduce* the noise compared to equilibrium.

We also observe an additional important and quite surprising feature. For $\lambda = 1, 1/3, 1/5$, the small bias expansions (27) are in fact excellent approximations within a surprisingly broad range of the temperature bias ratio $T_1/T_2$. This result suggests that for these values, the coefficients $\mathcal{C}^{(n)}$ in the expansion (27) decrease rapidly in magnitude with increasing $n$. Notably, for $\lambda = 1/3$, the leading order expansion [i.e., keeping only $\mathcal{C}^{(2)}$ in Eq. (27)] remains an excellent approximation to the full noise over two orders of magnitude of the temperature bias ratio. We anticipate that this observation will be very useful in future modelling of delta-$T$ noise for more complex FQH edge structures (see, e.g., Refs. [39, 41] for such cases). Furthermore, we

remark that the results in Fig. 4 strongly suggest that the asymptotic value (32) provides an upper bound (for any temperatures $T_1$ and $T_2$) to the tunneling noise $S_{TT}^{II}$ when $\lambda > 1/2$, but a lower bound when $\lambda < 1/2$. We leave a rigorous proof of this conjecture, along the lines of Refs. [34, 95, 97, 98], for future work.

$$\lambda = 1$$

$$\lambda = 1/3 \qquad\qquad \lambda = 1/5$$

$$\lambda = 3 \qquad\qquad \lambda = 5$$

Figure 4: Numerically computed backscattering charge-current noise $S_{TT}^{II}$, normalized to $S_0^{II}$ (solid, dark green line) for different scaling dimensions $\lambda$. The values $\lambda = 1/3, 1/5$ correspond to the ideal ones in the weak backscattering regime at fillings $\nu = 1/3, 1/5$, while $\lambda = 3, 5$ are the ideal values in the strong backscattering regime at the same filling. We also plot the small-$\Delta T$ expansions [see Eq. (27)] at second and fourth order, (light green, dashed and yellow, dashed curves, respectively). The large bias limits (32) are given as black, dot-dashed lines. The noise is plotted vs $T_1/T_2 = [1 + \Delta T/(2\bar{T})]/[1 - \Delta T/(2\bar{T})]$. Note that the large bias limit $T_1/T_2 \gg 1$ is obtained for $\Delta T \to 2\bar{T}$, $T_1 \to T_{\text{hot}}$, whereas in the opposite limit $T_1/T_2 \ll 1$, $T_2 \to T_{\text{hot}}$.

While we focused our numerical evaluation on the tunneling noise, the same analysis can be repeated for the cross correlation $S_{34}^{II}$, and we find very similar results: The first two expansion coefficients in (29) provide an excellent approximation for $S_{34}^{II}$ over an extended range of the temperature bias ratio. Moreover, the cross-correlation noise is always negative and appears to be bounded from below by the large bias limit (36) for all scaling dimensions $\lambda$.

## 4 Heat currents and heat-current noise

In this section, we analyze the heat-current noise for a pure temperature bias, without any voltage bias: $V = 0$. In the same manner as for the charge currents and the charge-current noise (see Sec. 3), we derive zero-frequency expressions for heat currents and heat-current noise (detailed calculations including finite-frequency noise expressions are presented in Appendix B, which also includes the case $V \neq 0$).

First, we obtain the average heat tunneling current in Eq. (3) as

$$J_T = -2i|\Lambda|^2 \int_{-\infty}^{+\infty} d\tau\, G_L(\tau)\partial_\tau G_R(\tau), \tag{37}$$

where the Green's functions are given in Eq. (19). In contrast to the charge tunneling current (18), we see that the average heat tunneling current is finite even for $V = 0$. Indeed, a vanishing average heat tunneling current requires also $T_1 = T_2$, i.e., no temperature bias. From Eq. (37), we next define the heat tunneling conductance

$$g_T^Q(\bar{T}) = \lim_{\Delta T \to 0} \frac{\partial J_T}{\partial \Delta T} = \frac{\pi\lambda^2}{1+2\lambda}\frac{|\Lambda|^2}{2v_F^2}\bar{T}(2\pi\bar{T}\tau_0)^{2\lambda-2}\frac{\Gamma^2(\lambda)}{\Gamma(2\lambda)} = \gamma\kappa_0\bar{T}g_T(\bar{T})\frac{2\pi}{e^2}. \tag{38}$$

Here, in the final equality, we identified the charge tunneling conductance (22), and used that $\kappa_0\bar{T} = \pi\bar{T}/6$ is the heat conductance quantum [in conventional units, $\kappa_0\bar{T} = \pi^2 k_B^2\bar{T}/(3h)$]. Moreover, the prefactor

$$\gamma = \frac{\lambda^2}{v^2} \times \frac{3}{2\lambda+1}, \tag{39}$$

characterizes the deviation from the Wiedemann-Franz law [99–101] as

$$\frac{g_T^Q(\bar{T})}{g_T(\bar{T})\bar{T}} = \gamma L_0, \tag{40}$$

where $L_0 = (\pi^2/3)(k_B/e)^2$ is the Lorenz number. The deviation from the Wiedemann-Franz law ($\gamma \neq 1$) in the FQH regime highlights that charge and heat are not carried by free electrons in the QPC tunneling, but instead by fractionalized quasiparticles.

Next, we obtain the zero-frequency heat-current noise components as

$$S_{11}^{JJ} = 2\frac{\pi^2 k_B^3}{3h}T_1^3, \tag{41a}$$

$$S_{22}^{JJ} = 2\frac{\pi^2 k_B^3}{3h}T_2^3, \tag{41b}$$

$$S_{TT}^{JJ} = 4|\Lambda|^2 \int_{-\infty}^{+\infty} d\tau\, \partial_\tau G_R(\tau)\partial_\tau G_L(\tau), \tag{41c}$$

$$S_{33}^{JJ} = S_{11}^{JJ} + S_{TT}^{JJ} - 4k_B\lambda T_1 J_T - 8i|\Lambda|^2 k_B T_1 \int_{-\infty}^{+\infty} d\tau\, \tau\, \partial_\tau G_R(\tau)\partial_\tau G_L(\tau), \tag{41d}$$

$$S_{44}^{JJ} = S_{22}^{JJ} + S_{TT}^{JJ} + 4k_{\rm B}\lambda T_2 J_T - 8i|\Lambda|^2 k_{\rm B} T_2 \int_{-\infty}^{+\infty} d\tau\, \tau\, \partial_\tau G_L(\tau)\, \partial_\tau G_R(\tau), \tag{41e}$$

$$S_{34}^{JJ} = -S_{TT}^{JJ} + 2\lambda k_{\rm B}(T_1 - T_2) J_T + 4i|\Lambda|^2 k_{\rm B}(T_1 + T_2) \int_{-\infty}^{+\infty} d\tau\, \tau\, \partial_\tau G_R(\tau)\, \partial_\tau G_L(\tau), \tag{41f}$$

$$S_{43}^{JJ} = S_{34}^{JJ}. \tag{41g}$$

By plugging these expressions into Eq. (6), we see that they satisfy energy conservation. Next, we evaluate the expressions (41) for equilibrium $T_1 = T_2 = \bar{T}$. We then have $S_{11}^{JJ} = S_{22}^{JJ} = S_{33}^{JJ} = S_{44}^{JJ} = 2\kappa_0 k_{\rm B}\bar{T}^3$, $S_{34}^{JJ} = S_{43}^{JJ} = 0$, and $S_{TT}^{JJ} = 4G_T^Q(\bar{T})\bar{T}^2$, which are precisely the expected equilibrium expressions [48,102]. We also have that for $\lambda = 1$, Eqs. (41) correctly reduce to the expressions for non-interacting electrons, obtained within scattering theory.

In the following subsections, we consider, just as for the delta-$T$ noise in Sec. 3, the two analytically tractable limits of small and large temperature biases. The results are presented below in Secs. 4.1 and 4.2, respectively.

## 4.1 Heat-current noise for small temperature bias

In the small temperature bias regime, $\Delta T \ll \bar{T}$ with $T_{1,2} = \bar{T} \pm \Delta T/2$, we expand the heat tunneling noise (41c) in powers of $\Delta T/(2\bar{T})$, and integrate term by term. We then find

$$S_{TT}^{JJ} = S_0^{JJ}\left[1 + \mathcal{C}_Q^{(2)}\left(\frac{\Delta T}{2\bar{T}}\right)^2 + \mathcal{C}_Q^{(4)}\left(\frac{\Delta T}{2\bar{T}}\right)^4 + \dots\right], \tag{42}$$

where the zeroth order, or equilibrium, heat tunneling noise reads

$$S_0^{JJ} = \frac{2\pi\lambda^2}{1 + 2\lambda}\frac{|\Lambda|^2}{v_F^2}\bar{T}^3 (2\pi\bar{T}\tau_0)^{2\lambda-2}\frac{\Gamma^2(\lambda)}{\Gamma(2\lambda)} = 4g_T^Q(\bar{T})\bar{T}^2, \tag{43}$$

where we identified the heat tunneling conductance Eq. (38) in the final equality. Equation (43) manifests the fluctuation-dissipation theorem for zero-frequency heat transport [48, 102].

The heat-current noise expansion coefficients in Eq. (42) read

$$\mathcal{C}_Q^{(2)} = \frac{\left(\pi^2(3\lambda + 4) - 2(2\lambda + 7)\right)\lambda^2 - 2(3\lambda + 4)\lambda^2\psi^{(1)}(\lambda) + 8}{2\lambda(2\lambda + 3)}, \tag{44a}$$

$$\begin{aligned}\mathcal{C}_Q^{(4)} =\ & \frac{\lambda\{12[(1 + 2\lambda)(2\lambda^2 + 13\lambda + 23) - \pi^2(2 + \lambda)(6\lambda^2 + 23\lambda - 10)]\}}{24(3 + 2\lambda)(5 + 2\lambda)} \\ &+ \frac{\lambda\pi^4(15\lambda^3 + 60\lambda^2 + 64\lambda + 16)}{24(3 + 2\lambda)(5 + 2\lambda)} \\ &- \frac{\lambda[\pi^2(15\lambda^3 + 60\lambda^2 + 64\lambda + 16) - 2(2 + \lambda)(6\lambda^2 + 23\lambda - 10)]}{2(3 + 2\lambda)(5 + 2\lambda)}\psi^{(1)}(1 + \lambda) \\ &+ \frac{\lambda(15\lambda^3 + 60\lambda^2 + 64\lambda + 16)}{2(3 + 2\lambda)(5 + 2\lambda)}[\psi^{(1)}(1 + \lambda)]^2 + \frac{\lambda(15\lambda^3 + 60\lambda^2 + 64\lambda + 16)}{12(3 + 2\lambda)(5 + 2\lambda)}\psi^{(3)}(1 + \lambda).\end{aligned} \tag{44b}$$

We plot these coefficients in Fig. 5. We see that the coefficient $\mathcal{C}_Q^{(2)}$ changes its sign at $\lambda = \lambda^* \approx 0.28$ which, somewhat surprisingly, shows that $\mathcal{C}_Q^{(2)} < 0$ for all ideal Laughlin states, *except* $\nu = 1/3$ for which it is positive. This feature stands in contrast to the charge tunneling noise expansion coefficient $\mathcal{C}^{(2)}$ (see Eq. (28b) and the discussion below it), which is negative for all Laughlin states. However, we believe that this different behavior has no deeper meaning and, in particular, it does not imply any fundamental differences between the 1/3 state and

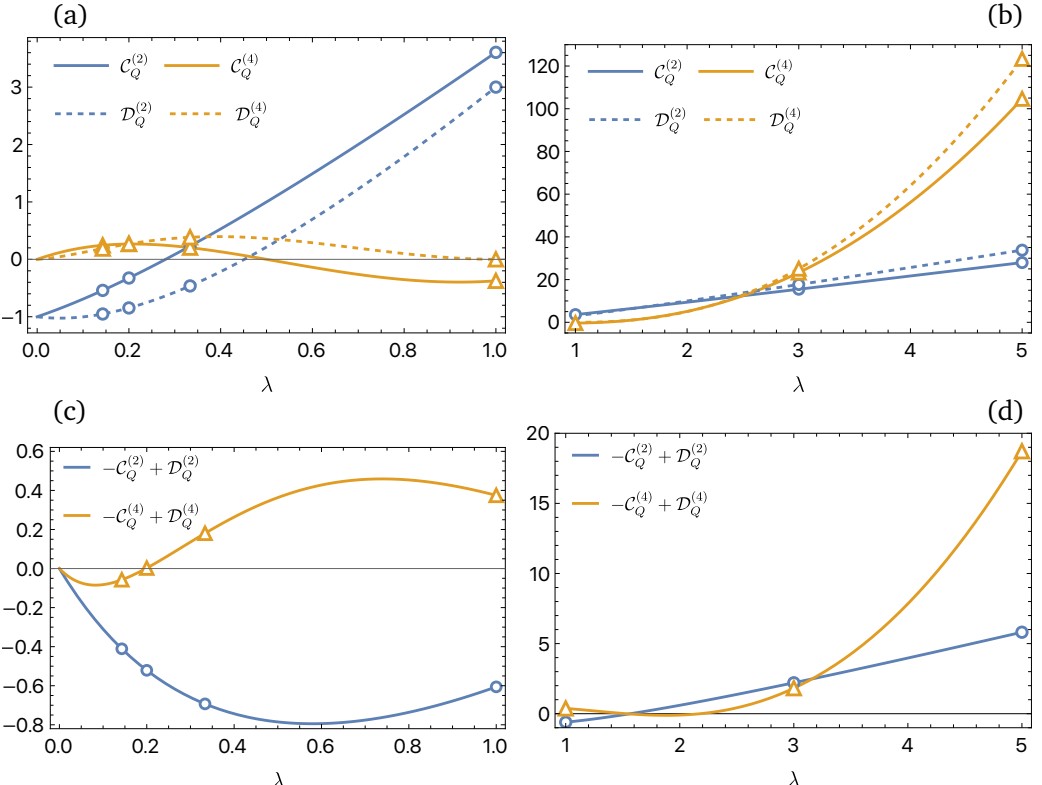

Figure 5: Second- and fourth-order delta-$T$ noise expansion coefficients $\mathcal{C}_Q^{(2)}$, $\mathcal{C}_Q^{(4)}$, $\mathcal{D}_Q^{(2)}$, and $\mathcal{D}_Q^{(4)}$ (Eq. (47a), (47b), (47c), and (47d), respectively) as functions of the scaling dimension $\lambda$. Triangles and circles mark the values for $\lambda = 1, 1/3, 1/5, 1/7$ (panels a and c) and $\lambda = 1, 3, 5$ (panels b and d).

the other Laughlin states. Rather, the difference between the delta-$T$ and heat-current noise is their different dependence on the scaling dimensions. Ultimately, this feature is related to the fact that the transported heat depends on the energy at which it is transferred, while the charge does not [compare in particular Eqs. (68) and (71) in Sec. 5 below]. In turn, the scaling dimension dependency affects the results of those integrals that arise when the noises are expanded in powers of $\Delta T$.

Moving on to cross correlation heat-current noise (41f), we obtain the expansion

$$S_{34}^{JJ} = S_{43}^{JJ} = S_0^{JJ}\left[(-\mathcal{C}_Q^{(2)} + \mathcal{D}_Q^{(2)})\left(\frac{\Delta T}{2\bar{T}}\right)^2 + (-\mathcal{C}_Q^{(4)} + \mathcal{D}_Q^{(4)})\left(\frac{\Delta T}{2\bar{T}}\right)^4 + \dots\right], \quad (45)$$

with the additional coefficients

$$\mathcal{D}_Q^{(2)} = \frac{\lambda(4+3\lambda)[\pi^2 - 6\psi^{(1)}(1+\lambda)] + 2(1+2\lambda)(\lambda-3)}{2(3+2\lambda)}, \quad (46a)$$

$$\mathcal{D}_Q^{(4)} = \frac{3\lambda(1+2\lambda)(5-5\lambda-2\lambda^2)}{2(3+2\lambda)(5+2\lambda)} + \frac{\lambda(6\lambda^3+71\lambda^2+54\lambda-140)}{6(3+2\lambda)(5+2\lambda)}\left[6\psi^{(1)}(1+\lambda) - \pi^2\right]$$

$$+ \frac{\pi^2\lambda(16+64\lambda+60\lambda^2+15\lambda^3)}{24(3+2\lambda)(5+2\lambda)}\left[\pi^2 - 20\psi^{(1)}(1+\lambda)\right]$$

$$+ \frac{5\lambda(16+64\lambda+60\lambda^2+15\lambda^3)\{\psi^{(3)}(1+\lambda) + 6[\psi^{(1)}(1+\lambda)]^2\}}{12(3+2\lambda)(5+2\lambda)}. \quad (46b)$$

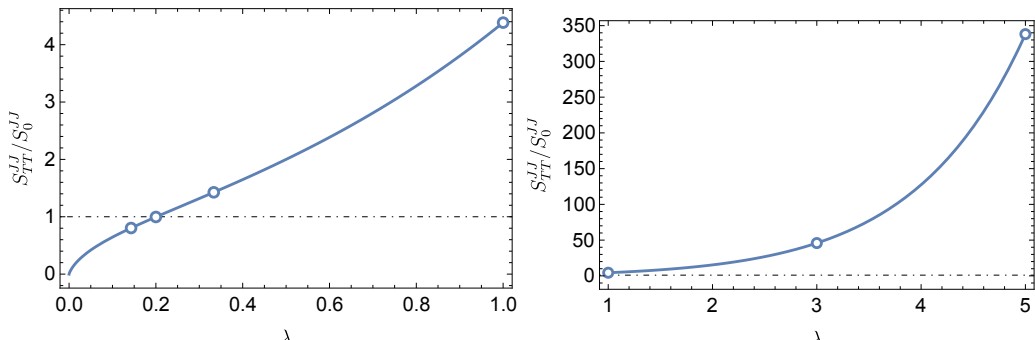

Figure 6: Tunneling heat delta-$T$ noise (48) in the large bias regime, normalized to the equilibrium noise $S_0^{JJ}$ in Eq. (43), as a function of the scaling dimension $\lambda$. Circles mark the values for $\lambda = \nu$ for $\nu = 1, 1/3, 1/5, 1/7$ (left panel) and $\lambda = 1/\nu$ for $\nu = 1, 1/3, 1/5$ (right panel).

For non-interacting electrons $\lambda = 1$, the expansion coefficients reduce to

$$\mathcal{C}_Q^{(2)} = \frac{1}{15}(7\pi^2 - 15) \approx 3.6\,, \tag{47a}$$

$$\mathcal{C}_Q^{(4)} = 2\pi^2\left(\frac{7}{15} - \frac{31}{630}\pi^2\right) \approx -0.37\,, \tag{47b}$$

$$\mathcal{D}_Q^{(2)} = 3\,, \tag{47c}$$

$$\mathcal{D}_Q^{(4)} = 0\,, \tag{47d}$$

in full agreement with the scattering approach, see Appendix G. Importantly, as shown in the bottom panels of Fig. 5, the leading-order cross correlation expansion coefficient in Eq. (45), i.e., $-\mathcal{C}_Q^{(2)} + \mathcal{D}_Q^{(2)}$ is *always* negative for all scaling dimensions $\lambda \leq 1$. In particular, it has the same sign for all ideal Laughlin states, in contrast to the auto-correlation coefficient $\mathcal{C}_Q^{(2)}$, which may change sign as discussed above.

## 4.2 Heat-current noise for large temperature bias

Here, we consider the heat-current noise in the large bias limit $T_1 = T_{\text{hot}} \gg T_2$, so that the cold temperature can effectively be set to $T_2 \to 0$. In this limit, we obtain the heat tunneling noise (41c) as

$$S_{TT}^{JJ} = 4(k_B T_{\text{hot}})^2 g_T^Q(T_{\text{hot}})\frac{8}{\pi^2}\frac{1+2\lambda}{\lambda^2}\mathcal{I}_1(\lambda)\,, \tag{48}$$

with $\mathcal{I}_1(\lambda)$ given in Eq. (33). We have not been able to evaluate this integral analytically for generic $\lambda$, but for $\lambda = 1$ we find

$$S_{TT}^{JJ} = \frac{|\Lambda|^2}{v_F^2}\frac{8T_{\text{hot}}^3}{\pi^2}\frac{3}{8}\pi\zeta(3) = \frac{3}{\pi}D\zeta(3)T_{\text{hot}}^3\,, \tag{49}$$

where $\zeta(z)$ is the Riemann zeta function with $\zeta(3) \approx 1.2$. In the final equality in Eq. (49), we identified the QPC reflection probability $D$ from Eq. (24). The expression (49) is equivalent to that which we obtain with a scattering approach (see Appendix G). The evolution of the asymptotic value (48) as a function of the scaling dimension is shown in Fig. 6.

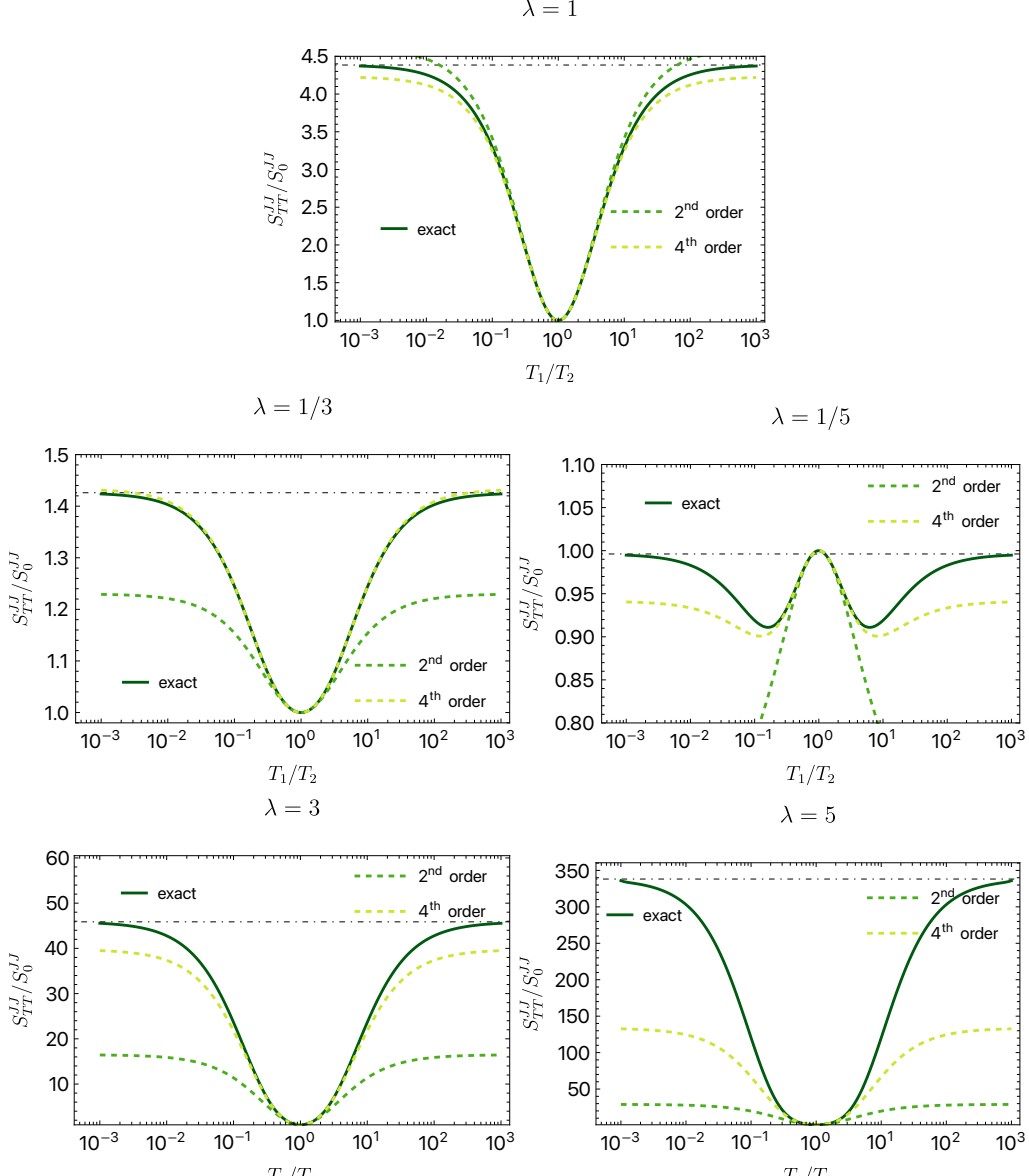

Figure 7: Numerically computed backscattering heat-current noise (solid, green line), normalized to $S_0^{JJ}$ for different scaling dimensions $\lambda$. The values $\lambda = 1/3, 1/5$ correspond to the ideal ones in the weak backscattering regime at fillings $\nu = 1/3, 1/5$, while $\lambda = 3, 5$ are the ideal values in the strong backscattering regime at the same filling. We also plot the small-$\Delta T$ expansions [see Eq. (42)] to second and fourth order (green, dashed and yellow, dashed curves, respectively). The large bias limits (48) are given as black, dot-dashed lines. The noise is plotted vs $T_1/T_2 = [1 + \Delta T/(2\bar{T})]/[1 - \Delta T/(2\bar{T})]$. Note that for $T_1/T_2 \gg 1$, $T_1 \to T_{\text{hot}}$, whereas in the opposite limit $T_1/T_2 \ll 1$, $T_2 \to T_{\text{hot}}$.

## 4.3 Full heat-current noise and comparison to asymptotic limits

Here, we numerically compute the noise ratio $S_{TT}^{JJ}/S_0^{JJ}$ and plot it together with the asymptotic limits (42) and (48) in Fig. 7. We first note the very contrasting behavior between $\nu = \lambda = 1$ and the Laughlin states with $\lambda < 1/3$. This feature reflects the distinct scaling dimension dependence of the tunneling heat-current noise for $\lambda > \lambda^\star$ and $\lambda < \lambda^\star$, where $\lambda^\star \approx 0.28$

marks the value where the dominant $\mathcal{C}_Q^{(2)}$ coefficient changes sign (see Sec. 4.1). We also see that for $\lambda = \nu = 1$ and $\nu = 1/3$, keeping four orders in the small bias expansion (42) is enough to quite accurately capture the tunneling heat-current noise over a very broad range of temperatures. In contrast, for $\lambda = 1/5$, terms beyond the fourth order are required for an accurate approximation.

Another crucial difference in comparison to the charge tunneling noise is that, below the scaling dimension $\lambda^\star$ (for which $\mathcal{C}_Q^{(2)} = 0$), the tunneling heat noise displays a non-monotonic behavior as a function of the temperature ratio $T_1/T_2$, particularly pronounced in Fig. 7 for $\lambda = 1/5$. Such features are absent in the charge tunneling noise $S_{TT}^{II}$. The non-monotonic behavior of the tunneling heat noise allows us to conclude that the asymptotic large bias expression in Eq. (48) is neither an upper nor a lower bound on the heat tunneling noise when $\lambda < \lambda^\star \approx 0.28$.

The conclusion of the above analysis is that the heat-current noise has a scaling dimension dependence that is quite distinct from the delta-$T$ noise. As elaborated above, this follows since the heat transferred across the QPC depends on the energy at which it occurs while the charge transfer does not. Still, as detailed in the next section and in the same spirit of Ref. [37], it is possible to use heat-current fluctuations to define Fano factors [55] that allow an extraction of the scaling dimension, thereby eliminating additional non-universal effects possibly present in the tunneling amplitude.

## 4.4 Generalized heat Fano factors

In Ref. [55], for the setup in Fig. 1, the authors define a "heat Fano factor" as

$$\mathcal{F}^J \equiv \frac{\Delta S_{33}^{JJ}}{2J_T}, \tag{50}$$

where $\Delta S_{33}^{JJ} \equiv S_{33}^{JJ} - S_{11}^{JJ}$ is the excess heat-current noise in drain contact 3. The Fano factor (50) can be viewed as a heat transport analogue of the usual Fano factor in weak FQH tunneling used to detect fractional charges [67–69]. In contrast with the standard Fano factor, which involves both the scaling dimension and the charge of the tunneling quasiparticles [37], the heat Fano factor has the advantage of providing a way to extract the scaling dimension without any reference to the charge of the tunneling quasiparticles, thus providing a very appealing complementary tool for investigating complex FQH edge structures, especially those involving neutral modes [70–73]. In the small temperature bias regime, with the parametrization $T_1 = T_{\text{cold}}$ and $T_2 = T_{\text{cold}} + \Delta T$, Ref. [55] reports that the heat Fano factor evaluates to

$$\mathcal{F}^J = (2\lambda + 1)T_{\text{cold}} + \mathcal{O}\left(\frac{\Delta T}{T_{\text{cold}}}\right), \tag{51}$$

thereby providing a measure of the scaling dimension $\lambda$. The result (51) follows as both $\Delta S_{33}^{JJ}$ and the tunneling current $J_T$ are linear in $\Delta T$ to leading order.

In this section, we generalize the Fano factor (50) by introducing additional heat Fano factors as

$$\mathcal{F}_{\alpha\beta}^J = \frac{\Delta S_{\alpha\beta}^{JJ}}{2J_T}, \qquad \alpha, \beta = 3, 4, \tag{52}$$

where $\Delta S_{\alpha\beta}^{JJ}$ are excess heat-current noises, in which the equilibrium contributions, if present, are subtracted. More specifically, we have $\Delta S_{44}^{JJ} \equiv S_{44}^{JJ} - S_{22}^{JJ}$ and $\Delta S_{34}^{JJ} = \Delta S_{43}^{JJ} \equiv S_{43}^{JJ}$, since the cross-correlation heat-current noises vanish in equilibrium. Due to energy conservation, Eq. (6) dictates that, in the absence of voltage bias,

$$\mathcal{F}_{44}^J + \mathcal{F}_{33}^J + 2\mathcal{F}_{34}^J = 0, \tag{53}$$

so that there are only two independent heat Fano factors. Moreover, the explicit expressions for the heat Fano factors may depend on the chosen parametrization of the temperature biases. To investigate this, we next derive explicit results for the generic heat Fano factors (52) for different parametrizations and temperature bias strengths.

### 4.4.1 Small bias regime

**Symmetric temperature bias:** Here, we choose the symmetric temperature bias parametrization (26). We then expand the heat tunneling current (37) to leading order in $\Delta T/(2\bar{T}) \ll 1$ and find

$$J_T = S_0^{JJ} \times \frac{1}{2\bar{T}} \frac{\Delta T}{2\bar{T}} + \mathcal{O}\left[\left(\frac{\Delta T}{2\bar{T}}\right)^2\right], \tag{54}$$

where $S_0^{JJ} = 4g_T^Q(\bar{T})\bar{T}^2$ is the equilibrium heat tunneling noise (43). Combining Eq. (54) with the expanded cross-correlation heat-current noise (45), we obtain the "crossed" heat Fano factor as

$$\mathcal{F}_{34}^J = \frac{1}{2}\left[-\mathcal{C}_Q^{(2)} + \mathcal{D}_Q^{(2)}\right]\Delta T, \tag{55}$$

with the scaling-dimension-dependent coefficients $\mathcal{C}_Q^{(2)}$ and $\mathcal{D}_Q^{(2)}$ given in Eq. (44a) and (46a), respectively. We see that the Fano factor (55) depends on the temperature difference $\Delta T$, in contrast with Eq. (51) which was derived in Ref. [55]. The reason for this is that the excess auto-correlations satisfy $\Delta S_{33}^{JJ} = -\Delta S_{44}^{JJ}$ to linear order in $\Delta T$. This observation, combined with the sum rule (53), shows that keeping second-order terms in $\Delta T$ is required to get a finite Fano factor for the cross correlations. Explicitly, we find

$$\Delta S_{33}^{JJ} = S_0^{JJ}\left\{-(2\lambda+1)\left(\frac{\Delta T}{2\bar{T}}\right) + \left[\mathcal{C}_Q^{(2)} - \mathcal{D}_Q^{(2)}\right]\left(\frac{\Delta T}{2\bar{T}}\right)^2\right\}, \tag{56a}$$

$$\Delta S_{44}^{JJ} = S_0^{JJ}\left\{+(2\lambda+1)\left(\frac{\Delta T}{2\bar{T}}\right) + \left[\mathcal{C}_Q^{(2)} - \mathcal{D}_Q^{(2)}\right]\left(\frac{\Delta T}{2\bar{T}}\right)^2\right\}, \tag{56b}$$

which upon division with $2J_T$ from Eq. (54) results in the two additional heat Fano factors

$$\mathcal{F}_{33}^J = -(2\lambda+1)\bar{T} + \frac{1}{2}\left[\mathcal{C}_Q^{(2)} - \mathcal{D}_Q^{(2)}\right]\Delta T, \tag{57a}$$

$$\mathcal{F}_{44}^J = +(2\lambda+1)\bar{T} + \frac{1}{2}\left[\mathcal{C}_Q^{(2)} - \mathcal{D}_Q^{(2)}\right]\Delta T. \tag{57b}$$

For non-interacting electrons, $\lambda = 1$, we find for the symmetric bias

$$\mathcal{F}_{33}^J\big|_{\lambda=1} = -3\bar{T} - \left(2 - \frac{7\pi^2}{30}\right)\Delta T, \tag{58a}$$

$$\mathcal{F}_{44}^J\big|_{\lambda=1} = +3\bar{T} - \left(2 - \frac{7\pi^2}{30}\right)\Delta T, \tag{58b}$$

$$\mathcal{F}_{34}^J\big|_{\lambda=1} = \left(2 - \frac{7\pi^2}{30}\right)\Delta T. \tag{58c}$$

**Asymmetric temperature bias:** Here, we pick the alternative asymmetric bias parametrization $T_1 = T_{\text{cold}} + \Delta T$ and $T_2 = T_{\text{cold}}$. Noticing that $\bar{T} = T_{\text{cold}} + \Delta T/2$, and keeping terms up to

second order in $\Delta T$ in expressions found in Eq. (55) and Eq. (57), we obtain

$$\mathcal{F}^J_{33} = -(2\lambda + 1)T_{\text{cold}} + \frac{1}{2}\left[\mathcal{C}^{(2)}_Q - \mathcal{D}^{(2)}_Q - (1 + 2\lambda)\right]\Delta T, \tag{59a}$$

$$\mathcal{F}^J_{44} = +(2\lambda + 1)T_{\text{cold}} + \frac{1}{2}\left[\mathcal{C}^{(2)}_Q - \mathcal{D}^{(2)}_Q + (1 + 2\lambda)\right]\Delta T, \tag{59b}$$

$$\mathcal{F}^J_{34} = \mathcal{F}^J_{43} = \frac{1}{2}\left[-\mathcal{C}^{(2)}_Q + \mathcal{D}^{(2)}_Q\right]\Delta T, \tag{59c}$$

which thus extends the Fano factor from Ref. [55] with a correction that is linear in $\Delta T$. Note that an explicit calculation of the Fano factors with the asymmetric parametrization requires an expansion to second order in $\Delta T$ also for the tunneling current. We also remark that the opposite sign in the leading term of $\mathcal{F}^J_{33}$ compared to the result (50) in Ref. [55] follows from the fact that the authors choose $T_1$ as the coldest temperature, which leads to a sign change in the tunneling current. For $\lambda = 1$, we have for the asymmetric bias

$$\mathcal{F}^J_{33}\big|_{\lambda=1} = -3T_{\text{cold}} + \left(\frac{7\pi^2}{30} - 5\right)\Delta T, \tag{60a}$$

$$\mathcal{F}^J_{44}\big|_{\lambda=1} = +3T_{\text{cold}} + \left(\frac{7\pi^2}{30} + 1\right)\Delta T, \tag{60b}$$

$$\mathcal{F}^J_{34}\big|_{\lambda=1} = \left(2 - \frac{7\pi^2}{30}\right)\Delta T. \tag{60c}$$

### 4.4.2 Large bias regime

For the large temperature bias, we take $T_1 = T_{\text{hot}}$ and $T_2 \to 0$. Then, the heat-current noises (41d)-(41f) simplify to

$$\Delta S^{JJ}_{33} = S^{JJ}_{TT} - 8\lambda T_{\text{hot}} J_T, \tag{61a}$$

$$\Delta S^{JJ}_{44} = S^{JJ}_{TT}, \tag{61b}$$

$$\Delta S^{JJ}_{34} = -S^{JJ}_{TT} + 4\lambda T_{\text{hot}} J_T. \tag{61c}$$

Plugging into these expressions the heat tunneling current (37) in the large bias regime,

$$J_T = T_{\text{hot}} g^Q_T(T_{\text{hot}}) \frac{4}{\pi^2} \frac{1 + 2\lambda}{\lambda^2} \mathcal{I}_0(\lambda), \tag{62}$$

and the tunneling heat-current noise $S^{JJ}_{TT}$ from Eq. (48), we find

$$\mathcal{F}^J_{33} = 2T_{\text{hot}}\left[\frac{\mathcal{I}_1(\lambda)}{\mathcal{I}_0(\lambda)} - 2\lambda\right], \tag{63a}$$

$$\mathcal{F}^J_{44} = 2T_{\text{hot}}\frac{\mathcal{I}_1(\lambda)}{\mathcal{I}_0(\lambda)}, \tag{63b}$$

$$\mathcal{F}^J_{34} = 2T_{\text{hot}}\left[\lambda - \frac{\mathcal{I}_1(\lambda)}{\mathcal{I}_0(\lambda)}\right], \tag{63c}$$

with the integral functions $\mathcal{I}_n(\lambda)$ from Eq. (33). For free electrons, the large bias heat Fano factors reduce to

$$\mathcal{F}^J_{33}\big|_{\lambda=1} = 2T_{\text{hot}}\left[\frac{9\zeta(3)}{\pi^2} - 2\right] \approx -1.8T_{\text{hot}}, \tag{64a}$$

$$\mathcal{F}^J_{44}\big|_{\lambda=1} = 2T_{\text{hot}}\left[\frac{9\zeta(3)}{\pi^2}\right] \approx 2.2T_{\text{hot}}, \tag{64b}$$

$$\mathcal{F}^J_{34}\big|_{\lambda=1} = 2T_{\text{hot}}\left[1 - \frac{9\zeta(3)}{\pi^2}\right] \approx -0.2T_{\text{hot}}. \tag{64c}$$

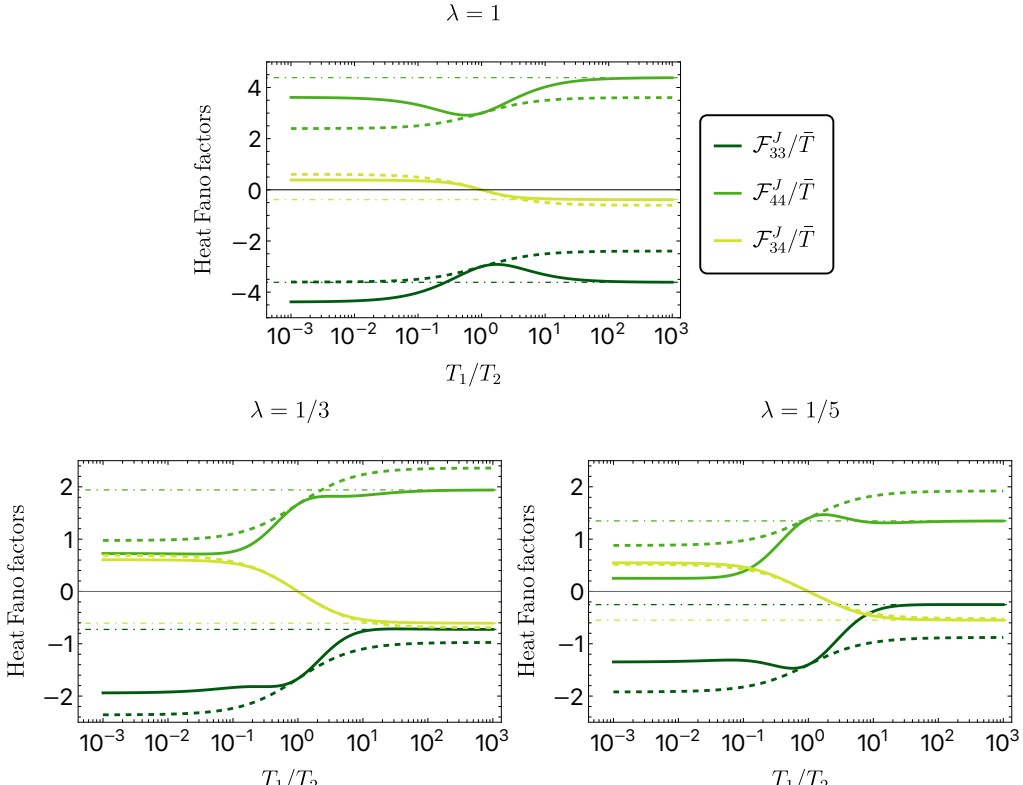

Figure 8: Numerically computed heat Fano factors normalized to $\bar{T} = (T_1 + T_2)/2$, for different scaling dimensions $\lambda$. The full lines are the exact results for $\mathcal{F}_{33}^J$, $\mathcal{F}_{44}^J$, and $\mathcal{F}_{34}^J$, while the dashed lines refer to the small-$\Delta T$ results (55) and (57). The large bias limits (63) are shown as horizontal, dot-dashed lines. The Fano factors are plotted as a function of $T_1/T_2 = [1 + \Delta T/(2\bar{T})]/[1 - \Delta T/(2\bar{T})]$. The legend in the box applies to all plots.

We note that the different form of $\mathcal{F}_{33}^J$ and $\mathcal{F}_{44}^J$ is simply due to the chosen bias parametrization. By inverting the temperature bias (i.e., taking instead $T_1 \to 0$ and $T_2 = T_{\text{hot}}$), we simply get $\mathcal{F}_{33}^J \leftrightarrow -\mathcal{F}_{44}^J$, while the cross-correlation noise, $\mathcal{F}_{34}^J$, does not change. This feature is very distinct from voltage-biased charge-current noise, where the noise and Fano factor depend on the voltage *difference* between the source contacts. Our results in this subsection thus highlight that temperature biased induced noise behaves very differently, as there is no corresponding "gauge invariance" for the temperature bias.

Just as for the noise, it is instructive to compare the derived asymptotic limits for the Fano factors with the exact results obtained by numerical integration of both the tunneling current and the noise. We plot the exact results for all Fano factors as a function of $T_1/T_2$ in Fig. 8, together with the asymptotic expressions that we have derived in the previous sections. As expected, $\mathcal{F}_{34}^J$ vanishes when $T_1 = T_2$, while the other two Fano factors do not and approach the values $\pm(2\lambda + 1)\bar{T}$, as derived in Eq. (57). The dashed lines show the effect of the linear-in-$\Delta T$ corrections of Eq. (57), which must be included to better estimate the Fano factors, even for small $\Delta T$. Finally, we also see that the symmetry $\mathcal{F}_{33}^J \leftrightarrow -\mathcal{F}_{44}^J$ upon exchange of $T_1 \leftrightarrow T_2$ is valid for generic values of $T_1/T_2$ and not only in the large bias regime as discussed previously. This property can be proven explicitly by manipulating the integral expressions for $J_T$, $S_{33}^{JJ}$, and $S_{44}^{JJ}$ (Eqs. (37), (41d), and (41e), respectively).

# 5 Effective single-particle picture

To gain additional insights into the properties of the delta-$T$ and heat-current noise, we find it useful to introduce an effective density of states (EDOS) [38,103,104]. We define the EDOS $D_\lambda(E)$ by the relation

$$\frac{P_\alpha(E)}{2\pi a} = D_\lambda(E, T_\alpha)f_\alpha(-E), \tag{65}$$

where $f_\alpha(E) = [\exp(E/T_\alpha) + 1]^{-1}$ is the Fermi-Dirac distribution at zero electrochemical potential $\mu_\alpha = 0$ and $P_\alpha(E)$ is the quasiparticle Green's function (19) in energy space (see Appendix F for details). Alternatively, one may interpret the product $D_\lambda(E, T_\alpha)f_\alpha(-E)$ as an effective anyon distribution, an approach recently pursued in Ref. [105]. Straightforward manipulation of $P_\alpha(E)$ gives the explicit expression

$$D_\lambda(E, T) = \frac{1}{v_F}\left(\frac{2\pi a}{v_F}\right)^{\lambda-1} T^{\lambda-1} \frac{\left|\Gamma\left(\frac{\lambda}{2} + i\frac{E}{2\pi T}\right)\right|^2}{\Gamma(\lambda)\left|\Gamma\left(\frac{1}{2} + i\frac{E}{2\pi T}\right)\right|^2}, \tag{66}$$

along with its zero-temperature limit

$$D_\lambda(E, 0) = \frac{1}{v_F \Gamma(\lambda)}\left(\frac{a}{v_F}\right)^{\lambda-1} |E|^{\lambda-1}. \tag{67}$$

For non-interacting electrons, $D_1(E, T) = 1/v_F$, which, notably, has no energy and temperature dependencies.

With the EDOS (66), we use a Fourier transform to write the charge tunneling noise $S_{TT}^{II}$ in Eq. (25c) as

$$S_{TT}^{II} = \frac{4e^2 \nu^2 |\Lambda|^2}{(2\pi a)^2} \frac{1}{2\pi} \int_{-\infty}^{+\infty} dE\, P_1(E)P_2(-E) \equiv \frac{4\nu^2 e^2}{2\pi} \int_{-\infty}^{+\infty} dE\, D_{\text{eff}}(E)f_1(-E)f_2(E). \tag{68}$$

Here, in the final equality, we defined the effective energy-dependent tunneling probability

$$D_{\text{eff}}(E) \equiv |\Lambda|^2 D_\lambda(E, T_1)D_\lambda(-E, T_2), \tag{69}$$

which reduces to $D_{\text{eff}}(E) = |\Lambda|^2/v_F^2 = D$ [see Eq. (24)] for $\lambda = \nu = 1$. In this case, the expression (68) is fully equivalent to the scattering formula in Eq. (G.9) (see Appendix G), for weak tunneling. By inspecting Eqs. (66) and (69), we see that both $D_\lambda(E, T_\alpha)$ and $D_{\text{eff}}(E)$ are even functions of energy. This feature is a consequence of the particle-hole symmetry inherent to the linearized bosonic spectrum, which is a key feature of the chiral Luttinger model. By using this symmetry, we further express the tunneling charge noise (68) as

$$S_{TT}^{II} = 2(e\nu)^2(\Gamma_{1\to 2} + \Gamma_{2\to 1}), \tag{70a}$$

$$\Gamma_{1\to 2} \equiv \frac{1}{2\pi} \int_{-\infty}^{+\infty} dE\, D_{\text{eff}}(E)f_1(E)[1 - f_2(E)], \tag{70b}$$

$$\Gamma_{2\to 1} \equiv \frac{1}{2\pi} \int_{-\infty}^{+\infty} dE\, D_{\text{eff}}(E)f_2(E)[1 - f_1(E)]. \tag{70c}$$

Here, $\Gamma_{1\leftrightarrow 2}$ are tunneling rates, in terms of which the charge tunneling current (18) reads $I_T = -2e\nu(\Gamma_{1\to 2} - \Gamma_{2\to 1})$. The rewriting of the tunneling noise in the form of Eq. (70) makes the result analogous to a conventional Landauer-Büttiker formula, where the EDOS plays the role of the transmission function.

The expressions for the tunneling current and the associated noise in terms of rates are a special instance of a general behavior of weak tunneling links [106]. An advantage of writing the tunneling noise in this way is that it permits a transparent interpretation of the large temperature bias regime discussed in Sec. 3.3. Indeed, setting $T_2 = 0$, the rates $\Gamma_{1\to2}$ and $\Gamma_{2\to1}$ select only negative and positive energies, respectively. For free-electron tunneling, this limit permits a clear interpretation of the non-interacting tunneling noise (34) as being proportional to the sum of electron and hole fluxes emanating from the hot source contact [95]. By analogy, the strongly correlated expression (32) can, via Eq. (70), viewed as a sum of fluxes of fractionally charged quasi-particles and quasi-holes, mediated by the effective tunneling probability $D_{\text{eff}}(E)$ in Eq. (69).

Analogously to the delta-$T$ noise, we can also express the heat-current noise by exploiting the EDOS. In particular, the heat tunneling noise (41c) can be written as

$$S_{TT}^{JJ} = \frac{4}{2\pi} \int_{-\infty}^{+\infty} dE\, E^2 D_{\text{eff}}(E) f_1(-E) f_2(E), \tag{71}$$

which reduces to the scattering formula Eq. (G.9) when $\lambda = \nu = 1$ (see Appendix G). However, in contrast to the charge noise, it is not possible to introduce rates in such a way that the tunneling current is given by their difference and the noise by their sum. The reason for this is that the transported heat depends on the energy at which it is transferred. As a consequence, the rates for the heat transfers includes integration over $E D_{\text{eff}}(E)$, while the noise instead includes integration over $E^2 D_{\text{eff}}(E)$. For non-interacting systems, this fact was recently noted in Ref. [95], and we thus establish here the same property also for weak tunneling in the FQH regime.

The above approach shows that by introducing $D_{\text{eff}}(E)$, we can put our perturbative approach to weak tunneling in the FQH regime on a similar footing with non-interacting particles treated with a scattering approach. As such, insofar as the tunneling currents and the associated noise are concerned, we may view the FQH setup in Fig. 1 as two fermionic reservoirs (the sources) bridged by a conductor fully captured in terms of the energy-dependent transmission $D_{\text{eff}}(E)$. With the EDOS and the effective tunneling probability, we see that the non-trivial scaling dimension behavior of the tunneling delta-$T$ and heat-current noises, $S_{TT}^{II}$ and $S_{TT}^{JJ}$, respectively, comes entirely from the correlation-induced energy and temperature dependence in $D_{\text{eff}}(E)$. Furthermore, the peculiar feature of negative excess charge noise can with the EDOS be seen to be essentially the same energy filtering mechanism that was identified in scattering theory in Ref. [107] (see also Ref. [38] for a discussion).

## 6 Mixed noise

While our focus in this work is on delta-$T$ and heat-current noise —corresponding to Eq. (1), with both involved operators referring to either charge, or heat current— we may consider also correlations between a charge current operator and a heat current operator. Such quantities are known as mixed noise (see e.g. Ref. [58]). Explicitly, the mixed charge-heat noise is defined as

$$S_{\alpha\beta}^{IJ}(\omega) = \int_{-\infty}^{+\infty} dt\, \left\langle \{\delta \hat{I}_\alpha(t), \delta \hat{J}_\beta(0)\} \right\rangle e^{i\omega t}, \tag{72}$$

with $\alpha, \beta$ labeling the drain contacts 3 and 4.

In this section, we comment briefly on this type of noise for the QPC device in Fig. 1. Before presenting our results in the FQH regime, we recall previously known results, based on scattering theory, for non-interacting systems. In this case, it was shown in Ref. [58] that, near

equilibrium, the zero-frequency mixed noise is closely related to thermoelectric conversion. More specifically, at equilibrium temperature $\bar{T}$, one finds for a non-interacting electron system

$$S_0^{IJ}(0) = 2k_B \bar{T}^2 \mathcal{S} g_T(\bar{T}), \tag{73}$$

where $g_T(\bar{T})$ is the charge tunneling conductance and $\mathcal{S}$ is the Seebeck coefficient. It is well-known that finite thermoelectric conversion (i.e., $\mathcal{S} \neq 0$) always requires some sort of energy filtering mechanism (via an energy-dependent transmission) of the transferred particles and holes, i.e., a mechanism that breaks particle-hole symmetry, see e.g., Ref. [108]. This feature suggests that, also in the FQH regime, particle-hole symmetry breaking is required to generate non-vanishing mixed noise. In the following, we show that this is indeed the case. When we evaluate the mixed noise, we exclude band curvature effects, or an asymmetric tunneling amplitude $\Lambda(E) \neq \Lambda(-E)$. Instead, we focus on the simple option of breaking particle-hole symmetry with a finite voltage bias $V \neq 0$ on top of the temperature bias.

With the same approach we used for the charge and heat noises, we compute (details are provided in Appendix C) all possible combinations $S_{\alpha\beta}^{IJ}$, with $\alpha, \beta = 3, 4$. At zero frequency, we have

$$S_{33}^{IJ}(0) = +M_{TT} - \frac{V}{2}S_{TT}^{II} - 2T_1(1+\lambda)I_T + 4T_1 \partial_V \langle \hat{J}_3^{(2)} \rangle, \tag{74a}$$

$$S_{44}^{IJ}(0) = +M_{TT} + \frac{V}{2}S_{TT}^{II} + 2T_2(1+\lambda)I_T - 4T_2 \partial_V \langle \hat{J}_4^{(2)} \rangle, \tag{74b}$$

$$S_{34}^{IJ}(0) = -M_{TT} - \frac{V}{2}S_{TT}^{II} - 2\lambda T_2 I_T + 2(T_1 + T_2) \partial_V \langle \hat{J}_4^{(2)} \rangle, \tag{74c}$$

$$S_{43}^{IJ}(0) = -M_{TT} + \frac{V}{2}S_{TT}^{II} + 2\lambda T_1 I_T - 2(T_1 + T_2) \partial_V \langle \hat{J}_3^{(2)} \rangle. \tag{74d}$$

Here, we introduced the tunneling-induced components of the average heat currents in the drains in the presence of a finite voltage bias, denoted $\langle \hat{J}_\alpha^{(2)} \rangle$. We obtain these components from the perturbative expansion in Eq. (15b) (see also Eq. (B.7) in Appendix B) as

$$\langle \hat{J}_3^{(2)} \rangle = 2i|\Lambda|^2 \int_{-\infty}^{+\infty} d\tau \cos(e\nu V\tau) G_L(\tau) \partial_\tau G_R(\tau), \tag{75a}$$

$$\langle \hat{J}_4^{(2)} \rangle = 2i|\Lambda|^2 \int_{-\infty}^{+\infty} d\tau \cos(e\nu V\tau) G_R(\tau) \partial_\tau G_L(\tau). \tag{75b}$$

For $V = 0$, they reduce to $\langle \hat{J}_4^{(2)} \rangle = -\langle \hat{J}_3^{(2)} \rangle = J_T$, i.e., the heat tunneling current (37). In Eq. (74), we also introduced the integral

$$M_{TT} = 2e\nu|\Lambda|^2 \int_{-\infty}^{+\infty} d\tau \sin(e\nu V\tau) [G_L(\tau) \partial_\tau G_R(\tau) - G_R(\tau) \partial_\tau G_L(\tau)]. \tag{76}$$

The first two terms on each line in Eq. (74) represent contributions from correlations of the first-order correction to the charge and heat currents, namely $\hat{I}_\alpha^{(1)}$ and $\hat{J}_\beta^{(1)}$, cf. Eqs. (C.3-C.4) in Appendix C. As a consequence, these terms are of similar nature as the tunneling charge noise, as they involve correlations between the tunneling charge current and the heat transfer between the upper and lower edge (note, however, that due to lack of heat conservation at $V \neq 0$, the tunneling heat current from the upper to the lower edge is not the same as the tunneling heat current in the opposite direction, i.e., $\langle \hat{J}_3^{(1)} \rangle \neq -\langle \hat{J}_4^{(1)} \rangle$).

To the best of our knowledge, the full expressions in Eq. (74) have not been previously reported, especially the terms stemming from the correlations between the tunneling currents and the unperturbed currents that flow unimpeded along the edges (these are the crossed

terms denoted by $M_{\alpha\beta}^{(02)}$ and $M_{\alpha\beta}^{(20)}$ in Appendix C). We see that all the terms involved in the mixed noises in Eq. (74) vanish when particle-hole symmetry is restored, i.e., by taking $V = 0$. This feature is in agreement with the intuitive anticipation stated at the beginning of this Section, that a finite mixed noise requires the breaking of particle-hole symmetry.

Importantly, just as for for the charge and heat noises (see Sec. 5), the "tunneling" contributions, $M_{TT} \pm V S_{TT}^{II}/2$, can be written in a form that is reminiscent of a scattering-theory expression for non-interacting systems, thus providing a link to the thermoelectric response. Explicitly, defining the electrochemical potentials $\mu_{1,2}$ so that $\mu_1 - \mu_2 = e\nu V$, we find that

$$
\begin{aligned}
M_{TT} \mp \frac{V}{2} S_{TT}^{II} = 2e\nu|\Lambda|^2 \int_{-\infty}^{+\infty} \frac{dE}{2\pi} (E - \mu_{1,2}) D_\lambda(E - \mu_1, T_1) D_\lambda(E - \mu_2, T_2) \\
\times \{f_1(E - \mu_1)[1 - f_2(E - \mu_2)] + f_2(E - \mu_2)[1 - f_1(E - \mu_1)]\},
\end{aligned}
\tag{77}
$$

with $f_\alpha(E) = [1 + \exp(E/T_\alpha)]^{-1}$ and $D_\lambda(E, T_\alpha)$ given in Eq. (66). In arriving at Eq. (77), we have used that $D_\lambda(E - \mu_j, T_j) = D_\lambda(-E + \mu_j, T_j)$. Furthermore, for $\lambda = 1$, Eq. (77) matches exactly the scattering theory result (at weak and energy-independent transmission) for a two-terminal system with reservoirs at temperatures $T_{1,2}$ and chemical potentials $\mu_{1,2}$ (see, e.g., Eq. (13) in Ref. [58]). When $\lambda \neq 1$, the effect of strong correlations is fully captured by the effective density of states $D_\lambda(E, T_\alpha)$.

The above analogy with scattering theory allows us to establish the termoelectric relation (73) also for edges in the FQH effect, at least when we analyze the tunneling contributions. Indeed, we can formally show that in equilibrium, i.e., in the limit $V, \Delta T \to 0$, Eq. (77) is related to the Seebeck coefficient $\mathcal{S}$. We achieve this connection by differentiating the charge tunneling current (18) with respect to the temperature bias $\Delta T$ and evaluating the result at equilibrium, which defines the thermoelectric conductance

$$
L \equiv \left. \frac{\partial I_T}{\partial \Delta T} \right|_{\substack{V \to 0 \\ \Delta T \to 0}} = e\nu|\Lambda|^2 \int_{-\infty}^{+\infty} \frac{dE}{2\pi} D_\lambda^2(E, \bar{T}) \frac{E}{\bar{T}^2} f(E)[1 - f(E)],
\tag{78}
$$

with the global equilibrium Fermi distribution $f_1(E) = f_2(E) \equiv f(E) = [1 + \exp(E/\bar{T})]^{-1}$. It is known [108] that $L$ is related to the Seebeck coefficient $\mathcal{S}$ and the charge tunneling conductance as $L = \mathcal{S} g_T(\bar{T})$. Considering then Eq. (77) in the limit $V, \Delta T \to 0$, we find that

$$
M_{TT} \pm \frac{V}{2} S_{TT}^{II} \to S_0^{IJ}(0) = 4\bar{T}^2 L,
\tag{79}
$$

which shows that Eq. (73) holds also in the FQH regime. However, as elaborated above, we have in our model $\mathcal{S} = 0$ due to the intrinsic particle-hole symmetry. Indeed, given the symmetry $D_\lambda(E, \bar{T}) = D_\lambda(-E, \bar{T})$, the integrand in (78) is odd, so that the relation $S_0^{IJ} = \mathcal{S} = L = 0$ becomes trivial. Nonetheless, it follows that measuring a nonzero mixed noise is a clear signature of mechanisms that violate particle-hole symmetry, resulting in an asymmetric effective density of states.

Complementary to the analogy with scattering theory, we further establish another relation between the mixed noise and the thermoelectric conductance in the linear response regime, i.e., for $eV/\bar{T} \ll 1$ but finite. This connection is possible since in linear response all mixed noise terms in Eq. (74) become proportional to $eV/\bar{T}$. Likewise, also the finite-bias thermoelectric conductance $\tilde{L} = \partial_{\Delta T} I_T|_{\Delta T \to 0}$ [notice the difference compared to the definition of $L$ in (78)] becomes proportional to $eV/\bar{T}$. It follows that

$$
S_{33}^{IJ}(0) = -S_{44}^{IJ}(0) = 2\lambda \bar{T}^2 \tilde{L},
\tag{80a}
$$

$$
S_{34}^{IJ}(0) = -S_{43}^{IJ}(0) = 2(\lambda - 1)\bar{T}^2 \tilde{L},
\tag{80b}
$$

to leading order in $eV/\bar{T}$. The explicit derivation of Eq. (80) is provided in Appendix C. Taking the limit $V \to 0$ in Eq. (80) produces vanishing left- and right-hand sides, in agreement with the previous analysis at equilibrium.

Since our main focus of this paper FQH tunneling induced by a pure temperature biases (in which case the mixed noise vanishes, as discussed above), we leave a broader analysis of the mixed noise correlators, with both temperature and voltage biases present, for future studies.

## 7 Summary and outlook

With the chiral Luttinger liquid model, we computed quantum transport observables in a QPC device (see Fig. 1) in the FQH regime at Laughlin fillings $\nu = (2n+1)^{-1}$. Focusing on the more unconventional configuration with a temperature bias between the source contacts, we derived detailed expressions for charge and heat currents entering the drain contacts, their auto- and cross-correlation noises, as well as mixed charge- and heat-current correlation noise. We complemented our calculations with an interpretation of the transport in terms of an effective density of states. This interpretation highlights a key aspect of temperature-biased noise: In essence, injecting particles into the QPC region via edge states results in noise that, when the edge temperatures are different, explicitly probes the scaling dimensions dependence of the effective density of states. Our findings thereby explicitly show how the scaling dimensions of the tunneling particles enter these unconventional noise observables, including delta-$T$ noise, heat-current noise, and mixed noise. As such, our work provides novel opportunities to extract the elusive scaling dimensions of quasiparticles in the FQH effect. In turn, these scaling dimensions are paramount to identify the anyonic statistics of these quasiparticles.

What are then the advantages and disadvantages of the different types of noise? The delta-$T$ noise is the simplest temperature-biased noise to measure and has already been implemented in the FQH regime (see, e.g., Ref. [31]). As its major feature, it detects only the scaling dimensions for tunneling of charged particles, although these scaling dimensions could be indirectly affected by the presence of neutral modes. As such, for more complex edge structures, it might be difficult to use delta-$T$ noise to isolate the scaling dimensions of a certain type of anyonic quasiparticles. In comparison, the heat-current noise directly reflects the contributions from both charge and neutral modes. Its experimental access is however more demanding (see discussion below). The mixed noise is mostly-advantageous in detecting the particle-hole symmetry of the system. Indeed, in particle-hole symmetric systems, like the presently considered low-energy dynamics of the FQH edge, the mixed noise vanishes in the absence of a voltage bias. Finite mixed noise due to a pure temperature bias, thus probes not only scaling dimensions via the effective density of states, but is also a signal of breaking particle-hole symmetry, e.g., deviations from the linear edge mode spectrum. Remarkably, this connection to particle-hole symmetry relates the mixed noise to the Seebeck coefficient, a quantity that also requires the presence of particle-hole symmetry breaking. However, among the observables considered in this work, mixed noise is probably the most difficult one to access experimentally, despite its connection to thermoelectric response, as derived in Eq. (80).

Our work is further potentially applicable to other platforms that support edge states. First, it paves the way for generalized noises as a tool to identify scaling dimensions of more involved FQH edges, including hierarchical states like $\nu = 2/5$ and $\nu = 2/3$ or as a tool to distinguish candidate states for non-Abelian states, e.g., at $\nu = 5/2$ and $\nu = 12/5$. Second, our calculations can be adapted to describe temperature-biased noise in related strongly correlated one-dimensional systems, such as disordered FQH line junctions [41, 74–77], disordered quantum wires [78], and quantum spin Hall edges [79].

We end by discussing the feasibility to experimentally measure our proposed noise components. FQH setups with temperature gradients across QPCs have been realized in GaAs-based devices (see e.g, Ref. [31]) and charge currents, heat currents, and charge noise are by now routinely measured. To also measure heat-current noise, it was proposed in Refs. [55, 109] that edge-coupled quantum dots, via thermoelectricity, may convert edge channel heat-current fluctuations to more easily measurable charge-current fluctuations. Alternatively, heat-current noise and mixed noise can be converted [1] to temperature fluctuations in a floating probe contact. A similar strategy could be used to access the mixed noise by monitoring electrical potential and temperature fluctuations in the floating probe [51]. Another indirect way to measure the mixed noise would be via the Seebeck coefficient. Devices with such implementations in the FQH regime remain, to the best of our knowledge, yet to be fabricated, but we believe they might be within reach with current experimental techniques.

## Acknowledgments

We thank Jinhong Park and Giacomo Rebora for useful comments on the manuscript.

**Funding information** C.S. acknowledges support from the Area of Advance Nano at Chalmers University of Technology and from the Swedish Vetenskapsrådet via Project No. 2023-04043. G.Z. acknowledges the support from National Natural Science Foundation of China (Grant No. 12374158) and Innovation Program for Quantum Science and Technology (Grant No. 2021ZD0302400). M.A. acknowledges financial support by the National Recovery and Resilience Plan MUR project No. PE0000023-NQSTI. This project has received funding from the European Union's Horizon 2020 research and innovation programme under grant agreement No. 101031655 (TEAPOT).

## A Derivations of charge currents and delta-T noise

### A.1 Currents

As our starting point, we recall that the unperturbed operators representing the charge currents entering the drain contacts 3 and 4 are given by

$$\hat{I}_3^{(0)}(t) = \frac{e v_F \sqrt{\nu}}{2\pi} \partial_x \hat{\phi}_R(x_3, t) + \frac{e^2 \nu}{2\pi} V_1\,, \tag{A.1}$$

$$\hat{I}_4^{(0)}(t) = -\frac{e v_F \sqrt{\nu}}{2\pi} \partial_x \hat{\phi}_L(x_4, t) + \frac{e^2 \nu}{2\pi} V_2\,, \tag{A.2}$$

where $x_3$ and $x_4$ are the locations of the drains and $V_{1,2}$ are the voltages applied at the source contacts. The corrections induced by the tunneling are given in Eqs. (15), which we evaluate at leading order to

$$\hat{I}_3^{(1)}(t) = i e \nu [\Lambda e^{-i e \nu V \tilde{t}} \hat{\psi}_R^\dagger(\tilde{t}) \hat{\psi}_L(\tilde{t}) - \Lambda^* e^{i e \nu V \tilde{t}} \hat{\psi}_L^\dagger(\tilde{t}) \hat{\psi}_R(\tilde{t})]\,, \tag{A.3a}$$

$$\hat{I}_4^{(1)}(t) = -i e \nu [\Lambda e^{-i e \nu V \bar{t}} \hat{\psi}_R^\dagger(\bar{t}) \hat{\psi}_L(\bar{t}) - \Lambda^* e^{i e \nu V \bar{t}} \hat{\psi}_L^\dagger(\bar{t}) \hat{\psi}_R(\bar{t})]\,. \tag{A.3b}$$

Here, $V = V_1 - V_2$ is the voltage bias between the two edges and $\tilde{t} = t - x_3/v_F$, $\bar{t} = t + x_4/v_F$. Notice that $\hat{I}_3^{(1)}(t) = -\hat{I}_4^{(1)}(t)$ when $x_3 = -x_4$, reflecting current conservation. The expressions (A.3) are valid "downstream" of the QPC on the respective edge (i.e., for $x_3 > 0$ and $x_4 < 0$), because corrections to the unperturbed currents may only occur on these sides of the

QPC due to the chiral propagation along the edge. Due to the imbalance of Klein factors in Eq. (A.3), the first-order corrections vanish when taking the average:

$$\left\langle \hat{I}_3^{(1)}(t) \right\rangle = \left\langle \hat{I}_4^{(1)}(t) \right\rangle = 0. \tag{A.4}$$

Moving on to the second-order corrections, we find that they are given by

$$\hat{I}_3^{(2)}(t) = e\,\nu|\Lambda|^2 \int_{-\infty}^{\tilde{t}} dt'' e^{-ie\nu V(t''-\tilde{t})} [\hat{\psi}_R^\dagger(t'')\hat{\psi}_L(t''), \hat{\psi}_L^\dagger(\tilde{t})\hat{\psi}_R(\tilde{t})]$$

$$- e\,\nu|\Lambda|^2 \int_{-\infty}^{\tilde{t}} e^{ie\nu V(t''-\tilde{t})} [\hat{\psi}_L^\dagger(t'')\hat{\psi}_R(t''), \hat{\psi}_R^\dagger(\tilde{t})\hat{\psi}_L(\tilde{t})], \tag{A.5a}$$

$$\hat{I}_4^{(2)}(t) = -e\,\nu|\Lambda|^2 \int_{-\infty}^{\bar{t}} dt'' e^{-ie\nu V(t''-\bar{t})} [\hat{\psi}_R^\dagger(t'')\hat{\psi}_L(t''), \hat{\psi}_L^\dagger(\bar{t})\hat{\psi}_R(\bar{t})]$$

$$+ e\,\nu|\Lambda|^2 \int_{-\infty}^{\bar{t}} e^{ie\nu V(t''-\bar{t})} [\hat{\psi}_L^\dagger(t'')\hat{\psi}_R(t''), \hat{\psi}_R^\dagger(\bar{t})\hat{\psi}_L(\bar{t})], \tag{A.5b}$$

where we only kept the terms with balanced Klein factors. Just as for the first-order corrections, we have $\hat{I}_3^{(2)}(t) = -\hat{I}_4^{(2)}(t)$ if $x_3 = -x_4$. Taking the averages, and making the change of variable $\tau = t'' - \tilde{t}$ (for $\alpha = 3$) and $\tau = t'' - \bar{t}$ (for $\alpha = 4$), we get

$$\left\langle \hat{I}_3^{(2)}(t) \right\rangle = -2ie\,\nu|\Lambda|^2 \int_{-\infty}^{+\infty} d\tau\, \sin(e\nu V\tau) G_R(\tau) G_L(\tau) \equiv -I_T, \tag{A.6a}$$

$$\left\langle \hat{I}_4^{(2)}(t) \right\rangle = +2ie\,\nu|\Lambda|^2 \int_{-\infty}^{+\infty} d\tau\, \sin(e\nu V\tau) G_R(\tau) G_L(\tau) \equiv I_T, \tag{A.6b}$$

where we identified the charge tunneling current in Eq. (18). Note that the average currents (A.6) do not depend on time, as expected for the constant voltage bias, and the currents are equal and opposite, as required by charge current conservation. Gathering the above results, we have that the average charge currents that enter the drains are given by

$$\left\langle \hat{I}_3 \right\rangle = \frac{e^2\nu}{2\pi}V_1 - I_T, \tag{A.7}$$

$$\left\langle \hat{I}_4 \right\rangle = \frac{e^2\nu}{2\pi}V_2 + I_T. \tag{A.8}$$

## A.2 Zeroth order (or equilibrium) charge-current noise

Similarly to the charge current, we decompose the charge-current noise $S_{\alpha\beta}^{II}$ as

$$S_{\alpha\beta}^{II} = S_{\alpha\beta}^{(00)} + S_{\alpha\beta}^{(11)} + S_{\alpha\beta}^{(02)} + S_{\alpha\beta}^{(20)} + \mathcal{O}(|\Lambda|^4), \tag{A.9}$$

where

$$S_{\alpha\beta}^{(ij)}(t_1 - t_2) = \left\langle \left\{ \hat{I}_\alpha^{(i)}(t_1), \hat{I}_\beta^{(j)}(t_2) \right\} \right\rangle - 2 \left\langle \hat{I}_\alpha^{(i)}(t_1) \right\rangle \left\langle \hat{I}_\beta^{(j)}(t_2) \right\rangle. \tag{A.10}$$

Here, the two superscripts $i, j$ denote the order of the current operator expansion terms in Eq. (14), while the subscripts $\alpha, \beta$ take the values 3 or 4, describing the drain contacts. We further note that the "crossed" terms $S_{\alpha\beta}^{(02)}$ and $S_{\alpha\beta}^{(20)}$ represent cross-correlations between the unperturbed currents along the edges and the tunneling current induced by the QPC. These terms are nothing but the contributions $S_{\alpha T}^{II}$ and $S_{T\alpha}^{II}$ appearing in Eq (4).

Next, we compute the zeroth order noise terms in (A.10). We start with

$$
\begin{aligned}
S_{44}^{(00)}(t_1 - t_2) &= \frac{e^2 \nu}{(2\pi)^2} \left\langle \partial_{t_1} \hat{\phi}_L(\tilde{t}_1) \partial_{t_2} \hat{\phi}_L(\tilde{t}_2) \right\rangle + (t_1 \leftrightarrow t_2) \\
&= \frac{e^2 \nu}{(2\pi)^2} \frac{-\pi^2 T_2^2}{\sinh^2[\pi T_2(i\tau_0 - (t_1 - t_2))]} + (t_1 \leftrightarrow t_2),
\end{aligned}
\tag{A.11}
$$

where we used the expression (20) for the bosonic Green's function. Next, by Fourier transforming with respect to the time difference $\tau \equiv t_1 - t_2$, we get

$$
S_{44}^{(00)}(\omega) = \frac{e^2 \nu}{(2\pi)^2} \int_{-\infty}^{+\infty} d\tau \left[ \frac{-\pi^2 T_2^2 e^{i\omega\tau}}{\sinh^2[\pi T_2(i\tau_0 - \tau)]} + (\tau \to -\tau) \right] = \frac{e^2 \nu}{2\pi} \omega \coth\left[ \frac{\omega}{2T_2} \right]. \tag{A.12}
$$

In the zero-frequency limit, this expression reduces to the expected Johnson-Nyquist expression

$$
S_{44}^{(00)}(\omega \to 0) = 2 \frac{e^2 \nu}{2\pi} T_2. \tag{A.13}
$$

This expression coincides with $S_{22}^{II}$ in the main text, cf. Eq. (25b), as it represents the fluctuations reaching drain 2 in the absence of tunneling. The results for $S_{33}^{(00)}(\omega)$ and $S_{33}^{(00)}(0)$ are obtained from Eqs. (A.12) and (A.13), respectively, by substituting $T_2 \to T_1$, yielding Eq. (25a). Identical calculations for the cross-correlation noises lead to

$$
S_{34}^{(00)}(\omega) = S_{43}^{(00)}(\omega) = 0, \tag{A.14}
$$

since at zeroth order, the two bosonic fields $\hat{\phi}_{R/L}$ are uncorrelated.

## A.3 First order, or tunneling, charge-current noise

The first order term in the noise (A.10) reads

$$
S_{\alpha\beta}^{(11)}(t_1 - t_2) = \left\langle \left\{ \hat{I}_\alpha^{(1)}(t_1), \hat{I}_\beta^{(1)}(t_2) \right\} \right\rangle - 2 \left\langle \hat{I}_\alpha^{(1)}(t_1) \right\rangle \left\langle \hat{I}_\beta^{(1)}(t_2) \right\rangle, \tag{A.15}
$$

where we used that the first-order corrections to the average current vanish. By next using the first order corrections (A.3), we see that

$$
S_{44}^{(11)}(t_1 - t_2) = S_{33}^{(11)}(t_1 - t_2) = -S_{34}^{(11)}(t_1 - t_2) = -S_{43}^{(11)}(t_1 - t_2), \tag{A.16}
$$

so there is only one independent term. Inserting Eq. (A.3b) into Eq. (A.15) we obtain

$$
S_{44}^{(11)}(t_1 - t_2) = 2(e\nu)^2 |\Lambda|^2 \cos[e\nu V(t_1 - t_2)] G_R(t_1 - t_2) G_L(t_1 - t_2) + (t_1 \leftrightarrow t_2), \tag{A.17}
$$

and thus, after a Fourier transform, we arrive at

$$
S_{44}^{(11)}(\omega \to 0) = 4(e\nu)^2 |\Lambda|^2 \int_{-\infty}^{+\infty} d\tau \cos(e\nu V\tau) G_R(\tau) G_L(\tau) \equiv S_{TT}^{II}, \tag{A.18}
$$

which defines the tunneling current noise $S_{TT}^{II}$ in Eq. (25c).

## A.4 Crossed charge-current noise terms $S_{\alpha\beta}^{(02)} + S_{\alpha\beta}^{(20)}$

Here, we compute the remaining last terms in the noise expansion (A.10). These terms represent correlations between the unperturbed currents on the edge and the tunneling current induced by the QPC.

### A.4.1 $S_{44}^{(02)} + S_{44}^{(20)}$

We start with the contribution $S_{44}^{(02)}$. By using the previously found expressions for the current operators, Eqs. (A.2) and (A.5b), and recalling that $v_F \partial_x \hat{\phi}_L = \partial_t \hat{\phi}_L$, due to chiral propagation, we obtain [37, 87, 88]

$$
\begin{aligned}
S_{44}^{(02)}(t_{12}) = & -\frac{2i|\Lambda|^2(ev)^2}{2\pi} \int_{-\infty}^{\bar{t}_2} dt'' \cos[evV(t''-\bar{t}_2)]\Big[ G_R(t''-\bar{t}_2)G_L(t''-\bar{t}_2)\mathcal{K}(\bar{t}_1,t'',\bar{t}_2) \\
& + G_R(\bar{t}_2-t'')G_L(\bar{t}_2-t'')\mathcal{K}(\bar{t}_1,\bar{t}_2,t'') \Big] \\
& + \frac{2i(ev)^2|\Lambda|^2}{2\pi} \int_{-\infty}^{\bar{t}_2} dt'' \cos[evV(t''-\bar{t}_2)]\Big[ G_R(t''-\bar{t}_2)G_L(t''-\bar{t}_2)\mathcal{K}(-\bar{t}_1,-t'',-\bar{t}_2) \\
& + G_R(\bar{t}_1-t'')G_L(\bar{t}_2-t'')\mathcal{K}(-\bar{t}_1,-\bar{t}_2,-t'') \Big],
\end{aligned}
\tag{A.19}
$$

where we abbreviated $t_{12} = t_1 - t_2$, $\bar{t}_i = t_i + x_4/v_F$ for $i = 1, 2$, and also defined the function

$$
\mathcal{K}(t_1,t_2,t_3) = \pi T_2\{\coth[\pi T_2(i\tau_0-(t_1-t_2))] - \coth[\pi T_2(i\tau_0-(t_1-t_3))]\}.
\tag{A.20}
$$

Finally, taking advantage of the permutation identity $\mathcal{K}(1,3,2) = -\mathcal{K}(1,2,3)$ and introducing the variable $\tau = t'' - \bar{t}_2$, we arrive at

$$
\begin{aligned}
S_{44}^{(02)}(t_{12}) = & -\frac{2i(ev)^2|\Lambda|^2}{2\pi} \int_{-\infty}^{0} d\tau \cos(evV\tau)\mathcal{K}_0(t_{12},\tau)[G_R(\tau)G_L(\tau) - G_R(-\tau)G_L(-\tau)] \\
& -\frac{2i(ev)^2|\Lambda|^2}{2\pi} \int_{0}^{+\infty} d\tau \cos(evV\tau)\mathcal{K}_0(-t_{12},\tau)[G_R(\tau)G_L(\tau) - G_R(-\tau)G_L(-\tau)],
\end{aligned}
\tag{A.21}
$$

in which

$$
\mathcal{K}_0(t_{12},\tau) \equiv \mathcal{K}(\bar{t}_1,\bar{t}_2+\tau,\bar{t}_2) = \pi T_2\{\coth[\pi T_2(i\tau_0-(t_{12}-\tau))] - \coth[\pi T_2(i\tau_0-t_{12})]\}.
\tag{A.22}
$$

Equation (A.21) explicitly shows that the noise only depends on the time difference $t_{12} = t_1 - t_2$, as expected in the steady state.

The procedure to evaluate $S_{44}^{(20)}$ is identical to that for $S_{44}^{(02)}$. We find

$$
\begin{aligned}
S_{44}^{(20)}(t_{12}) = & -\frac{2i(ev)^2|\Lambda|^2}{2\pi} \int_{0}^{+\infty} d\tau \cos(evV\tau)\mathcal{K}_0(t_{12},\tau)[G_R(\tau)G_L(\tau) - G_R(-\tau)G_L(-\tau)] \\
& -\frac{2i(ev)^2|\Lambda|^2}{2\pi} \int_{-\infty}^{0} d\tau \cos(evV\tau)\mathcal{K}_0(-t_{12},\tau)[G_R(\tau)G_L(\tau) - G_R(-\tau)G_L(-\tau)].
\end{aligned}
\tag{A.23}
$$

We can therefore combine Eqs. (A.21) and (A.23) into a single integral

$$
\begin{aligned}
S_{44}^{(02+20)}(t_{12}) = & -\frac{2i(ev)^2|\Lambda|^2}{2\pi} \int_{-\infty}^{+\infty} d\tau \cos(evV\tau)G_R(\tau)G_L(\tau)[\mathcal{K}_0(t_{12},\tau) - \mathcal{K}_0(t_{12},-\tau)] \\
& -\frac{2i(ev)^2|\Lambda|^2}{2\pi} \int_{-\infty}^{+\infty} d\tau \cos(evV\tau)G_R(\tau)G_L(\tau)[\mathcal{K}_0(-t_{12},\tau) - \mathcal{K}_0(-t_{12},-\tau)],
\end{aligned}
\tag{A.24}
$$

and we obtain the finite-frequency expression by Fourier transforming with respect to the time difference $t_{12}$. The final result thus involves the function

$$
\mathcal{K}_0(\omega,\tau) = \int_{-\infty}^{+\infty} dt_{12}\, e^{i\omega t_{12}}\mathcal{K}_0(t_{12},\tau),
\tag{A.25}
$$

which can be evaluated with the residue theorem. We obtain

$$S_{44}^{(02+20)}(\omega) = -4i(e\nu)^2|\Lambda|^2 \coth\left(\frac{\omega}{2T_2}\right)\int_{-\infty}^{+\infty} d\tau \cos(e\nu V\tau)G_R(\tau)G_L(\tau)\sin(\omega\tau). \quad \text{(A.26)}$$

Taking the zero-frequency limit, we get

$$S_{44}^{(02+20)}(0) = -4T_2 \times 2i(e\nu)^2|\Lambda|^2 \int_{-\infty}^{+\infty} d\tau \cos(e\nu V\tau)\tau\, G_R(\tau)G_L(\tau) = -4T_2\frac{\partial I_T}{\partial V}, \quad \text{(A.27)}$$

where in the final equality, we identified the differential charge tunneling conductance (21).

### A.4.2 $S_{33}^{(02)} + S_{33}^{(20)}$

We evaluate these terms by following an identical procedure as in the previous subsection. The result is simply obtained by the substitutions $L \to R$ and $T_2 \to T_1$:

$$S_{33}^{(02+20)}(\omega) = -4i(e\nu)^2|\Lambda|^2 \coth\left(\frac{\omega}{2T_1}\right)\int_{-\infty}^{+\infty} d\tau \cos(e\nu V\tau)G_R(\tau)G_L(\tau)\sin(\omega\tau), \quad \text{(A.28)}$$

$$S_{33}^{(02+20)}(0) = -4T_1 \times 2i(e\nu)^2|\Lambda|^2 \int_{-\infty}^{+\infty} d\tau \cos(e\nu V\tau)\tau\, G_R(\tau)G_L(\tau) = -4T_1\frac{\partial I_T}{\partial V}. \quad \text{(A.29)}$$

### A.4.3 $S_{34}^{(02)} + S_{34}^{(20)}$ and $S_{43}^{(02)} + S_{43}^{(20)}$

The evaluation of these contributions is very similar to the calculation of the previous terms. The only difference is that we find not only the function $\mathcal{K}_0$ defined in Eq. (A.22), but also a corresponding one with $T_1$ instead of $T_2$. As a result, the final expression reads

$$S_{34}^{(02+20)}(\omega) = 2i(e\nu)^2|\Lambda|^2\left[\coth\left(\frac{\omega}{2T_1}\right) + \coth\left(\frac{\omega}{2T_2}\right)\right]\int_{-\infty}^{+\infty} d\tau \cos(e\nu V\tau)G_R(\tau)G_L(\tau)\sin(\omega\tau). \quad \text{(A.30)}$$

The zero-frequency limit is therefore

$$S_{34}^{(02+20)}(0) = 2(T_1 + T_2) \times 2i(e\nu)^2|\Lambda|^2 \int_{-\infty}^{+\infty} d\tau \cos(e\nu V\tau)\tau\, G_R(\tau)G_L(\tau) = 2(T_1 + T_2)\frac{\partial I_T}{\partial V}. \quad \text{(A.31)}$$

## A.5 Summary of charge current fluctuations

Gathering the results from all above subsections in Appendix A, we have that the tunneling current, tunneling conductance, and the associated noise to leading order in the tunneling amplitude $\Lambda$ are given by

$$I_T = 2ie\nu|\Lambda|^2 \int_{-\infty}^{+\infty} d\tau \sin(e\nu V\tau)G_R(\tau)G_L(\tau), \quad \text{(A.32a)}$$

$$\frac{\partial I_T}{\partial V} = 2i(e\nu)^2|\Lambda|^2 \int_{-\infty}^{+\infty} d\tau\, \tau \cos(e\nu V\tau)G_R(\tau)G_L(\tau), \quad \text{(A.32b)}$$

$$S_{TT}^{II} = 4(e\nu)^2|\Lambda|^2 \int_{-\infty}^{+\infty} d\tau \cos(e\nu V\tau)G_R(\tau)G_L(\tau). \quad \text{(A.32c)}$$

These expressions are stated in Eqs. (18), (21), and Eq. (25c) in the main text. The expressions for the auto- and cross-correlated charge-current noises at zero frequency, $S_{\alpha\beta}^{II}(0)$, are summarized in Tab. 1. It can readily be checked that these noise components obey the conservation law (6).

Table 1: Auto- and cross-correlation charge-current noise at zero frequency $S^{II}_{\alpha\beta}$ with the drain reservoir indices $\alpha, \beta = 3, 4$ (see Fig. 1). All expressions are given to $\mathcal{O}(|\Lambda|^2)$ in the tunneling amplitude $\Lambda$.

| $S^{II}_{\alpha\beta}(0)$ | 3 | 4 |
|---|---|---|
| 3 | $2\dfrac{e^2 \nu}{h} k_B T_1 + S^{II}_{TT} - 4k_B T_1 \dfrac{\partial I_T}{\partial V}$ | $2k_B(T_1 + T_2)\dfrac{\partial I_T}{\partial V} - S^{II}_{TT}$ |
| 4 | $2k_B(T_1 + T_2)\dfrac{\partial I_T}{\partial V} - S^{II}_{TT}$ | $2\dfrac{e^2 \nu}{h} k_B T_2 + S^{II}_{TT} - 4k_B T_2 \dfrac{\partial I_T}{\partial V}$ |

# B  Derivations of heat currents and heat-current noise

## B.1  Currents

The unperturbed operators representing the heat currents entering the drain contacts 3 and 4 are given by

$$\hat{J}^{(0)}_3(t) = \frac{v_F^2}{4\pi}[\partial_x \hat{\phi}_R(x_3, t)]^2 - \frac{q^2 \nu}{4\pi} V_1^2, \tag{B.1a}$$

$$\hat{J}^{(0)}_4(t) = \frac{v_F^2}{4\pi}[\partial_x \hat{\phi}_L(x_4, t)]^2 - \frac{q^2 \nu}{4\pi} V_2^2. \tag{B.1b}$$

The corresponding average values are readily obtained as

$$\left\langle \hat{J}^{(0)}_3(t) \right\rangle = \frac{\pi T_1^2}{12} - \frac{q^2 \nu}{4\pi} V_1^2, \tag{B.2a}$$

$$\left\langle \hat{J}^{(0)}_4(t) \right\rangle = \frac{\pi T_2^2}{12} - \frac{q^2 \nu}{4\pi} V_2^2. \tag{B.2b}$$

Here, we identified the free boson stress energy tensor $\hat{\mathcal{T}}_{R,L}(t) = [\partial_x \hat{\phi}_{R,L}(x_{3,4}, t)]^2/2$, and used that $\langle \hat{\mathcal{T}}_{R,L}(t) \rangle = \pi^2 T_{1,2}^2/(6v_F^2)$ at finite temperature [60, 110]. We find the corrections to the unperturbed current operators by evaluating the commutators in Eq. (15). At first order, we find

$$\hat{J}^{(1)}_3(t) = -\left\{ \Lambda e^{-ie\nu V \tilde{t}} \left[\partial_t \hat{\psi}^\dagger_R(\tilde{t})\right] \hat{\psi}_L(\tilde{t}) + \Lambda^* e^{ie\nu V \tilde{t}} \hat{\psi}^\dagger_L(\tilde{t}) \left[\partial_t \hat{\psi}_R(\tilde{t})\right] \right\}, \tag{B.3a}$$

$$\hat{J}^{(1)}_4(t) = -\left\{ \Lambda e^{-ie\nu V \bar{t}} \hat{\psi}^\dagger_R(\bar{t}) \left[\partial_t \hat{\psi}_L(\bar{t})\right] + \Lambda^* e^{ie\nu V \bar{t}} \left[\partial_t \hat{\psi}^\dagger_L(\bar{t})\right] \hat{\psi}_R(\bar{t}) \right\}, \tag{B.3b}$$

where $\tilde{t} = t - x_3/v_F$ and $\bar{t} = t + x_4/v_F$. Similarly to the charge transport, the expressions in Eq. (B.3) are finite only "downstream" of the QPC on the respective edge (i.e., for $x_3 > 0$ and $x_4 < 0$), because corrections to the unperturbed currents may only occur on these sides of the QPC due to the chiral propagation. Due to the imbalance of Klein factors, the first-order corrections vanish on average:

$$\left\langle \hat{J}^{(1)}_3(t) \right\rangle = \left\langle \hat{J}^{(1)}_4(t) \right\rangle = 0. \tag{B.4}$$

We find that the second-order corrections become

$$\hat{J}_3^{(2)}(t) = -i|\Lambda|^2 \int_{-\infty}^{\bar{t}} dt'' \left\{ e^{-ie\nu V(t''-\bar{t})} \left[ \hat{\psi}_R^\dagger(t'')\hat{\psi}_L(t''), \hat{\psi}_L^\dagger(\tilde{t})\partial_t\hat{\psi}_R(\tilde{t}) \right] \right.$$
$$\left. + e^{ie\nu V(t''-\bar{t})} \left[ \hat{\psi}_L^\dagger(t'')\hat{\psi}_R(t''), \partial_t\hat{\psi}_R^\dagger(\tilde{t})\hat{\psi}_L(\tilde{t}) \right] \right\}, \tag{B.5a}$$

$$\hat{J}_4^{(2)}(t) = -i|\Lambda|^2 \int_{-\infty}^{\bar{t}} dt'' \left\{ e^{-ie\nu V(t''-\bar{t})} \left[ \hat{\psi}_R^\dagger(t'')\hat{\psi}_L(t''), \partial_t\hat{\psi}_L^\dagger(\bar{t})\hat{\psi}_R(\bar{t}) \right] \right.$$
$$\left. + e^{ie\nu V(t''-\bar{t})} \left[ \hat{\psi}_L^\dagger(t'')\hat{\psi}_R(t''), \hat{\psi}_R^\dagger(\bar{t})\partial_t\hat{\psi}_L(\bar{t}) \right] \right\}, \tag{B.5b}$$

where we kept only the terms with balanced Klein factors. Evaluating the averages, we find

$$\left\langle \hat{J}_3^{(2)} \right\rangle = 2i|\Lambda|^2 \int_{-\infty}^{+\infty} d\tau \cos(e\nu V\tau) G_L(\tau)\partial_\tau G_R(\tau), \tag{B.6a}$$

$$\left\langle \hat{J}_4^{(2)} \right\rangle = 2i|\Lambda|^2 \int_{-\infty}^{+\infty} d\tau \cos(e\nu V\tau) G_R(\tau)\partial_\tau G_L(\tau). \tag{B.6b}$$

These results can be also expressed in the following equivalent form:

$$\left\langle \hat{J}_3^{(2)} \right\rangle = -i|\Lambda|^2 \int_{-\infty}^{+\infty} d\tau \cos(e\nu V\tau)[G_R(\tau)\partial_\tau G_L(\tau) - G_L(\tau)\partial_\tau G_R(\tau)] + \frac{V}{2}I_T, \tag{B.7a}$$

$$\left\langle \hat{J}_4^{(2)} \right\rangle = +i|\Lambda|^2 \int_{-\infty}^{+\infty} d\tau \cos(e\nu V\tau)[G_R(\tau)\partial_\tau G_L(\tau) - G_L(\tau)\partial_\tau G_R(\tau)] + \frac{V}{2}I_T, \tag{B.7b}$$

with $V = V_1 - V_2$. Differently from the charge currents, these expressions are not equal and opposite, as the edge heat current is not conserved for $V \neq 0$. The terms $VI_T/2$ in Eq. (B.7) are Joule heating contributions. When there is no bias between the edges, $V = 0$, the heat current coincides with the energy current and is then conserved. Then, Eq. (B.7) reduces to

$$\left\langle \hat{J}_4^{(2)} \right\rangle = -\left\langle \hat{J}_3^{(2)} \right\rangle = 2i|\Lambda|^2 \int_{-\infty}^{+\infty} d\tau \, G_R(\tau)\partial_\tau G_L(\tau) \equiv J_T, \tag{B.8}$$

which is indeed the heat tunneling current at zero voltage bias, as defined in Eq. (37).

## B.2 Zeroth order, or equilibrium, heat-current noise

We use the following notation to indicate the decomposition of the heat noise:

$$S_{\alpha\beta}^{JJ} = \Sigma_{\alpha\beta}^{(00)} + \Sigma_{\alpha\beta}^{(11)} + \Sigma_{\alpha\beta}^{(02)} + \Sigma_{\alpha\beta}^{(20)}, \tag{B.9}$$

with

$$\Sigma_{\alpha\beta}^{(ij)}(t_1 - t_2) = \left\langle \left\{ \hat{J}_\alpha^{(i)}(t_1), \hat{J}_\alpha^{(j)}(t_2) \right\} \right\rangle - 2\left\langle \hat{J}_\alpha^{(i)}(t_1) \right\rangle \left\langle \hat{J}_\alpha^{(j)}(t_2) \right\rangle. \tag{B.10}$$

We start with the evaluation of the equilibrium noise $\Sigma_{\alpha\beta}^{(00)}$, beginning with $\alpha = \beta = 4$. From the definition (B.10), we have

$$\Sigma_{44}^{(00)}(t_1 - t_2) = \left\langle \hat{J}_4^{(0)}(t_1)\hat{J}_4^{(0)}(t_2) \right\rangle - \langle \hat{J}_4^{(0)}(t_1) \rangle \langle \hat{J}_4^{(0)}(t_2) \rangle + (t_1 \leftrightarrow t_2)$$
$$= \frac{2v_F^4}{(4\pi)^2} \left( \langle \partial_x\hat{\phi}_L(x_0, t_1)\partial_x\hat{\phi}_L(x_0, t_2) \rangle \right)^2 + (t_1 \leftrightarrow t_2) = \frac{2v_F^4}{(4\pi)^2} \left( \lim_{y \to x} \partial_x\partial_y \mathcal{G}_L(x - y, \tau) \right)^2$$
$$+ (\tau \to -\tau) = \frac{2v_F^4}{(4\pi)^2} \frac{\pi^4 T_2^4}{v_F^4 \left( \sinh(\pi T_{1/2}(i\tau_0 - \tau)) \right)^4} + (\tau \to -\tau). \tag{B.11}$$

Here, in the second equality, we used the heat current operator definition (13b) together with Wick's theorem. In the third equality, we used the definition of the boson Green's function (20) and abbreviated $\tau = t_1 - t_2$. We evaluate the Fourier transform with the residue theorem as in previous sections and find

$$\Sigma_{44}^{(00)}(\omega) = \int_{-\infty}^{+\infty} d\tau e^{i\omega\tau} \Sigma_{44}^{(00)}(\tau) = \frac{\omega}{24\pi}\left((2\pi T_2)^2 + \omega^2\right)\coth\left[\frac{\omega}{2T_2}\right], \tag{B.12}$$

which in the zero-frequency limit reduces to

$$\Sigma_{44}^{(00)}(\omega \to 0) = \frac{\pi T_2^3}{3} = 4\langle \hat{J}_4^{(0)}(t)\rangle T_2, \tag{B.13}$$

upon using Eq. (B.2) for $V = 0$. Equation (B.13) is the equilibrium contribution that we denoted $S_{22}^{JJ}$ in the main text. Equations (B.12)-(B.13) manifest the equilibrium fluctuation-dissipation relation for heat transport [48].

The remaining non-vanishing contribution to the equilibrium noise, i.e., $\Sigma_{33}^{(00)}(\omega)$, is obtained by substituting $T_2 \to T_1$ in Eqs. (B.12)-(B.13). The zero-frequency component thus reads

$$\Sigma_{33}^{(00)}(0) = \frac{\pi T_1^3}{3} = 4\langle \hat{J}_3^{(0)}(t)\rangle T_1, \tag{B.14}$$

which gives $S_{11}^{JJ}$ in the main text. We also have trivially from Eq. (B.10) that

$$\Sigma_{34}^{(00)}(\omega) = \Sigma_{43}^{(00)}(\omega) = 0, \tag{B.15}$$

since the bosonic fields $\hat{\phi}_{R/L}$ are independent at zeroth order.

## B.3 First order or tunneling, heat-current noise

We now consider the heat-current noise for vanishing bias voltage $V_1 = V_2 = 0$. Using the heat current Eq. (B.3b), we obtain

$$\begin{aligned}
\Sigma_{44}^{(11)}(t_{12}) &= \langle\{\hat{J}_4^{(1)}(t_1), \hat{J}_4^{(1)}(t_2)\}\rangle - 2\langle\hat{J}_4^{(1)}(t_1)\rangle\langle\hat{J}_4^{(1)}(t_2)\rangle \\
&= |\Lambda|^2 \langle\hat{\psi}_R^\dagger(\bar{t}_1)\hat{\psi}_R(\bar{t}_2)\rangle \partial_{t_1}\partial_{t_2}\langle\hat{\psi}_L(\bar{t}_1)\hat{\psi}_L^\dagger(\bar{t}_2)\rangle + (t_1 \leftrightarrow t_2) \\
&\quad + |\Lambda|^2 \langle\hat{\psi}_R(\bar{t}_1)\hat{\psi}_R^\dagger(\bar{t}_2)\rangle \partial_{t_1}\partial_{t_2}\langle\hat{\psi}_L^\dagger(\bar{t}_1)\hat{\psi}_L(\bar{t}_2)\rangle + (t_1 \leftrightarrow t_2) \\
&= 2|\Lambda|^2[G_R(\bar{t}_1 - \bar{t}_2)\partial_{t_1}\partial_{t_2}G_L(\bar{t}_1 - \bar{t}_2) + G_R(\bar{t}_2 - \bar{t}_1)\partial_{t_1}\partial_{t_2}G_L(\bar{t}_2 - \bar{t}_1)]. \tag{B.16}
\end{aligned}$$

By performing a Fourier transform, we find

$$\begin{aligned}
\Sigma_{44}^{(11)}(\omega) &= 2|\Lambda|^2 \int_{-\infty}^{+\infty} dt_{12} e^{i\omega t_{12}}[G_R(t_{12})\partial_{t_1}\partial_{t_2}G_L(t_{12}) + G_R(-t_{12})\partial_{t_1}\partial_{t_2}G_L(-t_{12})] \\
&= -4|\Lambda|^2 \int_{-\infty}^{+\infty} d\tau \cos(\omega\tau)G_R(\tau)\partial_\tau^2 G_L(\tau). \tag{B.17}
\end{aligned}$$

In the zero-frequency limit, we thus obtain

$$\Sigma_{44}^{(11)}(0) = -4|\Lambda|^2 \int_{-\infty}^{+\infty} d\tau G_R(\tau)\partial_\tau^2 G_L(\tau) = 4|\Lambda|^2 \int_{-\infty}^{+\infty} d\tau \partial_\tau G_R(\tau)\partial_\tau G_L(\tau) \equiv S_{TT}^{JJ}, \tag{B.18}$$

which defines the tunneling heat-current noise $S_{TT}^{JJ}$ in Eq. (41c). With similar calculations, we also find $\Sigma_{33}^{(11)} = -\Sigma_{34}^{(11)} = -\Sigma_{43}^{(11)} = S_{TT}^{JJ}$. Similar calculations for the finite bias case $V \neq 0$, give

$$\Sigma_{33}^{(11)} = -4|\Lambda|^2 \int_{-\infty}^{+\infty} d\tau \cos(\omega\tau) \cos(e\nu V\tau) G_L(\tau) \partial_\tau^2 G_R(\tau), \tag{B.19a}$$

$$\Sigma_{44}^{(11)} = -4|\Lambda|^2 \int_{-\infty}^{+\infty} d\tau \cos(\omega\tau) \cos(e\nu V\tau) G_R(\tau) \partial_\tau^2 G_L(\tau), \tag{B.19b}$$

$$\Sigma_{34}^{(11)} = -4|\Lambda|^2 \int_{-\infty}^{+\infty} d\tau \cos(\omega\tau) \cos(e\nu V\tau) \partial_\tau G_L(\tau) \partial_\tau G_R(\tau) = \Sigma_{43}^{(11)}. \tag{B.19c}$$

By summing all the contributions, we find

$$\Sigma_{33}^{(11)} + \Sigma_{44}^{(11)} + \Sigma_{34}^{(11)} + \Sigma_{43}^{(11)} = V^2 S_{TT}^{II}, \tag{B.20}$$

which corresponds to the conservation of power fluctuations for the tunneling current (i.e., the equality of thermal power fluctuations and electrical power fluctuations).

## B.4  Crossed heat-current noise terms $\Sigma_{\alpha\beta}^{(02)} + \Sigma_{\alpha\beta}^{(20)}$

### B.4.1  $\Sigma_{44}^{(02)} + \Sigma_{44}^{(20)}$

We start with the contribution

$$\Sigma_{44}^{(02)}(t_1 - t_2) = \left\langle \delta\hat{J}_4^{(0)}(t_1) \delta\hat{J}_4^{(2)}(t_2) \right\rangle + \left\langle \delta\hat{J}_4^{(2)}(t_2) \delta\hat{J}_4^{(0)}(t_1) \right\rangle. \tag{B.21}$$

Considering the term $\langle \hat{J}_4^{(0)}(t_1) \hat{J}_4^{(2)}(t_2) \rangle$, we have

$$\begin{aligned}
\langle \hat{J}_4^{(0)}(t_1) \hat{J}_4^{(2)}(t_2) \rangle = -\frac{i|\Lambda|^2}{4\pi} \int_{-\infty}^{\bar{t}_2} dt'' \Big[ & \left\langle (\partial_{t_1}\hat{\phi}_L(\bar{t}_1))^2 \hat{\psi}_R^\dagger(t'') \hat{\psi}_L(t'') \partial_{t_2}\hat{\psi}_L^\dagger(\bar{t}_2) \hat{\psi}_R(\bar{t}_2) \right\rangle \\
& - \left\langle (\partial_{t_1}\hat{\phi}_L(\bar{t}_1))^2 \partial_{t_2}\hat{\psi}_L^\dagger(\bar{t}_2) \hat{\psi}_R(\bar{t}_2) \hat{\psi}_R^\dagger(t'') \hat{\psi}_L(t'') \right\rangle \\
& + \left\langle (\partial_{t_1}\hat{\phi}_L(\bar{t}_1))^2 \hat{\psi}_L^\dagger(t'') \hat{\psi}_R(t'') \hat{\psi}_R^\dagger(\bar{t}_2) \partial_{t_2}\hat{\psi}_L(\bar{t}_2) \right\rangle \\
& - \left\langle (\partial_{t_1}\hat{\phi}_L(\bar{t}_1))^2 \hat{\psi}_R^\dagger(\bar{t}_2) \partial_{t_2}\hat{\psi}_L(\bar{t}_2) \hat{\psi}_L^\dagger(t'') \hat{\psi}_R(t'') \right\rangle \Big].
\end{aligned} \tag{B.22}$$

By performing the averages, and subtracting the product of the currents, we obtain

$$\begin{aligned}
\langle \delta\hat{J}_4^{(0)}(t_1) \delta\hat{J}_4^{(2)}(t_2) \rangle = \frac{i\lambda|\Lambda|^2}{2\pi} & \underbrace{\int_{-\infty}^{\bar{t}_2} dt'' G_R(t'' - \bar{t}_2) \partial_{t_2}[K(\bar{t}_1, t'', \bar{t}_2) G_L(t'' - \bar{t}_2)]}_{\mathcal{J}_1} \\
& - \frac{i\lambda|\Lambda|^2}{2\pi} \underbrace{\int_{-\infty}^{\bar{t}_2} dt'' G_R(\bar{t}_2 - t'') \partial_{t_2}[K(\bar{t}_1, \bar{t}_2, t'') G_L(\bar{t}_2 - t'')]}_{\mathcal{J}_2},
\end{aligned} \tag{B.23}$$

with the function

$$K(\tau_1, \tau_3, \tau_4) = \frac{\pi^2 T_2^2 \sinh^2[\pi T_2(\tau_3 - \tau_4)]}{\sinh^2[\pi T_2(i\tau_0 - (\tau_1 - \tau_3))] \sinh^2[\pi T_2(i\tau_0 - (\tau_1 - \tau_4))]} = K(\tau_1, \tau_4, \tau_3). \tag{B.24}$$

By making a change of variable $t'' - \bar{t}_2 = \tau$ and expanding the derivatives, the integrals $\mathcal{J}_{1,2}$ in (B.23) become

$$\mathcal{J}_1(t_{12}) = \int_{-\infty}^0 d\tau\, G_R(\tau)\left[h(t_{12}, \tau)G_L(\tau) - K_0(t_{12}, \tau)\partial_\tau G_L(\tau)\right], \tag{B.25}$$

$$\mathcal{J}_2(t_{12}) = \int_{-\infty}^0 d\tau\, G_R(-\tau)\left[h(t_{12}, \tau)G_L(-\tau) - K_0(t_{12}, \tau)\partial_\tau G_L(-\tau)\right], \tag{B.26}$$

where

$$K_0(t_{12}, \tau) = \frac{\pi^2 T_2^2 \sinh^2(\pi T_2 \tau)}{\sinh^2[\pi T_2(i\tau_0 - t_{12})]\sinh^2[\pi T_2(i\tau_0 - (t_{12} - \tau))]}, \tag{B.27}$$

$$h(t_{12}, \tau) = -2\pi^2 T_2^2 \frac{\pi T_2 \coth[\pi T_2(i\tau_0 - t_{12})] - \pi T_2 \coth[\pi T_2(i\tau_0 - (t_{12} - \tau))]}{\sinh^2[\pi T_2(i\tau_0 - t_{12})]}. \tag{B.28}$$

The other term of interest, $\langle \hat{J}_4^{(2)}(t_2)\hat{J}_4^{(0)}(t_1)\rangle$, can be handled in a similar way. We find:

$$\langle \hat{J}_4^{(2)}(t_2)\hat{J}_4^{(0)}(t_1)\rangle - \langle \hat{J}_4^{(2)}(t_2)\rangle\langle \hat{J}_4^{(0)}(t_1)\rangle = \frac{i\lambda|\Lambda|^2}{2\pi}\left[\mathcal{J}_3(t_{12}) - \mathcal{J}_4(t_{12})\right], \tag{B.29}$$

with

$$\mathcal{J}_3(t_{12}) = \int_{-\infty}^0 d\tau\, G_R(\tau)\left[-h(-t_{12}, -\tau)G_L(\tau) - K_0(-t_{12}, -\tau)\partial_\tau G_L(\tau)\right], \tag{B.30}$$

$$\mathcal{J}_4(t_{12}) = \int_{-\infty}^0 d\tau\, G_R(-\tau)\left[-h(-t_{12}, -\tau)G_L(-\tau) - K_0(-t_{12}, -\tau)\partial_\tau G_L(-\tau)\right]. \tag{B.31}$$

Performing an analogous calculation for $\Sigma_{44}^{(20)}$, and taking a Fourier transform, we obtain

$$\begin{aligned}\Sigma_{44}^{(02)}(\omega) + \Sigma_{44}^{(20)}(\omega) = \frac{i\lambda|\Lambda|^2}{2\pi}\big[&\tilde{\mathcal{J}}_1(\omega) - \tilde{\mathcal{J}}_2(\omega) + \tilde{\mathcal{J}}_3(\omega) - \tilde{\mathcal{J}}_4(\omega)\\ &+ \tilde{\mathcal{J}}_1(-\omega) - \tilde{\mathcal{J}}_2(-\omega) + \tilde{\mathcal{J}}_3(-\omega) - \tilde{\mathcal{J}}_4(-\omega)\big],\end{aligned} \tag{B.32}$$

where

$$\tilde{\mathcal{J}}_\alpha(\omega) = \int_{-\infty}^{+\infty} dt_{12}\, \mathcal{J}_\alpha(t_{12})e^{i\omega t_{12}}. \tag{B.33}$$

It is clear from the expressions of the integrals $\mathcal{J}_\alpha(t_{12})$ that we need the Fourier transforms $\tilde{K}_0(\omega, \tau)$ and $\tilde{h}(\omega, \tau)$. The former is readily found by using the residue theorem and reads

$$\tilde{K}_0(\omega, \tau) = \pi i\left[1 + \coth\left(\frac{\omega}{2T_2}\right)\right]\left[i\omega\left(1 + e^{i\omega\tau}\right) + 2\pi T_2 \coth(\pi T_2 \tau)\left(1 - e^{i\omega\tau}\right)\right]. \tag{B.34}$$

For the latter, we use the following manipulation

$$\tilde{h}(\omega, \tau) \equiv \int_{-\infty}^{+\infty} dt_{12}\, h(t_{12}, \tau)e^{i\omega t_{12}} = e^{i\omega\tau}\int_{-\infty}^{+\infty} dt_{12}\, e^{i\omega t_{12}}h(t_{12} + \tau, \tau). \tag{B.35}$$

The reason for this is that

$$\begin{aligned}h(t_{12} + \tau, \tau) &= 2\pi^2 T_2^2 \frac{\pi T_2 \coth[\pi T_2(i\tau_0 - t_{12})] - \pi T_2 \coth[\pi T_2(i\tau_0 - (t_{12} + \tau))]}{\sinh^2[\pi T_2(i\tau_0 - (t_{12} + \tau))]}\\ &= \left(\partial_y K_0(t_{12}, y)\right)_{y=-\tau} = -\partial_\tau K_0(t_{12}, -\tau),\end{aligned} \tag{B.36}$$

Therefore,

$$\tilde{h}(\omega,\tau)=-e^{i\omega\tau}\partial_\tau\int_{-\infty}^{+\infty}dt_{12}\,e^{i\omega t_{12}}K_0(t_{12},-\tau)=-e^{i\omega\tau}\partial_\tau\widetilde{K}_0(\omega,-\tau)=e^{i\omega\tau}\left(\partial_y\widetilde{K}_0(\omega,y)\right)_{y=-\tau},\quad\text{(B.37)}$$

which allows us to obtain $\tilde{h}(\omega,\tau)$ from (B.34), yielding

$$\tilde{h}(\omega,\tau)=i\pi\left[\coth\left(\frac{\omega}{2T_2}\right)+1\right]\left[\pi T_2\frac{2\pi T_2\left(1-e^{i\tau\omega}\right)+i\omega\sinh(2\pi\tau T_2)}{\sinh^2(\pi\tau T_2)}-\omega^2\right].\quad\text{(B.38)}$$

By combining all integrals in Eq. (B.32), we arrive at the expression

$$\Sigma_{44}^{(02)}(\omega)+\Sigma_{44}^{(20)}(\omega)=\frac{i\lambda|\Lambda|^2}{2\pi}\int_{-\infty}^{+\infty}d\tau\left\{G_R(\tau)[(\tilde{h}(\omega,\tau)-\tilde{h}(\omega,-\tau))G_L(\tau)\right.\quad\text{(B.39)}$$
$$\left.-(\widetilde{K}_0(\omega,\tau)+\widetilde{K}_0(\omega,-\tau))\partial_\tau G_L(\tau)]+(\omega\to-\omega)\right\}.$$

This formula, together with Eqs. (B.38) and (B.34), provides the expression for the finite-frequency noise. We can also obtain an equivalent formula, which is more convenient to evaluate the zero-frequency limit. By repeatedly integrating by parts, and exploiting the relation (B.37) between the functions $\tilde{h}$ and $\widetilde{K}_0$, we arrive at

$$\Sigma_{44}^{(02)}(\omega)+\Sigma_{44}^{(20)}(\omega)=\frac{i\lambda|\Lambda|^2}{2\pi}\int_{-\infty}^{+\infty}d\tau\left\{\partial_\tau G_R(\tau)G_L(\tau)\left[\widetilde{K}_0(\omega,\tau)e^{-i\omega\tau}+\widetilde{K}_0(\omega,-\tau)e^{i\omega\tau}\right]\right.$$
$$+G_R(\tau)\partial_\tau G_L(\tau)\left[\left(e^{-i\omega\tau}-1\right)\widetilde{K}_0(\omega,\tau)+\left(e^{i\omega\tau}-1\right)\widetilde{K}_0(\omega,-\tau)\right]\quad\text{(B.40)}$$
$$\left.-i\omega G_R(\tau)G_L(\tau)\left[\widetilde{K}_0(\omega,\tau)e^{-i\omega\tau}-\widetilde{K}_0(\omega,-\tau)e^{i\omega\tau}\right]+(\omega\to-\omega)\right\}.$$

The zero-frequency limit is therefore given by

$$\Sigma_{44}^{(02)}(0)+\Sigma_{44}^{(20)}(0)=2\times\frac{i\lambda|\Lambda|^2}{2\pi}\int_{-\infty}^{+\infty}d\tau\,\partial_\tau G_R(\tau)\left[\widetilde{K}_0(0,\tau)+\widetilde{K}_0(0,-\tau)\right]G_L(\tau)$$
$$=8iT_2\lambda|\Lambda|^2\int_{-\infty}^{+\infty}d\tau\,\partial_\tau G_R(\tau)[-1+\pi T_2\tau\coth(\pi T_2\tau)]G_L(\tau).\quad\text{(B.41)}$$

Finally, we exploit the Green's function identity

$$\lambda\pi T_2\coth(\pi T_2\tau)G_L(\tau)=-\partial_\tau G_L(\tau),\quad\text{(B.42)}$$

and we arrive at two equivalent final expressions

$$\Sigma_{44}^{(02)}(0)+\Sigma_{44}^{(20)}(0)=4(\lambda-1)T_2 J_T+8i|\Lambda|^2 T_2\int_{-\infty}^{+\infty}d\tau\,\tau G_L(\tau)\partial_\tau^2 G_R(\tau)\quad\text{(B.43)}$$

$$=4\lambda T_2 J_T-8i|\Lambda|^2 T_2\int_{-\infty}^{+\infty}d\tau\,\tau\,\partial_\tau G_L(\tau)\,\partial_\tau G_R(\tau),\quad\text{(B.44)}$$

where we recalled the expression for the heat tunneling current (B.8).

The remaining terms are obtained with very similar calculations and they read

$$\Sigma_{33}^{(02)}(0)+\Sigma_{33}^{(20)}(0)=-4\lambda T_1 J_T-8i|\Lambda|^2 T_1\int_{-\infty}^{+\infty}d\tau\,\tau\,\partial_\tau G_L(\tau)\,\partial_\tau G_R(\tau),\quad\text{(B.45)}$$

$$\Sigma_{34}^{(02)}(0)+\Sigma_{34}^{(20)}(0)=2\lambda(T_1-T_2)J_T+4i|\Lambda|^2(T_1+T_2)\int_{-\infty}^{+\infty}d\tau\,\tau\,\partial_\tau G_L(\tau)\,\partial_\tau G_R(\tau).\quad\text{(B.46)}$$

Table 2: Auto- and cross-correlation heat-current noises at zero voltage bias and zero frequency, $S_{\alpha\beta}^{JJ}$ with $\alpha, \beta = L, R$. The expressions are given to $\mathcal{O}|(\Lambda)|^2$ in the tunneling amplitude $\Lambda$, and we have defined the integral $\mathcal{J} \equiv 4i|\Lambda|^2 \int_{-\infty}^{+\infty} d\tau\, \tau\, \partial_\tau G_R\, \partial_\tau G_L$.

| $S_{\alpha\beta}^{JJ}$ | 3 | 4 |
|---|---|---|
| 3 | $2\dfrac{\pi^2 k_{\mathrm{B}}^3}{3h}T_1^3 - 4\lambda k_{\mathrm{B}}T_1 J_T + S_{TT}^{JJ} - 2k_{\mathrm{B}}T_1\mathcal{J}$ | $-S_{TT}^{JJ} + 2\lambda k_{\mathrm{B}}(T_1 - T_2)J_T + k_{\mathrm{B}}(T_1 + T_2)\mathcal{J}$ |
| 4 | $-S_{TT}^{JJ} + 2\lambda k_{\mathrm{B}}(T_1 - T_2)J_T + k_{\mathrm{B}}(T_1 + T_2)\mathcal{J}$ | $2\dfrac{\pi^2 k_{\mathrm{B}}^3}{3h}T_2^3 + 4\lambda k_{\mathrm{B}}T_2 J_T + S_{TT}^{JJ} - 2k_{\mathrm{B}}T_2\mathcal{J}$ |

In the presence of a finite voltage bias, $V \neq 0$, in addition to the temperature bias, the above results are generalized as follows:

$$
\begin{aligned}
\Sigma_{44}^{(02)}(0) + \Sigma_{44}^{(20)}(0) = {} & 4\lambda T_2 \langle \hat{J}_4^{(2)}\rangle - 4V T_2 \partial_V \langle \hat{J}_4^{(2)}\rangle \\
& - 8i|\Lambda|^2 T_2 \int_{-\infty}^{+\infty} d\tau\, \tau\, \cos(e\nu V\tau)\partial_\tau G_L(\tau)\,\partial_\tau G_R(\tau),
\end{aligned}
\tag{B.47}
$$

$$
\begin{aligned}
\Sigma_{33}^{(02)}(0) + \Sigma_{33}^{(20)}(0) = {} & 4\lambda T_1 \langle \hat{J}_3^{(2)}\rangle - 4V T_1 \partial_V \langle \hat{J}_3^{(2)}\rangle \\
& - 8i|\Lambda|^2 T_1 \int_{-\infty}^{+\infty} d\tau\, \tau\, \cos(e\nu V\tau)\partial_\tau G_L(\tau)\,\partial_\tau G_R(\tau),
\end{aligned}
\tag{B.48}
$$

$$
\begin{aligned}
\Sigma_{34}^{(02)}(0) + \Sigma_{34}^{(20)}(0) = {} & 2\lambda T_1 \langle \hat{J}_4^{(2)}\rangle + 2\lambda T_2 \langle \hat{J}_3^{(2)}\rangle \\
& + 4i|\Lambda|^2 (T_1 + T_2) \int_{-\infty}^{+\infty} d\tau\, \tau\, \cos(e\nu V\tau)\partial_\tau G_L(\tau)\,\partial_\tau G_R(\tau),
\end{aligned}
\tag{B.49}
$$

where the expressions for the average heat currents $\langle \hat{J}_{3,4}^{(2)}\rangle$ are given in Eq. (B.6).

### B.5 Summary of heat-current noises

Gathering the results from all above subsections in Appendix B, we summarize the expressions for the auto- and cross-correlated heat-current noises in Tab. 2. These results are those stated in Eqs. (37) and (41) in the main text.

## C  Derivation of mixed noise components

### C.1 General expressions

We decompose the mixed noise perturbatively as

$$
S_{\alpha\beta}^{IJ} = M_{\alpha\beta}^{(00)} + M_{\alpha\beta}^{(11)} + M_{\alpha\beta}^{(02)} + M_{\alpha\beta}^{(20)},
\tag{C.1}
$$

where, in analogy to the charge and heat noise components, we define

$$
M_{\alpha\beta}^{(ij)} = \left\langle \left\{ \delta \hat{I}_\alpha^{(i)}(t_1), \delta \hat{J}_\beta^{(j)}(t_2)\right\}\right\rangle.
\tag{C.2}
$$

We readily find that the equilibrium component $M_{\alpha\beta}^{(00)}$ vanishes, as it reduces to expectation values of the form $\langle \partial_{t_1}\hat{\phi}_\alpha(t_1)[\partial_{t_2}\hat{\phi}_\beta(t_2)]^2\rangle$, which contain an unbalanced number of bosonic

operators and thus evaluates to zero by Wick's theorem. With the same approach as for the charge and heat noises in the above Appendixes, we obtain the "tunneling" terms as

$$M_{33}^{(11)} = 4e\nu|\Lambda|^2 \int_{-\infty}^{+\infty} d\tau \sin(e\nu V\tau)G_L(\tau)\partial_\tau G_R(\tau) = -M_{43}^{(11)} \equiv M_{TT} - \frac{V}{2}S_{TT}^{II}, \qquad \text{(C.3)}$$

$$M_{44}^{(11)} = -4e\nu|\Lambda|^2 \int_{-\infty}^{+\infty} d\tau \sin(e\nu V\tau)G_R(\tau)\partial_\tau G_L(\tau) = -M_{34}^{(11)} \equiv M_{TT} + \frac{V}{2}S_{TT}^{II}, \qquad \text{(C.4)}$$

with

$$M_{TT} \equiv 2e\nu|\Lambda|^2 \int_{-\infty}^{+\infty} d\tau \sin(e\nu V\tau)[G_L(\tau)\partial_\tau G_R(\tau) - G_R(\tau)\partial_\tau G_L(\tau)]. \qquad \text{(C.5)}$$

We note here the relations $M_{33}^{(11)} = -M_{43}^{(11)}$ and $M_{44}^{(11)} = -M_{34}^{(11)}$ which are a direct consequence of the operator identity $\hat{I}_3^{(1)} = -\hat{I}_4^{(1)}$, see Eq. (A.3). These relations also show that the "tunneling" mixed noise components satisfy the sum rule $\sum_{\alpha\beta} M_{\alpha\beta}^{(11)} = 0$ for $\alpha = 3, 4$.

Next, a straightforward but long calculation of the correlations between the unperturbed currents and their corrections induced by the tunneling leads to the following expressions for the crossed terms

$$\begin{cases} M_{33}^{(20)} = -2\lambda T_1 I_T + 2T_1\partial_V \langle\hat{J}_3^{(2)}\rangle \\ M_{33}^{(02)} = -2T_1 I_T + 2T_1\partial_V \langle\hat{J}_3^{(2)}\rangle \end{cases} \rightarrow M_{33}^{(02+20)} = -2T_1(1+\lambda)I_T + 4T_1\partial_V \langle\hat{J}_3^{(2)}\rangle, \qquad \text{(C.6a)}$$

$$\begin{cases} M_{44}^{(20)} = +2\lambda T_2 I_T - 2T_2\partial_V \langle\hat{J}_4^{(2)}\rangle \\ M_{44}^{(02)} = +2T_2 I_T - 2T_2\partial_V \langle\hat{J}_4^{(2)}\rangle \end{cases} \rightarrow M_{44}^{(02+20)} = +2T_2(1+\lambda)I_T - 4T_2\partial_V \langle\hat{J}_4^{(2)}\rangle, \qquad \text{(C.6b)}$$

$$\begin{cases} M_{34}^{(20)} = -2\lambda T_2 I_T + 2T_2\partial_V \langle\hat{J}_4^{(2)}\rangle \\ M_{34}^{(02)} = +2T_2\partial_V \langle\hat{J}_4^{(2)}\rangle \end{cases} \rightarrow M_{34}^{(02+20)} = -2\lambda T_2 I_T + 2(T_1+T_2)\partial_V \langle\hat{J}_4^{(2)}\rangle, \qquad \text{(C.6c)}$$

$$\begin{cases} M_{43}^{(20)} = +2\lambda T_1 I_T - 2T_1\partial_V \langle\hat{J}_3^{(2)}\rangle \\ M_{43}^{(02)} = -2T_2\partial_V \langle\hat{J}_3^{(2)}\rangle \end{cases} \rightarrow M_{43}^{(02+20)} = +2\lambda T_1 I_T - 2(T_1+T_2)\partial_V \langle\hat{J}_3^{(2)}\rangle, \qquad \text{(C.6d)}$$

with the average heat currents $\langle\hat{J}_\alpha^{(2)}\rangle$ given in Eq. (B.6). Combining all components, we arrive at the mixed noise components

$$S_{33}^{IJ} = +M_{TT} - \frac{V}{2}S_{TT}^{II} - 2T_1(1+\lambda)I_T + 4T_1\partial_V \langle\hat{J}_3^{(2)}\rangle, \qquad \text{(C.7a)}$$

$$S_{44}^{IJ} = +M_{TT} + \frac{V}{2}S_{TT}^{II} + 2T_2(1+\lambda)I_T - 4T_2\partial_V \langle\hat{J}_4^{(2)}\rangle, \qquad \text{(C.7b)}$$

$$S_{34}^{IJ} = -M_{TT} - \frac{V}{2}S_{TT}^{II} - 2\lambda T_2 I_T + 2(T_1+T_2)\partial_V \langle\hat{J}_4^{(2)}\rangle, \qquad \text{(C.7c)}$$

$$S_{43}^{IJ} = -M_{TT} + \frac{V}{2}S_{TT}^{II} + 2\lambda T_1 I_T - 2(T_1+T_2)\partial_V \langle\hat{J}_3^{(2)}\rangle, \qquad \text{(C.7d)}$$

which are given in Eq. (74) in the main text.

## C.2 Relation with the thermoelectric response

In this section, we prove Eq. (80) in the main text, namely the relation between mixed noise and the differential thermoelectric conductance.

To this end, consider a nonequilibrium situation with finite voltage bias ($V \neq 0$), but vanishing temperature bias, $\Delta T \to 0$. As a result, our calculations involve only a single Green's

function at temperature $\bar{T}$, denoted as

$$G_L(\tau) = G_R(\tau) \equiv G(\tau) = \frac{1}{2\pi a} \left( \frac{\sinh(i\pi\bar{T}\tau_0)}{\sinh[\pi\bar{T}(i\tau_0 - \tau)]} \right)^\lambda. \tag{C.8}$$

As our next step, we combine the mixed noise components (C.3), (C.4), the average heat currents (B.6), and the charge tunneling current (A.32a) and perform an expansion at first order in $eV/\bar{T}$. This expansion results in

$$M_{33}^{(11)} = +4(e\nu)^2|\Lambda|^2 \frac{V}{\bar{T}} \int_{-\infty}^{+\infty} dx\, x\, G(x/\bar{T})G'(x/\bar{T}) \equiv +4\mathcal{L}_1 \frac{V}{\bar{T}}, \tag{C.9a}$$

$$M_{44}^{(11)} = -4(e\nu)^2|\Lambda|^2 \frac{V}{\bar{T}} \int_{-\infty}^{+\infty} dx\, x\, G(x/\bar{T})G'(x/\bar{T}) \equiv -4\mathcal{L}_1 \frac{V}{\bar{T}}, \tag{C.9b}$$

$$T_{1,2}\partial_V \langle \hat{J}_3^{(2)} \rangle = -2i(e\nu)^2|\Lambda|^2 \frac{V}{\bar{T}} \int_{-\infty}^{+\infty} dx\, x^2\, G(x/\bar{T})G'(x/\bar{T}) \equiv -2\mathcal{L}_2 \frac{V}{\bar{T}}, \tag{C.9c}$$

$$T_{1,2}\partial_V \langle \hat{J}_4^{(2)} \rangle = -2i(e\nu)^2|\Lambda|^2 \frac{V}{\bar{T}} \int_{-\infty}^{+\infty} dx\, x^2\, G(x/\bar{T})G'(x/\bar{T}) \equiv -2\mathcal{L}_2 \frac{V}{\bar{T}}, \tag{C.9d}$$

$$T_{1,2}I_T = +2i(e\nu)^2|\Lambda|^2 \frac{V}{\bar{T}} \int_{-\infty}^{+\infty} dx\, x\, [G(x/\bar{T})]^2 \equiv +2\mathcal{L}_0 \frac{V}{\bar{T}}, \tag{C.9e}$$

where we introduced the dimensionless variable $x = \bar{T}\tau$ and defined the three integrals

$$\mathcal{L}_0 = i(e\nu)^2|\Lambda|^2 \int_{-\infty}^{+\infty} dx\, x\, [G(x/\bar{T})]^2, \tag{C.10a}$$

$$\mathcal{L}_1 = (e\nu)^2|\Lambda|^2 \int_{-\infty}^{+\infty} dx\, x\, G(x/\bar{T})G'(x/\bar{T}), \tag{C.10b}$$

$$\mathcal{L}_2 = i(e\nu)^2|\Lambda|^2 \int_{-\infty}^{+\infty} dx\, x^2\, G(x/\bar{T})G'(x/\bar{T}). \tag{C.10c}$$

We define the finite-bias thermoelectric conductance as

$$\tilde{L} = \frac{\partial I_T}{\partial \Delta T} = 2i(e\nu)^2|\Lambda|^2 \int_{-\infty}^{+\infty} d\tau\, \sin(e\nu V\tau) \frac{\partial}{\partial \Delta T}[G_R(\tau)G_L(\tau)]. \tag{C.11}$$

In the limit $\Delta T \to 0$ and $eV/\bar{T} \ll 1$, we get

$$\tilde{L} = \frac{2i(e\nu)^2|\Lambda|^2}{\bar{T}^2} \frac{V}{\bar{T}} \int_{-\infty}^{+\infty} dx\, x^2\, G(x/\bar{T})G'(x/\bar{T}) = \frac{2\mathcal{L}_2}{\bar{T}^2} \frac{V}{\bar{T}}. \tag{C.12}$$

The integrals (C.10) can be evaluated analytically as follows (see also App. E)

$$\mathcal{L}_0 = (e\nu)^2 \bar{T}^{2\lambda} \tau_0^{2\lambda-2} \frac{|\Lambda|^2}{v_F^2} \int_{-\infty}^{+\infty} \frac{dx}{4\pi^2} ix \left( \frac{i\pi}{\sinh[\pi(i\bar{T}\tau_0 - x)]} \right)^{2\lambda} \tag{C.13a}$$

$$= (e\nu)^2 \frac{\bar{T}^{2\lambda}}{8\pi} (\pi\tau_0)^{2\lambda-2} \frac{|\Lambda|^2}{v_F^2} \int_{-\infty}^{+\infty} \frac{dz}{[\cosh(z)]^{2\lambda}} = (2\pi\tau_0)^{2\lambda-2} \frac{|\Lambda|^2}{v_F^2} (e\nu)^2 \frac{\bar{T}^{2\lambda}}{4\pi} \frac{\Gamma^2(\lambda)}{\Gamma(2\lambda)},$$

$$\mathcal{L}_1 = (ev)^2 \tau_0^{2\lambda-2} \frac{|\Lambda|^2}{v_F^2} \int_{-\infty}^{+\infty} \frac{dx}{4\pi^2} x \left( \frac{i\pi\bar{T}}{\sinh[\pi(i\bar{T}\tau_0 - x)]} \right)^\lambda \partial_x \left( \frac{i\pi\bar{T}}{\sinh[\pi(i\bar{T}\tau_0 - x)]} \right)^\lambda$$

$$= -(ev)^2 \frac{\bar{T}^{2\lambda}}{4\pi} (\pi\tau_0)^{2\lambda-2} \frac{|\Lambda|^2}{v_F^2} \lambda \int_{-\infty}^{+\infty} dz \frac{z \sinh(z)}{[\cosh(z)]^{1+2\lambda}} = -\mathcal{L}_0, \tag{C.13b}$$

$$\mathcal{L}_2 = (ev)^2 \tau_0^{2\lambda-2} \frac{|\Lambda|^2}{v_F^2} \int_{-\infty}^{+\infty} \frac{dx}{4\pi^2} ix^2 \left( \frac{i\pi}{\sinh[\pi(i\bar{T}\tau_0 - x)]} \right)^\lambda \partial_x \left( \frac{i\pi}{\sinh[\pi(i\bar{T}\tau_0 - x)]} \right)^\lambda$$

$$= -(ev)^2 \frac{\bar{T}^{2\lambda}}{4\pi} (\pi\tau_0)^{2\lambda-2} \frac{|\Lambda|^2}{v_F^2} \lambda \int_{-\infty}^{+\infty} dz \frac{z \sinh(z)}{[\cosh(z)]^{1+2\lambda}} = -\mathcal{L}_0. \tag{C.13c}$$

In evaluating all these integrals, we performed the change of variable $x = z/\pi + \tau_0 - i/2$ in the complex plane and deformed the contour back to the real axis, exploiting the finite cutoff $\tau_0$ [21]. Substituting the evaluated $\mathcal{L}_i$ integrals into the mixed noise components (C.9) and then into Eq. (C.7), we find the relations

$$S_{33}^{IJ}(0) = -S_{44}^{IJ}(0) = -4\lambda\mathcal{L}_0 \frac{V}{\bar{T}}, \tag{C.14a}$$

$$S_{34}^{IJ}(0) = -S_{43}^{IJ}(0) = 4(1-\lambda)\mathcal{L}_0 \frac{V}{\bar{T}}. \tag{C.14b}$$

Similarly, the conductance in Eq. (C.12) becomes

$$\tilde{L} = -\frac{2\mathcal{L}_0}{\bar{T}^2} \frac{V}{\bar{T}}, \tag{C.15}$$

and therefore we obtain Eq. (80) in the main text.

# D  Scaling dimension modification by inter-channel interaction

## D.1  Charge transport

In this Appendix, we give an example on how the addition of a local density-density interaction at the QPC modifies the scaling dimension $\lambda$ of the tunneling quasiparticles, from the ideal case $\lambda = v$ to $\lambda \neq v$.

To this end, we consider adding to the free Hamiltonian $\hat{H}_0$ in (7), not only the tunneling term (10), but also the following local coupling between the $R/L$ channels:

$$\hat{H}_u = \frac{2u}{4\pi} \int_{-\infty}^{+\infty} dx\, \delta(x) \partial_x \hat{\phi}_R(x) \partial_x \hat{\phi}_L(x). \tag{D.1}$$

Here, $u$ parametrizes the interaction strength and the location of the interaction coincides with that of the QPC, here at $x = 0$. With this addition, $\hat{H}_0 + \hat{H}_u$ is not diagonal in the bosons $\hat{\phi}_{R/L}$ anymore. Still, we need to evaluate the local quasiparticle Green's functions,

$$G_{R/L}(0, t) = \langle \hat{\psi}_{R/L}^\dagger(0, t) \hat{\psi}_{R/L}(0, 0) \rangle, \tag{D.2}$$

to compute observables related to the charge tunneling. To find these Green's functions when $u \neq 0$, we use the following approach: First, we locally diagonalize $\hat{H}_0 + \hat{H}_u$ with the transformation

$$\begin{pmatrix} \hat{\phi}_+(0, t) \\ \hat{\phi}_-(0, t) \end{pmatrix} = \begin{pmatrix} \alpha & \beta \\ \beta & \alpha \end{pmatrix} \begin{pmatrix} \hat{\phi}_R(0, t) \\ \hat{\phi}_L(0, t) \end{pmatrix}. \tag{D.3}$$

Here, the coefficients $\alpha, \beta$ depend on the interaction strength $u$ and the velocity $v_F$ as

$$\alpha = \cosh(\theta), \qquad \beta = \sinh(\theta), \qquad \tanh(2\theta) = u/v_F. \tag{D.4}$$

For $u = 0$, we have $\alpha = 1 - \beta = 1$, so that in this case $\hat{\phi}_{\pm}(0,t) = \hat{\phi}_{R/L}(0,t)$ as expected. The new modes $\hat{\phi}_{\pm}(0,t)$ are the local eigenmodes at the point $x = 0$ and the local Green's functions at this point can be straightforwardly evaluated. We may thus write

$$\langle \hat{\psi}_R^{\dagger}(0,t)\hat{\psi}_R(0,0)\rangle \times \langle \hat{\psi}_L^{\dagger}(0,t)\hat{\psi}_L(0,0)\rangle = \frac{1}{(2\pi a)^2} e^{\nu(\alpha^2+\beta^2)[\mathcal{G}_+(0,t)+\mathcal{G}_-(0,t)]}, \tag{D.5}$$

in terms of the diagonal bosonic Green's functions $\mathcal{G}_{\pm}(0,t) = \langle \hat{\phi}_{\pm}(0,t)\hat{\phi}_{\pm}(0,0)\rangle - \langle \hat{\phi}_{\pm}^2(0,0)\rangle$.

Our next step is to express $\mathcal{G}_{\pm}(0,t)$ in terms of the known, "incoming" Green's functions, i.e., $\mathcal{G}_{R/L}(x \neq 0, t)$, which are given in terms of the original bosonic fields $\hat{\phi}_{R/L}(t, \mp x_{1/2})$. These bosons are in equilibrium with their respective sources, at temperatures $T_1$ and $T_2$ and at the locations $\mp x_{1/2}$. To this end, we solve a bosonic scattering problem with three regions: 1) the region left of the QPC, 2) the central QPC region $x = 0$, and 3) the region right of the QPC. In brief, the matrix (D.3) constitutes the transfer matrix, $\mathcal{T}$ for this scattering problem:

$$\mathcal{T} = \begin{pmatrix} \alpha & \beta \\ \beta & \alpha \end{pmatrix} = \frac{1}{T}\begin{pmatrix} 1 & R \\ R & 1 \end{pmatrix}, \tag{D.6}$$

with $T^2 + R^2 = 1$. Solving the scattering problem for the central region bosons, $\hat{\phi}_{\pm}(0,t)$ in terms of the incoming modes, we find

$$\hat{\phi}_+(0,t) = \frac{R}{T}\hat{\phi}_L(x_2,t) + \frac{1}{T}\hat{\phi}_R(-x_1,t), \tag{D.7a}$$

$$\hat{\phi}_-(0,t) = \frac{1}{T}\hat{\phi}_L(x_2,t) + \frac{R}{T}\hat{\phi}_R(-x_1,t), \tag{D.7b}$$

and since the bosons $\hat{\phi}_{R/L}$ at the sources are uncorrelated, it follows that

$$\mathcal{G}_+(0,t) = \frac{R^2}{T^2}\mathcal{G}_L(x_2,t) + \frac{1}{T^2}\mathcal{G}_R(-x_1,t), \tag{D.8a}$$

$$\mathcal{G}_-(0,t) = \frac{1}{T^2}\mathcal{G}_L(x_2,t) + \frac{R^2}{T^2}\mathcal{G}_R(-x_1,t). \tag{D.8b}$$

Finally, we identify $R^2/T^2 = \beta^2$ and $1/T^2 = \alpha^2$ and upon inserting into Eq. (D.5), we arrive at

$$\langle \hat{\psi}_R^{\dagger}(0,t)\hat{\psi}_R(0,t)\rangle \times \langle \hat{\psi}_L^{\dagger}(0,t)\hat{\psi}_L(0,t)\rangle = \frac{1}{(2\pi a)^2} e^{\nu(\alpha^2+\beta^2)^2[\mathcal{G}_R(-x_1,t)+\mathcal{G}_L(x_2,t)]}. \tag{D.9}$$

Thus, we see that $H_u$ changes the scaling dimension from $\nu$ to

$$\lambda \equiv \nu(\alpha^2+\beta^2)^2 = \nu\cosh^2(2\theta) = \frac{\nu}{1-u^2/v_F^2}. \tag{D.10}$$

For $u = 0$ (i.e., without the coupling term $\hat{H}_u$), we have $\alpha = 1/T = 1$, $\beta = R/T = 0$ and $\lambda = \nu$ as expected. We emphasize that the temperatures entering the problem are the two source contact temperatures $T_{1/2}$.

## D.2 Heat transport

In the above calculation, we evaluated the product of $L$ and $R$ quasiparticle Green's functions, which is sufficient to obtain the observables related to the charge transport, as is clear from Eqs. (18) and (25). The situation changes when the heat transport is considered: in this case, we deal with (for example) quantities like $G_R(t)\partial_t G_L(t)$, see Eq. (41). Thus, it is important to critically analyze the behavior of the $L$ and $R$ local Green's functions separately. Within the toy model in this Appendix, we have (in terms of the "incoming" Green's functions)

$$\langle\hat{\psi}_R^\dagger(0,t)\hat{\psi}_R(0,t)\rangle = \frac{1}{2\pi a}e^{\lambda_+\mathcal{G}_R(t)+\lambda_-\mathcal{G}_L(t)}\,, \tag{D.11}$$

$$\langle\hat{\psi}_L^\dagger(0,t)\hat{\psi}_L(0,t)\rangle = \frac{1}{2\pi a}e^{\lambda_-\mathcal{G}_R(t)+\lambda_+\mathcal{G}_L(t)}\,, \tag{D.12}$$

with

$$\lambda_+ = \alpha^4+\beta^4\,, \tag{D.13}$$

$$\lambda_- = 2\alpha^2\beta^2\,. \tag{D.14}$$

When calculating the tunneling heat noise, this renormalization gives rise to the usual expansions in powers of $\Delta T/2\bar{T}$

$$S_{TT}^{JJ} = S_0^{JJ}\left[1+\mathcal{C}_Q^{(2)}\left(\frac{\Delta T}{2\bar{T}}\right)^2+\dots\right]\,, \tag{D.15}$$

with prefactor

$$S_0^{JJ} = \frac{|\Lambda|^2}{v_F^2}\bar{T}^3\frac{2\pi\lambda^2}{1+2\lambda}(2\pi\bar{T}\tau_0)^{2\lambda-2}\frac{\Gamma^2(\lambda)}{\Gamma(2\lambda)}\,, \qquad \lambda = \lambda_++\lambda_-\,, \tag{D.16}$$

and coefficient

$$\begin{aligned}
\mathcal{C}_Q^{(2)} &= \frac{\lambda\left(-4\lambda+\pi^2(\lambda+2)-2(\lambda+2)\psi^{(1)}(\lambda+1)-2\right)}{2(2\lambda+3)} \\
&\quad + (\lambda_+-\lambda_-)^2\times\frac{\lambda\left(\pi^2(\lambda+1)-6\right)-2\lambda(\lambda+1)\psi^{(1)}(\lambda+1)-3}{\lambda^2(2\lambda+3)}\,.
\end{aligned} \tag{D.17}$$

By comparing this result with Eq. (44a) in the main text, we see that, *at least within this toy model*, the two expressions agree only for $\lambda_- = 0$, which implies the ideal case $\lambda = \nu$. Otherwise, both parameters $\lambda_\pm$ appear in the result. This feature stands in stark contrast with the charge transport properties, where the relevant parameter is always the sum $\lambda = \lambda_++\lambda_-$. This happens because it is the simple product of $L$ and $R$ Green's functions that determines all the relevant observables. Then, for charge transport, we can equivalently *assume* that both local Green's functions separately have a renormalized exponent $\nu\to\lambda$. The same assumption is required for the validity of the results concerning heat-related observables in the main text (beyond the ideal case $\lambda = \nu$, for which they are obviously valid). This does not happen in our toy model, but it might apply in more complicated ones, where the scaling dimension renormalization relies on different physical mechanisms (see the discussion below Eq. (20) for examples).

## D.3 Unequal scaling dimensions on the two edges

Another possibility is that the two edges coupled by the tunneling Hamiltonian have inherently different scaling dimensions [111], which implies that the local quasiparticle Green's functions

read

$$\langle \hat{\psi}_{R,L}^{\dagger}(0,\tau)\hat{\psi}_{R,L}(0,\tau)\rangle = \frac{1}{2\pi a}\left(\frac{\sinh(i\pi T_{1,2}\tau_0)}{\sinh[\pi T_{1,2}(i\tau_0-\tau)]}\right)^{\lambda_{1,2}} \equiv G_{1,2}(\tau), \tag{D.18}$$

with $\lambda_1 \neq \lambda_2$. This property breaks the symmetry of the setup, introducing a difference between the top and the bottom edge. The heat transport observables now read

$$J_T = -2i|\Lambda|^2 \int_{-\infty}^{+\infty} d\tau\, G_2(\tau)\partial_\tau G_1(\tau), \tag{D.19a}$$

$$S_{TT}^{JJ} = 4|\Lambda|^2 \int_{-\infty}^{+\infty} d\tau\, \partial_\tau G_1(\tau)\partial_\tau G_2(\tau), \tag{D.19b}$$

$$S_{33}^{JJ} = S_{11}^{JJ} + S_{TT}^{JJ} - 4\lambda_1 T_1 J_T - 8i|\Lambda|^2 T_1 \int_{-\infty}^{+\infty} d\tau\, \tau\, \partial_\tau G_1(\tau)\partial_\tau G_2(\tau), \tag{D.19c}$$

$$S_{44}^{JJ} = S_{22}^{JJ} + S_{TT}^{JJ} + 4\lambda_2 T_2 J_T - 8i|\Lambda|^2 T_2 \int_{-\infty}^{+\infty} d\tau\, \tau\, \partial_\tau G_1(\tau)\partial_\tau G_2(\tau), \tag{D.19d}$$

$$S_{34}^{JJ} = -S_{TT}^{JJ} + 2(\lambda_1 T_1 - \lambda_2 T_2)J_T + 4i|\Lambda|^2(T_1+T_2)\int_{-\infty}^{+\infty} d\tau\, \tau\, \partial_\tau G_1(\tau)\partial_\tau G_2(\tau), \tag{D.19e}$$

$$S_{43}^{JJ} = S_{34}^{JJ}. \tag{D.19f}$$

As a consequence of the broken symmetry, we expect to find also odd coefficients in the $\Delta T$ power expansion, even in the presence of a symmetric bias. Indeed, using as an example the heat tunneling noise, we find the usual expansion

$$S_{TT}^{JJ} = S_0^{JJ}\left[1 + \mathcal{C}_Q^{(1)}\left(\frac{\Delta T}{2\bar{T}}\right) + \mathcal{C}_Q^{(2)}\left(\frac{\Delta T}{2\bar{T}}\right)^2 + \mathcal{C}_Q^{(3)}\left(\frac{\Delta T}{2\bar{T}}\right)^3 \dots\right], \tag{D.20}$$

with prefactor

$$S_0^{JJ} = \frac{|\Lambda|^2}{v_F^2} 2\pi\bar{T}^3 \frac{\lambda_1\lambda_2}{1+2\bar{\lambda}}(2\pi\bar{T}\tau_0)^{2\bar{\lambda}-2}\frac{\Gamma^2(\bar{\lambda})}{\Gamma(2\bar{\lambda})}, \quad \text{where} \quad \bar{\lambda} \equiv \frac{\lambda_1+\lambda_2}{2}, \tag{D.21}$$

and coefficients

$$\mathcal{C}_Q^{(1)} = (\lambda_1-\lambda_2)\frac{1+\bar{\lambda}-2\bar{\lambda}^2}{2\bar{\lambda}(1+\bar{\lambda})}, \tag{D.22a}$$

$$\mathcal{C}_Q^{(2)} = \frac{\left(\pi^2(3\bar{\lambda}+4)-2(2\bar{\lambda}+7)\right)\bar{\lambda}^2 - 2(3\bar{\lambda}+4)\bar{\lambda}^2\psi^{(1)}(\bar{\lambda})+8}{2\bar{\lambda}(2\bar{\lambda}+3)}$$
$$+(\lambda_1-\lambda_2)^2 \times \frac{4-3(4+\pi^2)\bar{\lambda}+8\bar{\lambda}^2+6\bar{\lambda}\psi^{(1)}(\bar{\lambda})}{8\bar{\lambda}(2\bar{\lambda}+3)}, \tag{D.22b}$$

$$\mathcal{C}_Q^{(3)} = (\lambda_1-\lambda_2)\left(\frac{12\bar{\lambda}^4+38\bar{\lambda}^3-12\bar{\lambda}^2-38\bar{\lambda}-12}{12\bar{\lambda}(1+\bar{\lambda})(2+\bar{\lambda})} + \frac{\bar{\lambda}^2(3\bar{\lambda}^2+\bar{\lambda}-6)[6\psi^{(1)}(1+\bar{\lambda})-3\pi^2]}{12\bar{\lambda}(1+\bar{\lambda})(2+\bar{\lambda})}\right)$$
$$+(\lambda_1-\lambda_2)^3 \times \frac{\pi^2(9\bar{\lambda}^2-9\bar{\lambda}-6)-4\bar{\lambda}(\bar{\lambda}-1)(2\bar{\lambda}-1)+6(3\bar{\lambda}^2-3\bar{\lambda}+2)\psi^{(1)}(\bar{\lambda})}{64\bar{\lambda}(\bar{\lambda}+1)(\bar{\lambda}+2)}. \tag{D.22c}$$

As expected, the odd coefficients vanish when $\lambda_1 = \lambda_2$ and the even ones reduce to those given in the main text.

# E  Some useful integral identities

Our approach to evaluating integrals over Green's functions and their derivatives is based on the integral identity [112]

$$\int_{-\infty}^{\infty} \frac{\cosh(2bz)}{(\cosh(z))^{2a}} dz = 2 \times 4^{a-1} \mathcal{B}(a+b, a-b). \tag{E.1}$$

Here,

$$\mathcal{B}(z_1, z_2) = \frac{\Gamma(z_1)\Gamma(z_2)}{\Gamma(z_1 + z_2)}, \tag{E.2}$$

is Euler's beta function and $\Gamma(z)$ is the Gamma function. By repeated differentiation of Eq. (E.1) with respect to $b$, we further obtain, for any positive integer $m$,

$$\int_{-\infty}^{\infty} \frac{z^{2m}\cosh(2bz)}{(\cosh(z))^{2a}} dz = \frac{1}{2^{2m}} \frac{\partial^{2m}}{\partial b^{2m}} \left[ 2 \times 4^{a-1} \mathcal{B}(a+b, a-b) \right], \tag{E.3}$$

$$\int_{-\infty}^{\infty} \frac{z^{2m-1}\sinh(2bz)}{(\cosh(z))^{2a}} dz = \frac{1}{2^{2m-1}} \frac{\partial^{2m-1}}{\partial b^{2m-1}} \left[ 2 \times 4^{a-1} \mathcal{B}(a+b, a-b) \right]. \tag{E.4}$$

Our strategy in this paper is to expand all integrals involving Green's functions and their derivatives into terms on the form (E.1), (E.3), or (E.4) and then sum up all contributions.

# F  Fourier transforms of the Green's function

In the time-domain, the exponentiated bosonic (retarded) Green's function at temperature $T_\alpha$ is given as

$$e^{\lambda \mathcal{G}_{R/L}(\tau)} = \left[ \frac{\sinh(i\pi T \tau_0)}{\sinh(\pi T_{1,2}(i\tau_0 - \tau))} \right]^{\lambda}, \tag{F.1}$$

where $\tau_0 = a/v_F$ is the UV cutoff in the time domain. The Fourier transform of (F.1) can be evaluated to [21]

$$P_{1,2}(E) \equiv \int_{-\infty}^{+\infty} d\tau\, e^{iE\tau} e^{\lambda \mathcal{G}_{R/L}(\tau)} = (2\pi T_{1,2}\tau_0)^{\lambda-1} \frac{\tau_0}{\Gamma(\lambda)} e^{E/2T_{1,2}} \left| \Gamma\left( \frac{\lambda}{2} + i\frac{E}{2\pi T_{1,2}} \right) \right|^2. \tag{F.2}$$

At zero temperature, this expression reduces to

$$P_{1,2}(E)\big|_{T_{1,2}\to 0} = \frac{2\pi \tau_0^\lambda}{\Gamma(\lambda)} E^{\lambda-1}\Theta(E), \tag{F.3}$$

where $\Theta(E)$ is the Heaviside step function. Finally, by comparing to the quasiparticle Green's function (19), we have the Fourier transforms

$$\int_{-\infty}^{\infty} d\tau\, e^{iE\tau} G_{R/L}(\tau) = \frac{1}{2\pi a} P_{1,2}(E). \tag{F.4}$$

# G  Scattering theory for non-interacting electrons

To describe the setup in Fig. 1 in the integer quantum Hall regime, here described by setting $\nu = 1$, we can alternatively use scattering theory, closely following Ref. [20]. The scattering

matrix describing the setup reads

$$s = \begin{pmatrix} 0 & 0 & 0 & 1 \\ 0 & 0 & 1 & 0 \\ t & -r & 0 & 0 \\ r & t & 0 & 0 \end{pmatrix}, \tag{G.1}$$

where the element $s_{\alpha\beta}$ is the amplitude for electron scattering from terminal $\beta$ to $\alpha$. In Eq. (G.1), we have introduced $t$ and $r$ (assumed to be energy independent) as the transmission and reflection amplitude, respectively, at the QPC. It holds that $|t|^2 + |r|^2 \equiv T + R = 1$. Note that the top right corner of $s$ describes ballistic propagation (unit entries) from terminal 4 to 1 and 3 to 2. These entries ensures the unitarity of $s$ as well as fully capturing that the ballistic edge channels propagate along the boundary of a two-dimensional electron gas. *Note that this propagation was not included in Sec. (2). For consistency, we shall therefore neglect these terms in the following.*

With the scattering matrix (G.1), the net charge ($\hat{X} = \hat{I}$) and heat ($\hat{X} = \hat{J}$) current flowing out of terminal $\alpha$ reads

$$\langle \hat{X}_{\alpha,\text{out}} \rangle = \frac{1}{h} \sum_{\beta=1}^{4} \int_{-\infty}^{+\infty} dE \, x_\alpha \left( \delta_{\alpha\beta} - (s_{\alpha\beta})^2 \right) f_\beta(E), \tag{G.2}$$

with $x_\alpha = -e$ for $\hat{X} = \hat{I}$, $x_\alpha = E - \mu_\alpha$ for $\hat{X} = \hat{J}$, and $f_\beta(E) = \{1 + \exp[(E - \mu_\beta)/k_B T_\beta]\}^{-1}$ is the Fermi function in reservoir $\beta$. Likewise, the zero-frequency correlations $S_{\alpha\beta}^{XX}(\omega = 0) \equiv S_{\alpha\beta}^{XX}$ between the $X$ current in terminal $\alpha$ and the $X$ current in terminal $\beta$ read

$$S_{\alpha\beta}^{XX} = \frac{2}{h} \sum_{\gamma,\delta=1}^{4} \int_{-\infty}^{+\infty} dE \, x_\alpha x_\beta \left( \delta_{\alpha\gamma} \delta_{\alpha\delta} - s_{\alpha\gamma} s_{\alpha\delta} \right) \left( \delta_{\beta\delta} \delta_{\beta\gamma} - s_{\beta\delta} s_{\beta\gamma} \right)$$
$$\times \left[ f_\gamma(E)(1 - f_\delta(E)) + f_\delta(E)(1 - f_\gamma(E)) \right]. \tag{G.3}$$

If we further assume that there are no voltage biases in the setup, it is possible to set $\mu_\alpha = \mu_0, \forall \alpha$ and $\mu_0 \equiv 0$ can be taken as energy reference. In such case, $x_\alpha$ loses the dependence on the chemical potential $\mu_\alpha$ and we can just write a single $x = -e$ for $\hat{X} = \hat{I}$ and $x = E$ for $\hat{X} = \hat{J}$. Note that this simplification arises in our setup also for $x_3$ and $x_4$, because terminal 3 and 4 are kept at the reference energy.

Of key interest in this section are the auto-correlation functions in the drain contacts, i.e. $\alpha, \beta = 3, 4$. By using the scattering matrix (G.1) in the correlation function (G.3), we find

$$S_{33}^{XX} = \frac{2}{h} \int_{-\infty}^{+\infty} dE x^2 \Big[ RT(f_1(E) - f_2(E))^2 + T f_1(E)(1 - f_1(E)) + R f_2(E)(1 - f_2(E))) \Big], \tag{G.4}$$

$$S_{44}^{XX} = \frac{2}{h} \int_{-\infty}^{+\infty} dE x^2 \Big[ RT(f_1(E) - f_2(E))^2 + R f_1(E)(1 - f_1(E)) + T f_2(E)(1 - f_2(E)) \Big], \tag{G.5}$$

$$S_{34}^{XX} = S_{43}^{XX} = -\frac{2}{h} \int_{-\infty}^{+\infty} dE x^2 \Big[ RT(f_1(E) - f_2(E))^2 \Big]. \tag{G.6}$$

We see that the correlators (G.4), (G.5), and (G.6) satisfy the conservation laws (6).

Next, we define the charge and heat tunneling currents as

$$\langle \hat{X}_T \rangle \equiv \langle \hat{X}_{1,\text{out}} \rangle - \langle \hat{X}_{3,\text{in}} \rangle. \tag{G.7}$$

Inserting this expression into the noise definition (1) leads to the tunneling noise

$$S_{TT}^{XX} = S_{11}^{XX} + S_{33}^{XX} + 2 S_{31}^{XX}, \tag{G.8}$$

which, via Eq. (G.3), we evaluate as

$$S_{TT}^{XX} = \frac{2}{h} \int_{-\infty}^{+\infty} dE x^2 \Big[ RT(f_1(E) - f_2(E))^2 + R f_1(E)(1 - f_1(E)) + R f_2(E)(1 - f_2(E)) \Big]. \quad \text{(G.9)}$$

Here, we note that the tunneling charge-current noise in the four-terminal setup we are investigating coincides with the total (thermal and shot) noise in a two-terminal setup with reservoirs described by Fermi functions $f_1$ and $f_2$ [20,24]. Similarly, the cross correlation noise $S_{34}^{II}$ coincides with the shot noise component (up to a sign) in the said two-terminal setup [34]. Now, since we assume energy independent tunneling, we can compare Eqs. (G.6) and (G.8) to relate $S_{34}^{XX}$ and $S_{TT}^{XX}$ as

$$S_{TT}^{XX} = -S_{34}^{XX} + R \left( S_{11}^{XX} + S_{22}^{XX} \right). \quad \text{(G.10)}$$

As follows, we are interested in the weak tunneling limit. We thus assume that $R = 1 - T \ll 1$, which we employ as taking

$$R \to D, \qquad RT \to D, \qquad T \to 1, \quad \text{(G.11)}$$

for $D \ll 1$, in the following subsections.

## G.1 Delta-T noise

For the delta-T noise, we have $\hat{X} = \hat{I}$ and $x = -e$. By inserting these specifications, together with the weak tunneling expressions (G.11), into the tunneling noise (G.9), we set $\mu_1 = \mu_2 = 0$, $T_{1/2} = \bar{T} \pm \Delta T / 2$, and then expand in powers of $\Delta T / (2\bar{T})$. We then obtain

$$S_{TT}^{II} = S_0^{II} \times \left[ 1 + \frac{\pi^2 - 6}{9} \left( \frac{\Delta T}{2\bar{T}} \right)^2 + \left( -\frac{7\pi^4}{675} + \frac{\pi^2}{9} - \frac{2}{15} \right) \left( \frac{\Delta T}{2\bar{T}} \right)^4 + \dots \right]. \quad \text{(G.12)}$$

Here, $S_0^{II} = 4e^2 D k_B \bar{T} / h \equiv 4 g_T(\bar{T}) k_B \bar{T}$. Note here that $g_T(\bar{T})$ is independent of $\bar{T}$. We thus obtain the expansion coefficients $\mathcal{C}^{(2)}$ and $\mathcal{C}^{(4)}$ as presented in Eqs. (31a)-(31b).

We repeat the above procedure for the cross correlation noise (G.6) and find

$$S_{34}^{II} = S_{43}^{II} = -S_0^{II} \times \left[ \frac{\pi^2 - 6}{9} \left( \frac{\Delta T}{2\bar{T}} \right)^2 + \left( -\frac{7\pi^4}{675} + \frac{\pi^2}{9} - \frac{2}{15} \right) \left( \frac{\Delta T}{2\bar{T}} \right)^4 + \dots \right]. \quad \text{(G.13)}$$

Upon identification of $\mathcal{C}^{(2)}$ and $\mathcal{C}^{(4)}$, we readily see that the coefficients $\mathcal{D}^{(2)}$ and $\mathcal{D}^{(4)}$ [see Eqs. (30a)-(30b)] both vanish at $\nu = 1$. This result is clear also from a direct comparison between the noises (G.6) and (G.9). In essence, absence of $\mathcal{D}^{(2)}$ and $\mathcal{D}^{(4)}$ follows because in contrast to strongly correlated electrons, the tunneling conductance for free electrons, $g_T(\bar{T}) = e^2 D / h$, does not depend on the temperature $\bar{T}$.

In the large temperature bias limit, $T_1 = T_{\text{hot}}$ and $T_2 \to 0$, the integrals (G.9) and (G.6) evaluate to

$$S_{TT}^{II} = \frac{4e^2 D}{h} k_B T_{\text{hot}} \ln 2, \quad \text{(G.14)}$$

$$S_{34}^{II} = S_{43}^{II} = -\frac{2e^2 D}{h} k_B T_{\text{hot}} (2 \ln 2 - 1), \quad \text{(G.15)}$$

which we obtained in Sec. 3.3 by setting $\lambda = \nu = 1$.

## G.2 Heat-current noise

For the heat-current noise, we have $\hat{X} = \hat{J}$ and $x = E$. Just as for the delta-$T$ noise, we use these specifications, set $\mu_1 = \mu_2 = 0$, assume weak tunneling (G.11), and expand in $\Delta T/(2\bar{T})$ the tunneling noise (G.9) and the cross correlation noise (G.6). We then obtain

$$S_{TT}^{JJ} = S_0^{JJ}\left[1 + \frac{1}{15}(7\pi^2 - 15)\left(\frac{\Delta T}{2\bar{T}}\right)^2 + 2\pi^2\left(\frac{7}{15} - \frac{31}{630}\pi^2\right)\left(\frac{\Delta T}{2\bar{T}}\right)^4 + \ldots\right], \qquad (G.16)$$

$$S_{34}^{JJ} = S_{43}^{JJ} = S_0^{JJ}\left[\frac{1}{15}\left(60 - 7\pi^2\right)\left(\frac{\Delta T}{2\bar{T}}\right)^2 + 2\pi^2\left(-\frac{7}{15} + \frac{31}{630}\pi^2\right)\left(\frac{\Delta T}{2\bar{T}}\right)^4 + \ldots\right]. \quad (G.17)$$

Here, we have identified the equilibrium heat tunneling conductance

$$S_0^{JJ} = \frac{2\pi^2}{3h}Dk_{\mathrm{B}}^3\bar{T}^3. \qquad (G.18)$$

By comparing $S_{TT}^{JJ}$ and $S_{34}^{JJ}$ term by term, we see that our scattering theory is in full agreement with the expansion coefficients (47a)-(47d).

Let us here briefly comment why we have $\mathcal{D}_Q^{(2)} = 3$ for the heat-current noise, in contrast to the delta-$T$ noise where $\mathcal{D}^{(2)} = 0$. If we compare the cross-correlation noise (G.6) to the tunneling noise (G.9), we see that they differ both by a negative sign and that the tunneling noise contains two contributions present even in equilibrium. For the charge-current noise, these parts contribute only to $S_0^{II}$. However for the heat-current noise, these contributions, when expanded in $\Delta T/(2\bar{T})$, produce

$$\begin{aligned} D\left(S_{11}^{JJ} + S_{22}^{JJ}\right) &= \frac{D}{h}\int_{-\infty}^{+\infty} dE E^2\Big[f_1(E)(1-f_1(E)) + f_2(E)(1-f_2(E))\Big] \\ &= \frac{2\pi^2}{3h}Dk_{\mathrm{B}}^3(T_1^3 + T_2^3) \\ &= \frac{2\pi^2}{3h}Dk_{\mathrm{B}}^3\bar{T}^3\left[1 + 3\left(\frac{\Delta T}{2\bar{T}}\right)^2\right]. \end{aligned} \qquad (G.19)$$

We thus see that while the zero-frequency charge-current noise is linear in the temperature $S^{II} \sim k_{\mathrm{B}}\bar{T}$, the heat-current noise is instead cubic: $S^{JJ} \sim (k_{\mathrm{B}}\bar{T})^3$. The reason for this is that for heat flow, the transported quantity depends on the energy (an $E^2$ weight to the Fermi functions) but for the charge flow, the charge $e$ does not depend on the energy. From the result (G.19), we thus see that already the lowest order term in (G.9) contributes a factor of 3 to the $\mathcal{C}^{(2)}$ coefficient. We see that this contribution is absent in the cross-correlation noise and is thus accounted for by the finite $\mathcal{D}_Q^{(2)}$ coefficient.

Finally, we compute the heat-current noise in the large temperature bias limit. We thus take $T_1 = T_{\mathrm{hot}}$ and $T_2 \to 0$, and the integrals (G.9) and (G.6) for the heat-current noise then evaluate to

$$S_{TT}^{JJ} = \frac{3D}{h}(k_{\mathrm{B}}T_{\mathrm{hot}})^3\zeta(3), \qquad (G.20)$$

$$S_{34}^{JJ} = S_{43}^{JJ} = -\frac{3D}{h}(k_{\mathrm{B}}T_{\mathrm{hot}})^3\left(\zeta(3) - \frac{\pi^2}{3}\right), \qquad (G.21)$$

where $\zeta(z)$ is the Riemann zeta-function with $\zeta(3) \approx 1.2$. These results were also obtained in the $\nu = 1$ limit in Sec. 4.2.

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
