# Peer review of "Role of scaling dimensions in generalized noises in fractional quantum Hall tunneling due to a temperature bias"

_SciPost Physics, doi:SciPost Phys. 18, 058 (2025)_

## Round 1 · Referee Report · Anonymous (Referee 1) · 2024-9-8

Strengths
1- treats an important and current subject of research in the physics of nano electronic devices 2- develops an approach to quantum charge and heat transport in fractional quantum Hall quantum point contacts using the standard approach of chiral Luttinger liquids 3- provides a comprehensive overview of noise characteristics of heat and charge transport 4- develops a picture based on an effective density of states and noninteracting tunneling to explain the physics
Weaknesses
1- contains a huge amount of results and it is not clear what is the main focus 2- many results are presented as lengthy formulas and they are not interpreted 3- results are mainly of theoretical interest since quantities like the mixed noise are not measurable currently 4- only results in the perturbative regime of weak backscattering or weak tunneling are presented
Report
In my opinion, the authors present a solid work that deserves to be published more or less in its present form. Regarding the acceptance criteria highlighted by the author, I find it less clear that they are all fulfilled: the manuscript does not really open a new path or research direction, but rather further develops a well-defined and established theoretical approach to study the various noise quantities. It is certainly a contribution to the field of fractional quantum Hall physics, but will most likely not lead to follow-up work outside this field. I'm also not convinced that it represents an important breakthrough of a research stumbling block. The way to relate the rather complicated results of the fully interacting theory to the effective non-interacting picture might have an impact on future research, but is somewhat hidden in the vast amount of results.
Given the strict criteria for acceptance in SciPost Physics, I would recommend that the authors make a substantial revision and highlight the main results in a compact form at the beginning. It is important that this summary be comprehensible to readers who are not specialists in chiral Luttinger liquids. Otherwise, the manuscript in its current form is acceptable for SciPost Physics Core, where it could be published without further review.
Recommendation
Ask for major revision
We thank the Referee for the overall positive assessment of the quality our work, in particular that it is "a solid work that deserves to be published more or less in its present form". We would like to reply here in depth to the Referee's concerns about the relevance of our paper considering the journal's acceptance criteria.
The determination of scaling dimensions is an important problem in strongly correlated, one-dimensional systems, and in the FQH effect it has been a notorious stumbling block to match experimental results with theoretical predictions. Clearly, solving this problem calls for models that could explain the discrepancies, an avenue that has been explored in some detail in previous works [e.g., Phys. Rev. Lett. 88, 096404 (2002)]. At the same time, it is critical to assemble a broad range of complementary physical observables that can probe the scaling dimensions. This latter need is the main motivation for our work, and the spirit with which we present our results: highlighting a set of complementary tools that can be used to extract scaling dimensions in the FQH effect. In doing so, we have provided a systematic analysis of several noise observables under general conditions with temperature and voltage biases, which has been missing in the FQH literature. As acknowledged by the second Referee, our detailed analysis paves the way for detailed comparison to both state-of-the-art and future experiments.
Moreover, our work extends naturally towards using temperature-biased noise to identify the scaling dimension in a variety of tunneling setups. The most relevant and important extensions involve probing more complicated edge structures with non-Abelian character, where the presence of neutral modes requires the use of charge-insensitive observables, like the heat-current fluctuations that we have analyzed here. We remark here that several non-Abelian candidate edge structures, e.g., at filling $\nu=5/2$, involve such neutral modes. Having at hand tools to probe their presence is thus an important step towards distinguishing between such candidates and realizing non-Abelian anyons, a major goal in the broader physics community. Our work thus sets the stage to compute temperature-biased noise for such edges. Furthermore, our work can be adapted to related systems of interest, like quantum wires and quantum spin Hall edges, where the quantities presented are of clear relevance.
Following the Referee's recommendation, we have rewritten both the introduction and the outlook sections of the manuscript. We think that these improvements provide non-expert readers with a general understanding of the main results, together with the appropriate context in which they find applications.
As for the introduction, we made two main modifications: - We have provided a more complete and accessible description of the key physical quantities of interest (particularly the scaling dimension), together with a broader context explaining why the determination of this parameter is both important and challenging. - We have made a clearer summary of our results in a list format, with references to key equations, allowing readers to more easily navigate the main results.
In the outlook, we now explicitly discuss the advantages and disadvantages of the different types of noise that we address, suggesting possible follow-up work, and we also comment in more detail on the experimental relevance of our predictions.
With these modifications, we are confident that the main message of the work is both clearer and more accessible to a non-expert readership. We are therefore convinced that the work meets the criteria of Scipost Physics.

Author: Matteo Acciai on 2024-11-21 [id 4977]
(in reply to Report 2 by Kyrylo Snizhko on 2024-09-23)We thank the Referee for the very positive assessment of our work, in particular that it "is of very high quality".
Given that the criticism is mainly concerned with the journal's acceptance criteria and is thus similar to that of the first Referee, we refer to our response to Report#1, where we reply to this criticism in detail.
Here, we reply to the second Referee's listed points with requested changes: 1. We have implemented this change, below Eq. (28). 2. We have added the reference suggested by the Referee. 3. We thank the Referee for spotting this mistake. The formula is now correct. 4. The analogy works in both cases. We have modified the text below Eq. (70) to emphasize that both (77) and (68-69) can indeed be written as a standard Landauer Büttiker formula, with an appropriate energy-dependent effective transmission which is due to the correlated nature of the system. The main feature that we exploit in these formulas is that the particle-hole symmetry of the FQH edge Hamiltonian produces effective "transmission functions" that are even in energy. 5. The duplicate reference has been removed.

---

## Round 1 · Referee Report · Kyrylo Snizhko (Referee 2) · 2024-9-23

Strengths
- Systematic consideration of:
- various observables;
- in both limiting regimes;
- of a general class of models.
- Detailed theoretical expressions. Useful for future reference.
Weaknesses
- No clear punchline. It is not clear which of the observables show the most promise.
Report
Further, the authors systematically introduce an effective single-particle picture, where the tunnelling properties of fractional quasiparticles are represented in terms of a Fermi distribution and power-law density of states.
Despite the high quality of the work, I do not feel it meets the SciPost acceptance criteria.
The authors state that their work meets two:
(i) Present a breakthrough on a previously-identified and long-standing research stumbling block;
(ii) Open a new pathway in an existing or a new research direction, with clear potential for multi-pronged follow-up work.
Concerning (i), I fail to see a long-standing stumbling block in the paper's consideration.
Concerning (ii). The paper does bring a systematic consideration of the scaling dimension into the temperature bias studies - which is an important aspect previously absent for the works in the area. However, the potential for follow-up work is rather narrow: performing experiments (potentially, on various observables considered in the work).
The paper may barely surpass criterion (ii), yet I do not see it opening a large space for thought.
However, I feel significantly more comfortable recommending the work to be published in SciPost Physics Core. It easily meets both acceptance criteria: addressing an important problem with a high degree of originality and significantly advance the knowledge in the field.
This will be an important reference work in the future for the people interested to perform one or another experiment and compare it to the theoretical predictions.
Requested changes
-
In the discussion of $|\mathcal{C}^{(4)}| \ll |\mathcal{C}^{(2)}|$ after Eq. (28), this relation is not evident from the expressions. It would be appropriate to refer the reader to Fig. 2.
-
When citing [77-79], it would be appropriate to cite also https://link.aps.org/doi/10.1103/PhysRevB.61.10929. This is the earliest work on the topic that I know.
-
In Eq. (67), the formula needs to be amended to account for the possibility of negative energies.
-
The expression in Eq. (77) looks like a conventional Landauer-Buttiker formula: all the Fermi functions and DOS correspond to the energy $E$, offset by the appropriate chemical potentials. At the same time, Eqs. (68-69) include integrands evaluated at $E$ and $-E$, which ruins the analogy with the Landauer-Buttiker formalism. I ask the authors to check the validity of these formulas; and if the mismatch persists, to comment about it in the text.
-
Ref. 102 is the same as Ref. 31. Please remove the duplicate.
Recommendation
Accept in alternative Journal (see Report)

---

## Round 2 · Referee Report · Kyrylo Snizhko (Referee 2) · 2024-12-5

Strengths

1- Detailed and comprehensive consideration

Weaknesses

1- Lack of definite focus point 2- For the purpose of experimental follow-up, selecting a specific observable and regime to focus on would be hard.

Report

I thank the authors for addressing my comments and for the effort put in rewriting the Introduction and the conclusions.

Nevertheless, I still find that the paper does not meet SciPost acceptance criteria. Referring to the criteria discussed previously and addressed in the response by the authors.

$\textit{(i) Present a breakthrough on a previously-identified and long-standing research stumbling block.}$
Discrepancies between the predicted and the observed scaling dimensions in fractional quantum Hall edges is indeed a long-standing problem. And the authors do fulfil an important mission of providing various results for its determination in one place. Nevertheless, this is not a breakthrough, only a list of possibilities. An important list. Yet, it does not provide a way for quick significant steps in resolving the problem.

$\textit{(ii) Open a new pathway in an existing or a new research direction, with clear potential for multi-pronged follow-up work.}$
Indeed, there are many possible experiments that can be performed to measure the predictions of this work. Nevertheless, it is not clear that any of such future experiments would provide a qualitatively better determination of the scaling dimension than is currently available.
While the authors propose an extended and rather comprehensive list of results, there is no clear observable or regime that promises to provide more information.
Therefore, the paper can be a basis for systematic explorative investigation (which is an important part of science!). Nevertheless, it is only likely to stimulate specialists with a focused interest. In its present form, the paper appeals principally to the readers possessing high motivation.

Therefore, my recommendation is to publish the paper in SciPost Physics Core, for which the paper meets all acceptance criteria.

Requested changes

1- In the phrase "By using this symmetry, we further express the tunneling charge noise (68)..." before Eq. (70), it is worth stating that the relation $f(-E)=1-f(E)$ has been used.

Recommendation

Accept in alternative Journal (see Report)

---

## Round 2 · Referee Report · Anonymous (Referee 1) · 2024-12-31

Report

The author have significantly improved the presentation by including an executive summary at the beginning providing a guide through the rather lengthy paper. I keep up my main comments of the first report that the manuscript is of significant importance for the field of noise correlation in fractional quantum Hall edges. In this field the results are promising and will most likely lead to follow-up work. E.g. the fact that the cross-current noise contains information that is not available otherwise can be seen as a breakthrough. Or, the observation that an effective density-of-states picture might be useful for quantum transport in correlated system in general can lead to new research directions in other fields. As such the acceptance criteria of SciPost Physics are met and I recommend publication.

Recommendation

Publish (meets expectations and criteria for this Journal)

---

## Round 2 · Author Response

We thank the Editors for processing our submission and the Referees for the useful feedback on our work.
We provide a detailed response to their criticism in the replies below the reports.
All major modifications in the resubmitted manuscript are marked in red.
We are confident that our revised work meets the expectations for publication.

---

## Round 2 · List of Changes

• An extensive part of the introduction has been rewritten, presenting a compact summary of the main results, with references to key equations.
  • The outlook section has been rewritten, better emphasizing directions for future research and discussing advantages and disadvantages of the observables we studied.
  • A reference to Fig. 2a has been added in the discussion below Equation (28).
  • Equation (67) has been corrected.
  • A sentence has been added below Equation (70).
  • A sentence has been added below Equation (77) .
  • Some typos have been corrected.
  • A couple of references have been added.

---

## Editorial Decision

published